# Florigen activation complex forms via multifaceted assembly in *Arabidopsis*

He Gao[1,5,6✉], Na Ding[1,6], Yuang Wu[1], Dongli Yu[1], Shi-Zhao Zhou[2], Sara Christina Stolze[1], Coral Vincent[1], Gabriel Rodríguez Maroto[1], Pedro de los Reyes[1], Anne Harzen[1], Martina Cerise[1], Vítor da Silveira Falavigna[1], Ertong Li[1,3], Ton Timmers[1], Ulla Neumann[1], Hirofumi Nakagami[1], Jin-Yong Hu[2], Jijie Chai[1,4] & George Coupland[1✉]

Florigen, encoded by *FT* genes, is synthesized in leaves and transported to the shoot apical meristem (SAM) to induce flower development[1–3]. At the SAM, 14-3-3 proteins are proposed to act as receptors for FT protein and to mediate the indirect interaction between FT and the basic leucine zipper (bZIP) transcription factor FD to form the florigen activation complex (FAC) that activates transcription of flowering genes[4–6]. Here we demonstrate a different mechanism of FAC assembly, diverse functions for the 14-3-3 proteins within the complex, and an unexpected spatiotemporal distribution of the FAC. We show that FT is not recruited by 14-3-3 alone, but that it interacts with the DNA–FD–14-3-3 complex through two interfaces, one of which binds DNA via the unstructured C terminus of FT. We also find that interaction of 14-3-3 proteins with the C terminus of phosphorylated FD reduces liquid phase condensation of the intrinsically disordered FD protein, allowing it to bind DNA, and that the 14-3-3 proteins strengthen DNA binding of FD by promoting dimerization, which ultimately results in the recruitment of FT. Unexpectedly, we also find that after FT movement to the shoot apex, *FT* and *FD* are co-transcribed in young floral primordia, forming a boundary with the suppressed bract and allowing formation of the FAC during the first stages of floral development. Our studies propose a new mechanism by which the florigen FT transcriptional complex is formed, and indicate distinct functions for the complex during SAM and floral primordium development.

The seasonal cue of day length synchronizes plant developmental programmes to the changing seasons, particularly the initiation of flowering, tuberization in potato and seasonal growth in trees[1,7–10]. All of these processes rely on a common regulatory system that initiates synthesis of FLOWERING LOCUS T (FT) proteins in the leaves. FT is related to phosphatidylethanolamine-binding proteins (PEBP) and is transmitted through the phloem vascular system from the leaves to the SAM[2,3,11–14], where it triggers the floral transition by conferring floral identity on primordia[2,3,12–14]. This movement of FT to the SAM led to the proposal that it is a component of the classical flowering hormone florigen. At the SAM of rice, 14-3-3 proteins are proposed to act as receptors for the FT homologue Heading date 3a (Hd3a), yielding a complex that is transported into the nucleus where it binds to a highly conserved bZIP transcription factor FD[4–6]. The resultant complex containing FT, 14-3-3 and FD is referred to as the FAC, which relies on FD to recognize genomic binding sites[5,6,15–17], whereas the recruitment of FT activates the transcription of genes that promote flowering[4,18,19]. By contrast, recruitment of the anti-florigen TERMINAL FLOWER 1 (TFL1), a homologue of FT, to 14-3-3 proteins and FD forms the florigen repression complex (FRC), inhibiting transcription of FD target genes and delaying flowering[17,20]. The 14-3-3 proteins are highly conserved in eukaryotes, and interact with phosphorylated residues in a broad range of protein clients[21–24]. Of note, they were recently proposed to act as chaperone-like proteins that can suppress the phase separation of client proteins in human cells[21]. However, the relevance of this observation for their role as FT receptors in mediating the formation of the FAC is unknown. Notably, despite FT protein movement having been established some time ago, the spatiotemporal accumulation of FT at the SAM relative to other FAC components, particularly when expressed from their endogenous regulatory sequences, remains unclear[2–4,13,14]. Here, we provide a biochemical and cellular model for how FAC components cooperate to regulate activity of the complex and induce flowering.

## *FT*–*FD* co-transcription in floral primordia

To follow the distribution of FT, a functional *gFT::FT-mVenus* genomic fusion containing the entire upstream, coding and downstream regulatory regions was introduced into *ft-10* mutants (Extended Data Fig. 1a–c, Supplementary Table 1 and Methods). FT–mVenus accumulation was first observed in the minor veins at the tips of cotyledons, and later in the major veins throughout the cotyledons (Extended Data Fig. 1d,e).

[1]Max Planck Institute for Plant Breeding Research, Cologne, Germany. [2]College of Horticulture Sciences and Engineering, Shandong Agricultural University, Tai'an, China. [3]School of Pharmaceutical Sciences, Zhengzhou University, Zhengzhou, China. [4]School of Life Sciences, WESTLAKE University, Hangzhou, China. [5]Present address: Institute of Genetics and Developmental Biology, Chinese Academy of Sciences, Beijing, China. [6]These authors contributed equally: He Gao, Na Ding. ✉e-mail: hgao@genetics.ac.cn; coupland@mpipz.mpg.de

FT−mVenus signals were found to overlap with the companion cell marker, *SUC2*::PP2A-mCherry (Extended Data Fig. 1f,g,i), supporting previous findings[25]. Accordingly, a similar pattern was observed in a *gFT::FT-Venus-Halo-Venus* line, in which the FT fusion protein is too large to be transported and does not complement *ft-10* (Extended Data Fig. 1a–c, j and Supplementary Table 1). Nevertheless, when expressed in the SAM from the *FD* promoter, this large fusion protein complemented *ft-10*, indicating that it retains FT function (Extended Data Fig. 1b,c). These observations support the idea that FT synthesis takes place in companion cells in the minor and major veins, and that movement between cells is required for its activity. Subsequently, *SUC2::PP2A-mCherry* signal was detected at the end of the vasculature close to the SAM at 10 long-day (LD; 16 h light:8 h dark) cycles, and FT−mVenus was also detected in those cells, but in addition extended beyond the end of the vasculature towards young primordia and the SAM (Extended Data Fig. 1h). The signal beyond the vasculature represents movement of FT−mVenus, and at least some of the FT−mVenus signal at the junction of the vasculature and provasculature at the shoot apex also represents transported protein. The presence of FT at these locations at 10 LD coincides with the earliest stages of floral transition.

We next compared the spatiotemporal accumulation of FT and the other FAC components in the SAM. 14-3-3 proteins were widely distributed in apical tissue in *14-3-3::mScarlet1-14-3-3* genomic fusion transgenic plants, and therefore their distribution overlaps with FT and FD (Extended Data Fig. 2a–e). To compare the distributions of FT and FD, *gFT::mVenus* was crossed with a functional *gFD::mScarlet1-FD* line (Supplementary Table 1). In these plants, FT−mVenus was observed in the provasculature near the shoot apex at 10 LD during the early stages of floral induction, and at 11 LD, 12 LD and 13 LD, the FT−mVenus protein could be observed in the rib region and organizing centre of the SAM (Fig. 1a–i and Supplementary Fig. 1a–p). To quantify the overlap between FT−mVenus and mScarlet1−FD, nuclei expressing mScarlet1−FD were segmented (Fig. 1j and Supplementary Fig. 2) and the intensity of FT−mVenus fluorescence in each of these nuclei was quantified (Fig. 1k and Supplementary Fig. 1q,r). More nuclei expressing both proteins were detected in the rib region, organizing centre, peripheral zone and to a lesser extent in the central zone at 11 LD, 12 LD and 13 LD than at 10 LD (Fig. 1k). Moreover, the intensity of FT−mVenus fluorescence per square micrometre was consistently higher in nuclei expressing mScarlet1-FD at 11 LD, 12 LD and 13 LD than at 10 LD in all meristematic regions (Fig. 1k). Therefore, the FT and FD proteins are present in the same nuclei in the SAM, and broadly overlap with 14-3-3 proteins, allowing the formation of the FAC in the SAM during floral induction.

Next, we assessed the accumulation pattern of the immobile *gFT::FT-Venus-Halo-Venus* signal within the SAM. At early stages the signal was absent in the vasculature at the SAM base (12 LD and 13 LD; Fig. 1l,m), but it was strongly detected on the adaxial side of cauline leaves in the axils of cauline leaves at 14 LD and in primordia of plants at 15 LDs as they initiate floral development (Fig. 1n,o). These findings suggest that *FT* has two temporally and spatially separated patterns of expression: initially, it is transcribed in leaf vasculature and FT protein moves from leaves to the SAM; subsequently, local *FT* transcription occurs within primordia.

Accordingly, RNA in situ hybridization detected *FT-mVenus* mRNA in the same characteristic pattern during floral transition and in older inflorescences (Extended Data Fig. 3a). To simultaneously compare *FT* mRNA localization in primordia with mRNAs of *FD* and the anti-florigen *TFL1*, which acts antagonistically to *FT*[26–28], we performed fluorescence-based, multiplexed RNA in situ hybridizations (RNAscope)[29]. At 10 LD, *FT* mRNA was not detected at the shoot apex, although *FD* and *TFL1* mRNAs were observed at the SAM (Fig. 2a). However, at 13 LD, as floral transition proceeded, *TFL1* mRNA appeared strongly in axillary meristems and *FT* mRNA appeared nearby on the adaxial side of the cauline leaf (Fig. 2b and Extended Data Fig. 3b).

As floral transition proceeded, we analysed the mRNAs in the first visible floral primordium on the flank of the SAM. At 14 LD, *FT* and *FD* were co-expressed on the adaxial side of the first floral primordium and *TFL1* mRNA was also detected in a similar pattern to *FT* and *FD* mRNAs (Fig. 2c,h). At 15 LD, *FT* mRNA was present in a strip of cells across the primordium, and *TFL1* mRNA was still present in the primordium, above the *FT*-expressing cells (Fig. 2i and Extended Data Fig. 3c). At 16 LD, floral transition was complete, *TFL1* mRNA was not detected in the primordium, and *FT* mRNA was expressed as a boundary across the floral primordium (Fig. 2d,j and Extended Data Fig. 3d,e). At this stage, *FD* mRNA is also present throughout the primordium. To assess the position of *FT* mRNA in the floral primordium more accurately, combinations of *FT*, *FD*, *APETALA1* (*AP1*) (Fig. 2k,o) and *FT*, *TFL1* and *AP1* (Fig. 2m,p) probes were used on 16 LD plants. *FT* mRNA was present across the base of the domain of expression of *AP1* mRNA in the boundary between the floral primordium and the suppressed bract[30,31] (Fig. 2k,o,q). Quantification of these images showed that *FT* mRNA levels are highest in regions corresponding to primordia of the 14 LD and 16 LD apices (130–140 μm from SAM tip at 16 LD). *FD* mRNA overlapped with *FT* in these deeper regions at 14 LD and 16 LD (Fig. 2e,f,h,j), and *FT* mRNA was located below *AP1* mRNA in the primordium (Fig. 2l). By contrast, the highest levels of *TFL1* mRNA were in the SAM in the mature inflorescence at 16 LD (Fig. 2m), and at 14 LD and 16 LD in deeper tissues corresponding to the axillary meristems (Fig. 2g,n). Similar patterns of endogenous *FT* mRNA along with *FD* and *TFL1* mRNAs were found in wild-type Col-0 plants (Extended Data Fig. 3f,g). Therefore, as the apex proceeds to floral transition, *FT* mRNA levels increase, and they accumulate at the boundary of newly formed primordia first on the adaxial side of cauline leaves, then at later nodes on the adaxial side of the floral primordium, and as the domain of *AP1* expression expands, *FT* mRNA is localized at the boundary between the primordium and the suppressed bract (Fig. 2q). Collectively, our protein and mRNA analyses demonstrate that FT is present in the same cells as FD and 14-3-3 proteins at different stages of the flowering process to enable formation of the FAC.

## DNA binding by FD−14-3-3 recruits FT

We next analysed FAC biochemistry in vivo by immunoprecipitation–mass spectrometry (IP–MS) and in vitro using proteins purified from *Escherichia coli*. IP–MS identified ten 14-3-3 proteins as interactors with FD using functional *gFD::3HA-mCherry-FD;fd-3* transgenic seedlings[32] (Supplementary Table 2). FT was not identified, suggesting that FT interacts only weakly or transiently with the FD−14-3-3 complex or that it is present below the detection threshold of IP–MS. To reconstitute how *Arabidopsis* 14-3-3 proteins link FD and FT in vitro, we purified recombinant FD, 14-3-3 and FT proteins from *E. coli*. Efficient phosphorylation of FD did not occur in *E. coli* cells (Supplementary Table 2), therefore, we purified the maltose-binding protein (MBP)-fused phospho-mimic mutant FD (T282E). Despite the observed interactions between rice Hd3a and 14-3-3 proteins[4], gel-filtration analysis detected no direct interaction between FT and *Arabidopsis* GRF7 (14-3-3ν) (Fig. 3a) or four other 14-3-3 isoforms (Extended Data Fig. 4a–d). Despite the stable formation of the MBP−FD(T282E)−GRF7 complex, no recruitment of FT was detected (Fig. 3b–d). These in vitro results suggest that the interaction between *Arabidopsis* 14-3-3 and FT is weak or unstable, and that another component might be required for FT recruitment to the complex.

FD guides the FAC to specific genomic binding sites[5,6,15–17]. Therefore, we tested whether the DNA-bound FD−14-3-3 complex can recruit FT. FT recruitment was detected by gel-filtration analysis in the presence of a DNA fragment containing a segment of the *SEPALLATA3* (*SEP3*) promoter, an FD target gene[15–17], along with MBP−FD(T282E) and GRF7 proteins (Fig. 3e and Extended Data Fig. 4i). We hypothesized that binding of the FD−14-3-3 complex to DNA may create interfaces that are

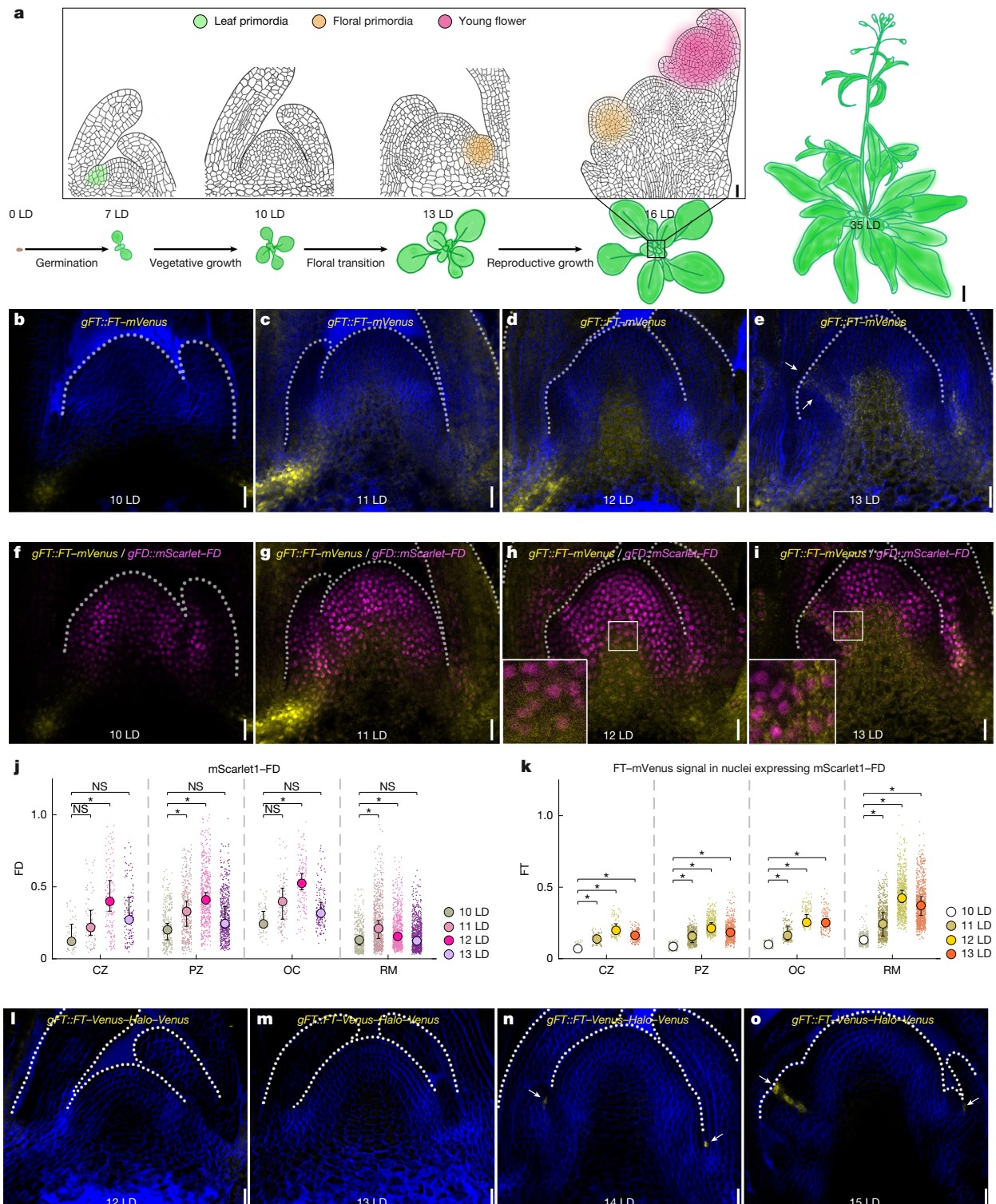

**Fig. 1 | Distribution of FT and co-localization with FD at the SAM during floral transition. a**, Schematic depicting the timing of developmental transitions in *Arabidopsis*. Scale bars, 20 μm (cross-sections; inset) and 1 cm (plant images). **b**–**i**, Confocal images of SAM cells co-expressing *gFT::FT–mVenus* at 10 LD (**b**), 11 LD (**c**), 12 LD (**d**) and 13 LD (**e**) and *gFD::mScarlet1–FD* at 10 LD (**f**), 11 LD (**g**), 12 LD (**h**) and 13 LD (**i**). Insets in **h**,**i** show higher magnification of the outlined regions. **j**,**k**, Nuclear concentration of FD (**j**) and FT (**k**). Nuclear concentration was calculated as the total pixel intensity within a segmented nucleus divided by its area. Measurements were normalized by the maximum FD and FT concentrations among all analysed meristems. Small dots represent FD and FT concentrations in individual nuclei. Large circles denote the median concentration of FD or FT for each SAM at a given time point. Medians were

compared across time points (*n* = 3, 4, 4 and 4 for 10 LD, 11 LD, 12 LD and 13 LD, respectively). Error bars represent the interquartile range for the previous median distribution. Statistical differences in median nuclear concentration between 10 LD and 11 LD, 12 LD, or 13 LD were assessed using the Brunner–Munzel test (*α* = 0.1), a non-parametric test for median comparison that is robust to small sample sizes, with two-sided pairwise comparisons. CZ, central zone; OC, organizing centre; PZ, peripheral zone; RM, rib meristem. **l**–**o**, Confocal images of SAM cells expressing *gFT::FT–Venus–Halo–Venus*; Col-0 at 12 LD (**l**), 13 LD (**m**), 14 LD (**n**) and 15 LD (**o**). Arrows in **e**,**n**,**o** indicate FT fluorescent protein signal. **b**–**e**,**l**–**o**, Cell walls (blue) were stained with Renaissance 2200. **b**–**i**,**l**–**o**, Scale bars, 20 μm. Confocal images in **b**–**i**,**l**–**o** are representative of three independent meristems. *P < 0.05; NS, not significant (*P* ≥ 0.05).

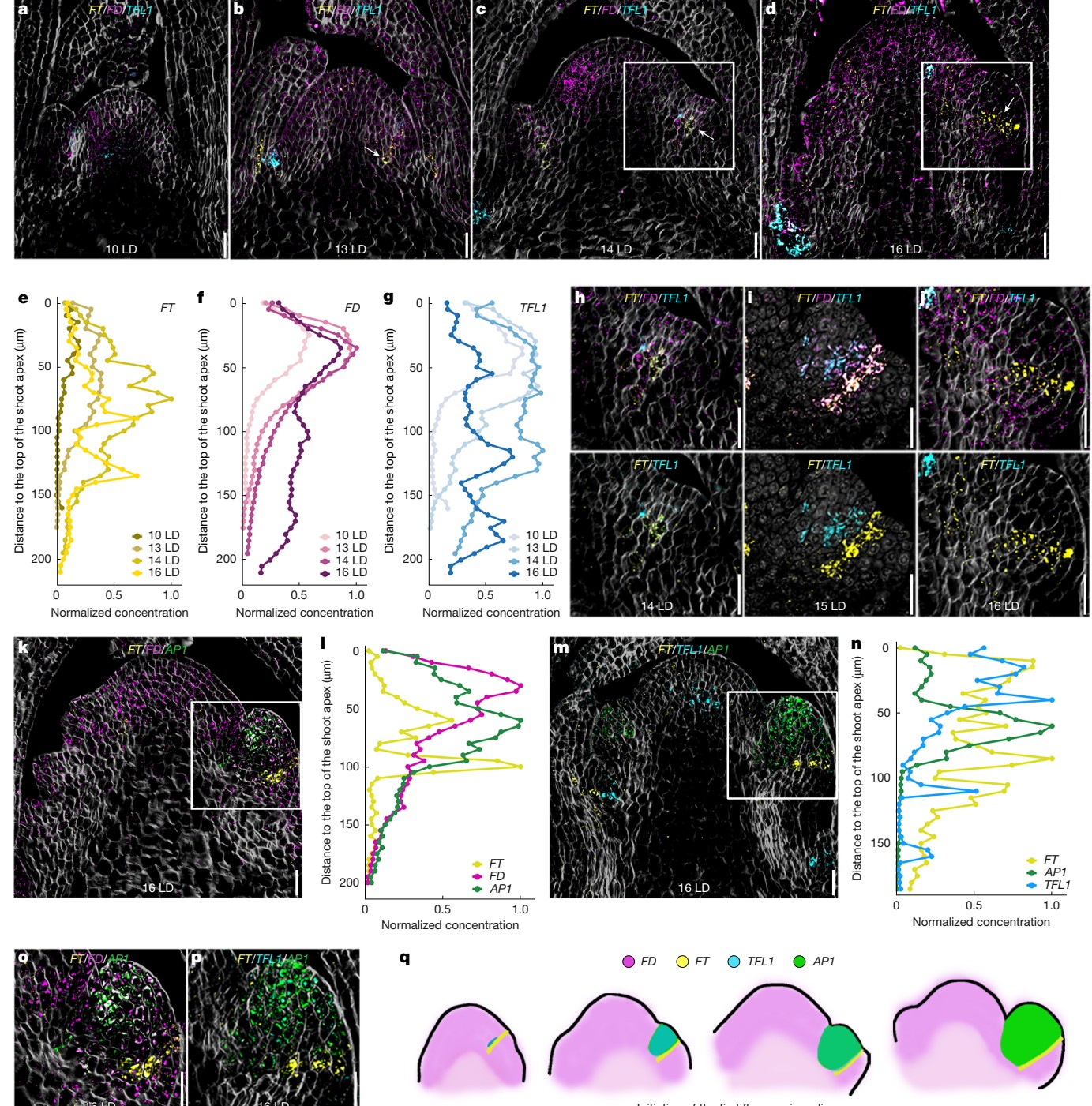

**Fig. 2 | Distribution of *FT* mRNA and co-localization with *FD* and *TFL1* mRNA at the SAM before, during and after floral transition. a**–**d**, Confocal images of SAM cells co-expressing *gFT::FT-mVenus* transgene, endogenous *FD* and *TFL1*, captured by RNAscope at 10 LD (**a**), 13 LD (**b**), 14 LD (**c**) and 16 LD (**d**). Arrows indicate cells that express *FT-mVenus*. Probes were used for *FT*, *FD* and *TFL1* mRNAs. **e**–**g**, Spatiotemporal quantification of *FT* (**e**), *FD* (**f**) and *TFL1* (**g**) mRNA accumulation at the SAM along the longitudinal axis from 10 LD to 16 LD from images represented in **a**–**d**. Each point denotes the reporter intensity within a 10 μm³ volume. Intensity profiles are normalized to the peak value measured for each reporter (Methods). **h**–**j**, Close-up images of floral primordia cells co-expressing *gFT::FT-mVenus* and endogenous *FD* and *TFL1* mRNA at 14 LD (**h**), 15 LD (**i**) and 16 LD (**j**); nuclei in **i** were staind with DAPI. **k**,**m**, Confocal

images of a shoot apex co-expressing *gFT::FT-mVenus* and endogenous *FD* and *AP1* (**k**) and endogenous *TFL1* and *AP1* (**m**) at 16 LD. **l**,**n** The spatiotemporal accumulation at 16 LD along the longitudinal axis of the SAM of *FT*, *FD* and *AP1* mRNA signals shown in **k** is quantified in **l**, and the *FT*, *FD* and *AP1* mRNA signals shown in **m** are quantified in **n**. **o**,**p**, Close-up images of floral primordia cells co-expressing *gFT::FT-mVenus* and endogenous *FD* and *AP1* (**o**) and endogenous *TFL1* and *AP1* (**p**) at 16 LD in images represented in **k**,**m**. **q**, SAM regions where *FT*, *FD*, *TFL1* and *AP1* mRNAs are transcribed during the initiation of floral transition. **a**–**d**,**h**,**j**,**k**,**m**, Cell walls were stained with Renaissance 2200 (light grey). **i**, Nuclei were stained with DAPI (light grey). **a**–**d**,**h**–**k**,**m**,**o**,**p**, Scale bars, 20 μm. RNAscope experiments in **a**–**d**,**h**–**k**,**m**,**o**,**p** are representative of three independent meristems.

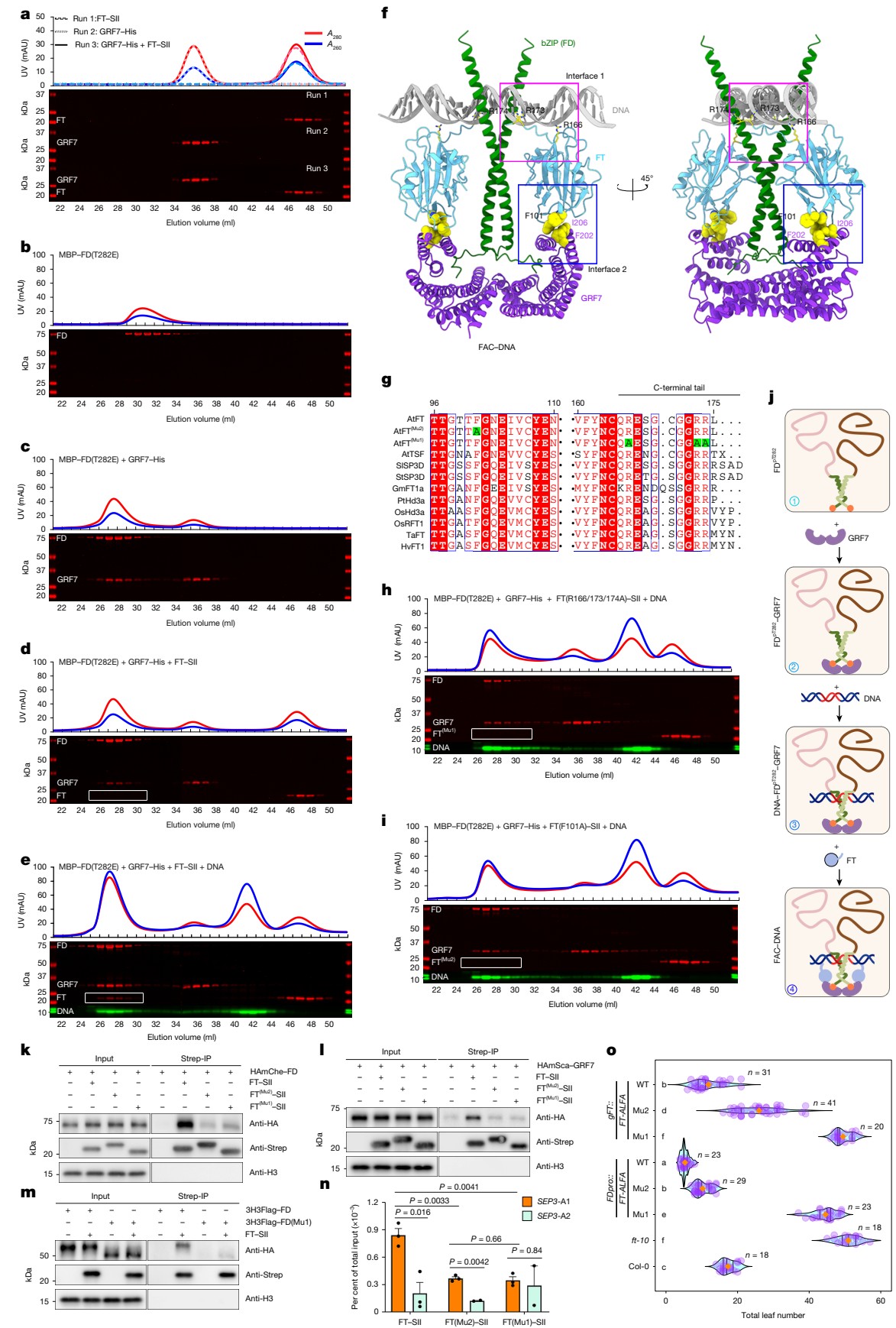

**Fig. 3 |** See next page for caption.

**Fig. 3 | The critical role of DNA binding in FT recruitment by the FD−14-3-3 complex. a–e**, Size-exclusion chromatography and gel analysis of GRF7 and FT proteins (**a**), MBP–FD(T282E) (**b**), MBP–FD(T282E) and GRF7 (**c**), MBP–FD(T282E), GRF7 and FT (**d**) and MBP–FD(T282E), GRF7 and FT proteins in the presence of a 24-bp *SEP3* DNA fragment (**e**). mAU, milli-absorbance unit; UV, ultraviolet. **f**, Modelled structure of the FAC–DNA complex. **g**, Alignment of FT proteins from *Arabidopsis* and some major crops, including FT mutations (Mu1 and Mu2). **h,i**, Size-exclusion chromatography and gel analysis of MBP–FD(T282E), GRF7 and *SEP3* DNA with FT(Mu1) (**h**) or FT(Mu2) (**i**) proteins. Gel-filtration assays in **a–e,h,i** are representative of two independent experiments. **j**, Schematic illustrating the step-by-step assembly of the FAC–DNA complex. FD$^{pT282}$ is phosphorylated at T282. **k,l**, Co-immunoprecipitation of purified wild-type and mutant FT–SII proteins with 3HA–mCherry–FD (HAmChe–FD; **k**) and 2HA–mScarlet1–GRF7 (HAmSca–GRF7; **l**). HA, haemagglutinin; Strep-IP, immunoprecipitation of Strep tag. **m**, Purified wild-type FT–SII protein

co-immunoprecipitates with 3HA–3Flag–FD (3H3Flag–FD) and 3HA–3Flag–FD(Mu1) (3H3Flag–FD(Mu1)) proteins. Co-immunoprecipitation experiments in **k–m** were performed with nuclear proteins extracted from 12-day-old seedlings grown in long-day conditions and are representative of two independent experiments. **n**, ChIP–qPCR analysis of purified wild-type and mutant FT–SII proteins binding to the *SEP3* promoter in nuclear extracts from 10-day-old *gFD::3HA-mCherry-FD;fd-3* seedlings grown in long-day conditions (16 h light, 8 h dark). Statistical significance was determined by pairwise two-sided *t*-test. Data are mean ± s.e.m. of three biological replicates (*n* = 3). **o**, Flowering time (measured as total leaf number) of transgenic *ft-10* plants grown in long-day conditions and carrying *gFT::FT-ALFA* or *FD::FT-ALFA* with wild-type or mutant *FT*. Letters indicate significant differences between genotypes (*P* < 0.05), using one-way ANOVA followed by Tukey's pairwise multiple comparison. The overall ANOVA result was significant (*P* = 1.11 × 10$^{-16}$).

directly recognized by FT, even though FT is not proposed to contact FD and/or DNA in current models of the FAC[4]. AlphaFold predicted an unstructured C-terminal tail for FT, which is highly conserved among FT homologues[4,33] (Fig. 3f,g and Extended Data Fig. 4e). Although this tail was removed for in vitro protein crystallography[4,27,34] and is not considered important for FT activity in vivo, overexpression of *FT* cDNA with random mutations revealed that substituting a glutamic acid for arginine at residue 173 (R173E) in the tail reduced FT activity[33]. We modelled the FAC–DNA complex, which suggested direct contact between three positively charged arginine residues (R166, R173 and R174) in the FT tail and negatively charged DNA (interface 1; Fig. 3f,g). Mutating all of these residues in FT (Mu1 (R166A/R173A/R174A)) abolished FT recruitment by the DNA−FD−14-3-3 complex in vitro (Fig. 3h). Therefore, the C-terminal tail of FT, which we call interface 1, is involved in FT recruitment to the DNA−FD−14-3-3 complex by directly interacting with DNA. Our model also suggested that the amino acids that are crucial for the interaction between rice Hd3a and 14-3-3 proteins (GF14b and GF14c) are required in FT for assembling the *Arabidopsis* FAC, and are referred to as interface 2 (Fig. 3f and Extended Data Fig. 4e–g). Mutation of the conserved amino acids in FT (Mu2 (F101A)) or GRF7 (Mu1 (F202A/I206A)) in interface 2 prevented recruitment of FT by the DNA−FD−14-3-3 complex (Fig. 3i and Extended Data Fig. 4h,i). Electrophoretic mobility shift assays (EMSA) validated the new FAC model on *SEP3* and another FD target *LEAFY*, with findings largely consistent with the results from gel filtration (Extended Data Fig. 4j,k and Supplementary information).

Next, the recruitment of wild-type and mutant FT proteins (Mu1 and Mu2) by the DNA−FD−14-3-3 complex was examined using nuclear protein extracts made from *gFD::3HA-mCherry-FD*; *fd-3* seedlings. We co-immunoprecipitated 3HA–mCherry–FD from these extracts with FT–Strep-tag (FT–SII) protein produced in *E. coli*, but the interaction was reduced when FT(Mu1)–SII or FT(Mu2)–SII proteins were used (Fig. 3k). Similarly, 2HA–mScarlet1–GRF7 was more efficiently co-immunoprecipitated with wild-type FT–SII than with FT(Mu1)–SII or FT(Mu2)–SII in nuclear protein extracts of *gGRF7::2HA-mScarlet1-GRF7* plants (Fig. 3l). Thus, the impairment of the interaction of FT with DNA in FT(Mu1)–SII also impairs its recruitment to the FAC. To further examine the dependency of FD−DNA binding on this interaction, nuclear protein extracts of g*FD::3HA-3Flag-FD$^{Mu1}$* plants expressing a basic domain mutant of FD that did not bind DNA were used (Extended Data Fig. 5 and Supplementary information). The 3HA–3Flag–FD(Mu1) protein was co-immunoprecipitated with wild-type FT–SII at lower efficiency than 3HA–3Flag–FD(WT) (Fig. 3m). Consistent with the protein co-immunoprecipitation results, chromatin immunoprecipitation with quantitative real-time PCR (ChIP–qPCR) analysis showed that in nuclear chromatin extracts of *gFD::3HA-mCherry-FD;fd-3* seedlings, wild-type FT–SII protein associated more effectively with the FD binding site in the *SEP3* promoter compared with FT(Mu1)–SII and FT(Mu2)–SII (Fig. 3n). These experiments support the requirement for DNA-bound FD−14-3-3 complex to recruit FT (Fig. 3f,j).

We next evaluated the in vivo relevance of these mutations in FT. The delayed flowering of *ft-10* mutants was fully rescued by *FT$^{WT}$-ALFA* transgenes, but only partially rescued by *FT$^{Mu2}$-ALFA* and not at all rescued by *FT$^{Mu1}$-ALFA* (Fig. 3o and Supplementary Table 1). Moreover, when expressed in the SAM from the *FD* promoter, *FT$^{Mu1}$* did not complement, whereas *FT$^{Mu2}$* largely complemented *ft-10*, although they were expressed at similar levels in T3 homozygous lines, supporting the idea that FT(Mu2) retains more biochemical function than FT(Mu1) (Supplementary Table 1). Therefore, the interaction of the FT tail with the DNA-bound FD−14-3-3 complex has a pivotal role in the assembly of the FAC and floral induction.

## 14-3-3 prevents FD condensation for DNA binding

Next, we explored whether 14-3-3 proteins affect FD DNA-binding activity. Phosphorylation of the unstructured C-terminal motif (LX(R/K)SX(pS/pT)XP) of FD is considered essential for interaction with 14-3-3 proteins, the activity of FD[5,6,32,35] and formation of the FAC[4]. However, to our knowledge, phosphorylation of residues in this motif (C4, SAP; Extended Data Fig. 5a) of FD has not been demonstrated in vivo[4,15]. Using mass spectrometry, we detected phosphorylation at threonine (T) 282 in 3HA–mCherry–FD proteins extracted from *gFD::3HA-mCherry-FD; fd-3* seedlings (Extended Data Fig. 6a). The T282A mutation did not affect FD abundance but prevented its interaction with 14-3-3 proteins and its capacity to promote flowering (Extended Data Figs. 6b–j and 7a–c, Supplementary Table 1 and Supplementary information).

The effect of T282 phosphorylation and 14-3-3 binding on FD cellular distribution was examined in *fd-3* mutants expressing *gFD::mVenus– FD* or non-phosphorylatable *gFD::mVenus–FD(T282A)*, whose flowering times were similar to those of Col-0 and *fd-3* plants, respectively (Extended Data Fig. 7a and Supplementary Table 1). Both proteins accumulated in nuclei and in similar spatial patterns at the SAM (Fig. 4a,b and Extended Data Fig. 7d,e). mVenus–FD(WT) appeared evenly distributed in nuclei, whereas mVenus–FD(T282A) predominantly formed large puncta (Fig. 4c–g), despite similar mRNA and protein expression levels (Extended Data Fig. 7f,g). These observations suggest that when 14-3-3 binding is impaired, the predicted disordered[36] feature of FD contributes to the formation of phase-separated condensates (Fig. 4h,i), as observed for other disordered transcription factors[37,38]. These findings are consistent with an overrepresentation of intrinsically disordered human proteins among 14-3-3 clients[21]. Also, FT is not involved in suppressing FD condensation and is not required for its interaction with 14-3-3 proteins (Extended Data Fig. 7h–j and Supplementary Table 2).

We next investigated the behaviour of FD alone or in combination with 14-3-3 or FT proteins in vitro. The MBP tag, along with a PreScission protease cleavage sequence and a fluorescent tag (mScarlet1) were fused in tandem with the FD N terminus (Fig. 4j). This configuration enabled MBP–mScarlet1–FD purification from *E. coli* and visualization of potential mScarlet1–FD phase dynamics using confocal microscopy

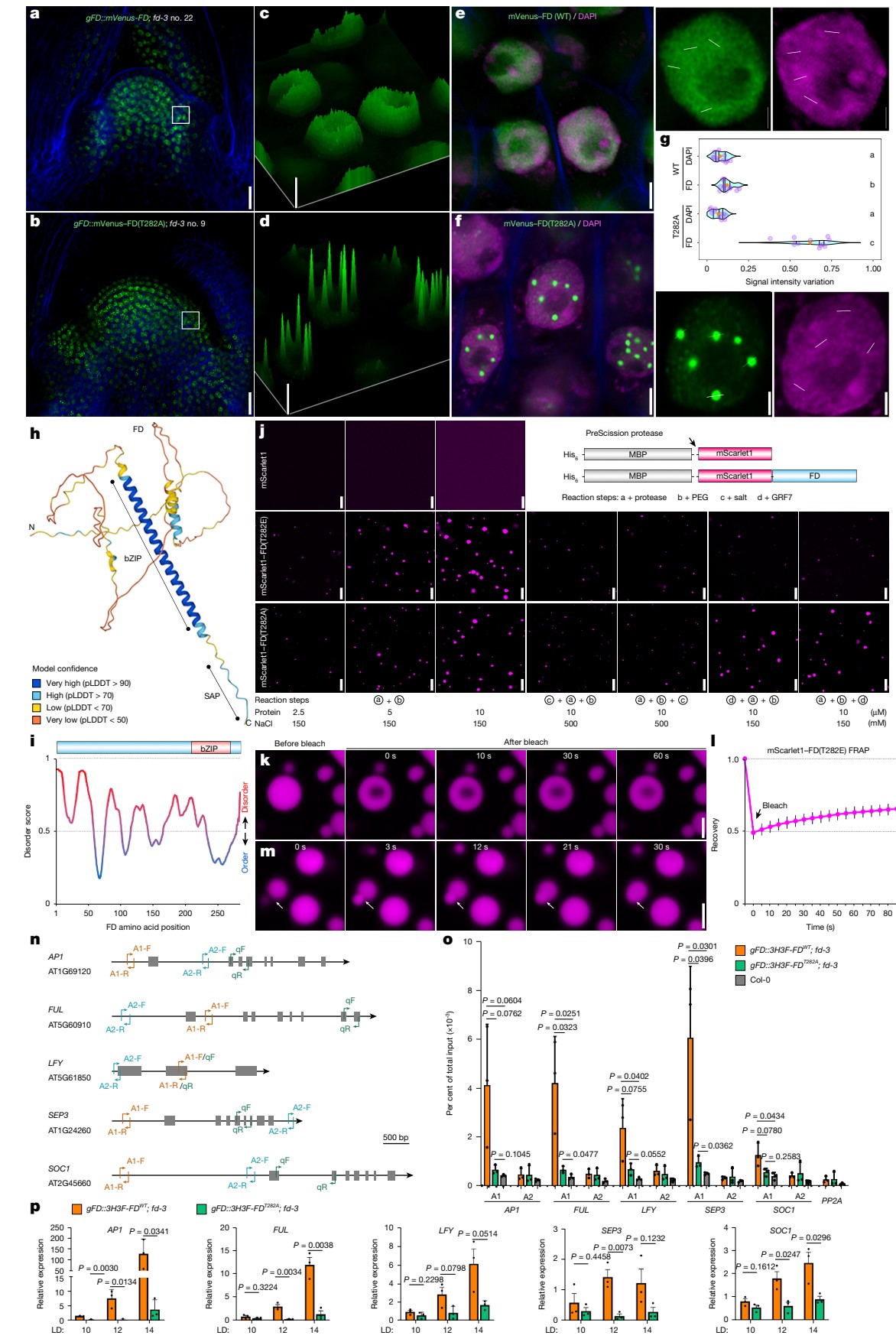

**Fig. 4 |** See next page for caption.

**Fig. 4 | 14-3-3 proteins repress FD condensation and enhance DNA binding.**
**a**,**b**, Confocal images of SAM cells of *gFD::mVenus-FD*; *fd-3* wild-type (no. 22; **a**) or *T282A* (no. 9; **b**) plants. mVenus–FD is shown in green and cell walls (blue) and nuclei (magenta) were stained with Direct Red 23 and DAPI, respectively. Scale bars, 20 μm. **c**,**d**, 2.5D images of cells from **e** (**c**) and **f** (**d**); bars indicate pixel value of signal intensity. **e**,**f**, Left, magnified views of cells in **a** (**e**) and **b** (**f**); scale bars, 2 μm. Right, individual cell nuclei; scale bars, 1 μm. White lines indicate representative regions that were analysed in **g**. **g**, The signal intensity variation (maximum – minimum/total signal intensity) for DAPI, mVenus–FD and mVenus–FD(T282A) pixel point distribution. **h**, AlphaFold-predicted structure of FD. Colours indicate model confidence (pLDDT). **i**, Top, protein domain structure of FD. Bottom, predictions of disordered regions by PrDOS algorithms[36]. **j**, Left, phase diagram of mScarlet1, mScarlet1–FD(T282E) and mScarlet1–FD(T282A) droplets. Right, schematic of protein fusion used for in vitro phase separation assay. Scale bars, 10 μm. **k**, Fluorescence recovery after photobleaching of mScarlet1–FD(T282E) droplets. The photobleaching pulse was applied at *t* = 0. Scale bar, 2.5 μm. **l**, Time course of recovery after photobleaching of mScarlet1–FD(T282E) droplets. Data are mean ± s.d. (*n* = 13). **m**, Fusion of droplets containing mScarlet1–FD(T282E) in in vitro phase separation assay. Scale bar, 2.5 μm. Data in **j**–**m** are representative of three independent experiments. **n**, Amplicons for ChIP–qPCR analysis in **o** and quantitative PCR with reverse transcription (RT–qPCR) in **p**. **o**, ChIP–qPCR showing FD enrichment in target gene regulatory or transcribed regions in *FD::3HA3Flag-FD*; *fd-3* and *FD::3HA3Flag-FD^{T282A}*; *fd-3* plants. **p**, RT–qPCR of *FD* and FD target gene mRNAs in *FD::3HA3Flag-FD*; *fd-3* and *FD::3HA3Flag-FD^{T282A}*; *fd-3* plants. All values are normalized to *ACTIN2*. Data in **o**,**p** are mean ± s.e.m. of three independent biological replicates. Statistical significance was determined by pairwise one-sided *t*-test.

after removal of the MBP tag. After MBP cleavage, solutions containing low concentrations of mScarlet1–FD(T282E) or mScarlet1–FD(T282A), but not mScarlet1 alone, exhibited turbidity and the proteins rapidly formed spherical droplets with dynamic liquid-like properties upon addition of the crowding agent PEG8000 under physiological salt conditions (150 mM NaCl; Fig. 4j). At higher protein concentrations, the droplet size increased, but droplet formation was significantly suppressed at high salt concentration (500 mM). Therefore, FD forms spherical droplets resembling condensates in vitro, and FD carrying T282E or T282A mutations behaves similarly when alone in solution. Droplet fusion was observed upon contact and fluorescence recovery after photobleaching analysis demonstrated rapid diffusion of mScarlet1–FD(T282E) molecules within droplets (Fig. 4k–m), underscoring the dynamic nature of FD condensation. Remarkably, interaction with GRF7 substantially suppressed the formation of or dissolved the large mScarlet1–FD(T282E) liquid-like droplets at 150 mM NaCl, but had no obvious effect on mScarlet1–FD(T282A) droplets (Fig. 4j). FT did not suppress FD droplet formation (Extended Data Fig. 7k). Overall, these data indicate that the physical interaction between FD and 14-3-3 proteins strongly suppresses the capacity of FD to undergo liquid phase condensation in vivo and in vitro. This conclusion was supported by generation of a second mutation in the SAP motif, in which the alanine at position 283 was converted to tryptophan (A283W; Extended Data Fig. 8a). Although this mutant protein retained the phosphorylated threonine-mimicking substitution at position 282 (T282E), it showed similar effects to T282A, including reduced interaction with 14-3-3, liquid phase condensation, and reduced capacity to promote flowering (Extended Data Fig. 8 and Supplementary Table 1).

To understand the influence of phase separation on the DNA binding of FD in vivo, we performed ChIP–qPCR on 3HA-3Flag-FD and 3HA-3Flag-FD(T282A) (Fig. 4n). 3HA-3Flag-FD(T282A) showed weaker binding to several FD target genes[15–17] compared with 3HA-3Flag-FD (Fig. 4o). Consistently, the expression levels of these genes were also significantly lower in *gFD::3HA-3Flag-FD^{T282A}*; *fd-3* compared with *gFD::3HA-3Flag-FD^{WT}*; *fd-3* plants (Fig. 4p), as observed in RNA-seq data of apices of Col-0 and *fd-3*[20] (Extended Data Fig. 7l). To explore the specific binding of FD to DNA in the context of phase separation in vitro, we used recombinant FD(T282E), FD(T282A) and FD(Mu1/T282E) proteins with N-terminal, protease-cleavable MBP tags. After MBP tag removal, FD(T282E) and FD(T282A) quickly formed dynamic spherical droplets (Extended Data Fig. 7m–o). Binding of MBP–FD(T282E) or MBP–FD(T282A) to *SEP3* DNA was detected in EMSA experiments, but protease-mediated removal of the MBP markedly diminished DNA binding of these proteins (Extended Data Fig. 7p). Nevertheless, DNA binding was enhanced when FD(T282E) interacted with GRF7. Notably, GRF7 had no observed effect on DNA binding by FD(T282A) and no DNA binding was observed by FD(Mu1/T282E) under any combination of protease and/or GRF7 addition. Therefore, 14-3-3 proteins have important roles in facilitating DNA binding by FD, both

by reducing its condensation and influencing the properties of its C terminus.

## 14-3-3 promotes FD dimerization

To investigate the biochemical properties of the FD C terminus and its interaction with 14-3-3 proteins, we purified an N-terminal MBP-fused, truncated FD (amino acids 215–285) that includes the bZIP, 14-3-3-binding regions, and the T282E mutation (hereafter MBP–FDc(T282E)). Consistent with the disordered feature of the FD C terminus (Fig. 4), size-exclusion chromatography with multi-angle light scattering (SEC–MALS) detected homodimers of MBP–FDc(T282E) at low concentration and homotetramers or homooctomers at higher concentration (Extended Data Fig. 9a). *SEP3* DNA was bound by MBP–FDc(T282E) dimers at low protein concentrations, but supershifts were detected with MBP–FDc(T282E) at higher concentrations (Extended Data Fig. 9b), reinforcing the idea that MBP–FDc(T282E) oligomers form at increased protein concentrations. Analysis of binding to mutant *SEP3* DNA showed that the dimer form of FDc(T282E) binds specifically to the G-box, but oligomers show reduced binding specificity. Moreover, 14-3-3 protein dimers strongly disassociated homooligomerization of MBP–FDc(T282E), converting it to dimers (Extended Data Fig. 9c). Accordingly, in the EMSA experiments, the patterns of supershifts were more uniform with increasing concentrations of MBP–FDc(T282E) when combined with GRF7 (Extended Data Fig. 9d).

We used AlphaFold2[39] protein structure modelling to explore the mechanisms underlying the formation of FDc dimers. The predicted dimerization interface of FDc(T282E) is highly conserved in the group A bZIPs; we designed mutations to disrupt the predicted dimerization interface (Mu2–Mu6; Extended Data Fig. 9e,f and Extended Data Fig. 5a). GRF7 dimers still bound these mutants, but they only recruited a single copy of the FDc(T282E) containing strong dimerization defective mutations that affect leucine residues in the zipper region (Mu3, Mu4 and Mu5), but could overcome the effect of weaker dimerization mutations (Mu2 (Y238A, E243A) and Mu6 (L263A); Extended Data Fig. 9g,h). These results suggest that 14-3-3 proteins enhance FDc(T282E) dimerization and overcome the defects caused by the weaker mutations. EMSA experiments confirmed these results (Extended Data Fig. 9i). Tests of the functionality of these monomer mutations in transgenic plants showed that late flowering of *fd-3* mutants was fully rescued by the *gFD::3HA-3Flag-FD^{Mu2}* transgene (Extended Data Fig. 9j and Supplementary Table 1), underlining its dimerization ability when 14-3-3 proteins and DNA are present. By contrast, FD(Mu5) and FD(Mu3) significantly impaired FD function. Moreover, these mutant mVenus–FD proteins appeared evenly distributed (Extended Data Fig. 9k and Supplementary Fig. 3), suggesting that these monomers interact with 14-3-3 proteins in vivo, but their activity is reduced. Moreover, chromatin immunoprecipitation (ChIP) experiments showed that DNA binding of Mu3 is significantly reduced compared with wild-type FD (Extended Data Fig. 9f–o). GRF7 not only facilitated the dimerization of FDc(T282E)

but also enhanced its DNA binding (Extended Data Figs. 7p and 9d,i). Our protein structure models suggest that binding of GRF7 to FD might restrict the flexibility of its C terminus by inducing conformational changes in the 13-amino-acid region (C13) between the bZIP and 14-3-3 binding site (Extended Data Figs. 5a and 10a,b). Mutagenesis of this region and functional analysis of the mutant proteins (Mu7 and Mu8) in vivo and in vitro (Extended Data Figs. 10c–e and 11 and Supplementary Table 1), supported the idea that 14-3-3 proteins enhance DNA binding of FD by limiting the flexibility of the C13 region. Thus, 14-3-3 proteins enhance DNA binding by FD in multiple ways, including preventing condensation, promoting dimerization and regulating the flexibility of the C13 region leading to FT recruitment and target gene transcription.

## Discussion

Here we find that FT accumulates in specific cells in floral primordia as well as in the rib zone, organizing centre and, to a lesser extent, the central zone of the SAM, and that the FD and 14-3-3 proteins co-localize with FT in these regions. Previous experiments also reported interaction between FD and FT in the SAM[12] using a strong heat shock promoter to express *FT*, whereas we used the endogenous regulatory sequences. We cannot exclude that in our experiments FT–mVenus is present in more nuclei below the level of detection for confocal microscopy, so we defined the minimum overlap of FT, FD and 14-3-3 proteins in the SAM. In the floral primordium, *FT* mRNA is first present on the adaxial side of the primordium, and later, as the primordium grows, it is present below the domain of expression of *AP1*, which represents the boundary with the suppressed bract[30,31]. This transcriptional pattern of *FT* may contribute to its genetically defined role in conferring floral identity, leading to fewer cauline leaves and branches[40,41]. Moreover, although FT interacts with FD to activate *AP1* transcription[6], our data suggest that this might occur only during the initiation of *AP1* transcription and not to maintain it throughout the primordium. BLADE ON PETIOLE (BOP) proteins also define the boundary between the floral primordium and the suppressed bract and activate *AP1* transcription[31,42–44], but their precise relationship with FT will require further detailed study. The activity of FT and FD later during floral development in the floral meristem to activate *SEP* genes[41] may involve a further round of *FT* transcription. Competition between TFL1 and FT was proposed to explain their antagonism during flowering and inflorescence development[17,45], but our analyses suggest that different spatial and temporal patterns of accumulation are a major contributor to their distinct functions. Nevertheless, there is a transient overlap between *FT* and *TFL1* mRNAs in the first floral primordium (Fig. 2h,i). We find that in *Arabidopsis*, FT activity occurs first at the base of the SAM and then in the primordium, and in rice a related two-step process occurs, whereby movement of florigen leads to transcriptional activation of *FT-LIKE1* at the SAM[46]. This amplification of florigen function, whether carried out by different genes in rice or by the same gene in *Arabidopsis*, may be highly conserved in flowering plants.

We found that binding of the FD–14-3-3 complex to DNA is a determinant of FT recruitment and that the unstructured FT C terminus interacts with DNA when 14-3-3–FD is present. The activity of this evolutionarily conserved interface in recruiting FT to the FAC seems stronger and more important in floral promotion than the interaction with 14-3-3 via interface 2. Nevertheless, both interfaces contribute to biological function. Our results also suggest that the recruitment of FT could stabilize the interaction of the FD–14-3-3 complex with chromatin.

In plants, biomolecular condensates integrate environmental cues with developmental programmes[47–49]. The transcriptional regulators EARLY FLOWERING 3 and AUXIN RESPONSE FACTOR 7 (ARF7) and ARF19 are negatively regulated by condensate formation[50,51]. In common with many other transcription factors, FD is predicted to possess intrinsically disordered regions, and phase-separated condensates of FD mutants that did not interact with 14-3-3 were formed in vivo,

suggesting that phase separation impairs FAC function. In wild-type plants, the T282 phosphorylated form of FD predominates but a pool of FD with non-phosphorylated T282 might occur under particular developmental or environmental conditions. Such condensates could serve as a reservoir of protein that is capable of switching to active FD upon phosphorylation. Intrinsically disordered proteins have been found to be enriched among 14-3-3 clients in humans, and 14-3-3 binding suppressed the condensation of disordered clients[21]. Accordingly, binding of 14-3-3 reduced excessive condensation of FD, and allowed DNA binding and transcription of target genes. Of note, interaction of FD with 14-3-3 proteins also stabilizes FD dimers, potentially inducing conformational changes in the tail at the C terminus of FD that enhance DNA binding. Therefore, 14-3-3 proteins may function as chaperone-like proteins that enhance FD activity at multiple levels.

In higher plants, group A bZIPs implement developmental transitions at the shoot apex and ABA responses[52–54], and our analysis of the role of 14-3-3 proteins in regulating FD is likely to be relevant for other members of this important group of transcription factors (Extended Data Fig. 12, Supplementary Text and Supplementary Table 3). Moreover, PEBP encoding-genes, which include *FT*, are present in basal green plants, whereas FT-like and TFL1-like PEBPs emerged in gymnosperms and are present throughout the seed plant lineage (gymnosperms and angiosperms)[55,56]. Our demonstration of the broad importance of 14-3-3 function for the activity of FD, the two important interfaces that link FT to DNA and the protein complex, and the dynamic pattern of FT accumulation in the SAM and primordia provide novel perspectives for understanding the mechanisms through which florigen promotes flowering.

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

## Methods

### Plant materials and growth conditions

The *Arabidopsis thaliana* Columbia (Col-0) ecotype was used as the main experimental organism. Seeds of Col-0 (N70000), *fd-3* (SALK_054421), *ft-10* (GK_290E08) and other transgenic plants were surface-sterilized with 70% ethanol for 10 min, rinsed with 99% ethanol for 5 min, air-dried and stratified at 4 °C for 3 days before sowing. Plants were grown on soil under long-day conditions (16 h light:8 h dark cycles) or were grown vertically on plates containing 1% agar supplemented with half-strength Murashige and Skoog (MS) medium (pH 5.7) at 22 °C with a light intensity of 160–180 µmol m$^{-2}$ s$^{-1}$ provided by LED bulbs (Philips F17T8/TL841 17 W).

### Plasmid construction

To generate different epitope-tagged fusions of FD, the genomic fragment carrying *FD* promoter (2,930 bp), the full-length coding region and the 3′ untranslated region (1,982 bp) were amplified from Col-0 genomic DNA. DNA encoding 3HA-3Flag and 2HA-mVenus[57] tags were synthesized and amplified with PrimeSTAR GXL DNA Polymerase (Takara Bio) for subcloning. Overlapping PCR was then performed to obtain genomic fusions with *FD* and the epitope tags. Corresponding PCR fragments were then cloned into a modified binary vector PER8 using a HiFi DNA Assembly Kit (NEB). The constructs were transformed into *fd-3* mutants using the floral dip method. The same strategy (with genomic fusions) was used to construct *gGRFs::2HAmScarlet-I*[58]-*GRFs* in Col-0 (*GRF2* and *GRF6*), *grf7* (SALK_084141) or *grf8* (SALK_148929) mutants, and three genomic constructs carrying the 9,149-bp promoter, the FT coding region fused with different tags (*gFT::FT-mVenus*, *gFT::FT-Venus-Halo-Venus* and *gFT::FT-ALFA*[59]), and a 3,159-bp downstream sequence were transformed separately into *ft-10* (GK-290E08) plants. *SUC2::pp2A.1-mCherry* was transformed in to Col-0 background. Primers used to amplify these sequences are listed in Supplementary Table 4.

### Generation of *Arabidopsis* mutants and transgenic plants

Corresponding Col-0, *fd-3* or *ft-10* mutant plants were grown in the greenhouse under LD conditions and were transformed by the floral dip method using *Agrobacterium tumefaciens* strain GV3101. The resulting transgenic T1 seeds were screened on half-strength MS medium with supplemented hygromycin for 7 LD and then they were transferred to soil for the measurement of flowering time.

### RNA extraction and RT–qPCR analysis

Total RNA was extracted from 13-day-old seedlings grown in long-day conditions using the RNeasy plant Mini Kit (QIAGEN) with an on-column DNase (QIAGEN) treatment. cDNA was synthesized from 1 µg RNA using a QuantiTect Reverse Transcription Kit (QIAGEN). Real-time PCR was performed with iQ SYBR Green Supermix (Bio-Rad) in a CFX384 Touch Real-Time PCR Detection System (Bio-Rad). The reference gene *ACTIN2* was used for normalization. Three technical replicates for each of three independent biological replicates were performed for each experiment and representative results are presented. The primers used for qRT–PCR are listed in Supplementary Table 4.

### Immunoblot assays

For western blots, approximately 30 mg of tissue from 13-day-old seedlings grown in long-day conditions was ground into fine powder with liquid nitrogen with a TissueLyser system (QIAGEN). Total protein was extracted using denaturing buffer (100 mM Tris-HCl pH 7.5, 100 mM NaCl, 30 mM EDTA pH 8.0, 4% (w/v) SDS, 20% (v/v) glycerol, 20 mM β-mercaptoethanol (Sigma-Aldrich), 20 mM DTT, 2 mM PMSF (Sigma-Aldrich), 1× Protease Inhibitor Cocktail (PIC, Sigma-Aldrich, P9599), 1× Phosphatase Inhibitor Cocktail 2 (PIC2, Sigma-Aldrich, P5726), 1× Phosphatase Inhibitor Cocktail 3 (PIC3, Sigma-Aldrich, P0044), 80 µM MG132 (Sigma-Aldrich), and 0.01% bromophenol blue) in a 1:5 (w/v) ratio

and was boiled at 95 °C for 10 min. Protein samples were centrifuged at 16,000*g* for 5 min at room temperature and the supernatants were transferred to a new low-protein-binding tube and separated by SDS–PAGE.

For immunoblotting, separated proteins from the gels were transferred onto a PVDF membrane by the Trans-Blot Turbo Transfer System (Bio-Rad). Blots were probed with anti-H–horseradish peroxidase (HRP) (Roche, 12013819001, 1,000-fold dilution), anti-ALFA–HRP (NanoTag, N1505, 1:2,000-fold dilution) or anti-actin–HRP (Santa Cruz, sc-47778, 1:5,000-fold dilution) antibodies conjugated to HRP were used at 1:2,000-fold dilution, in TBS-T buffer. The blots were developed with a 1:1 mix of SuperSignal West Femto Maximum Sensitivity and SuperSignal West Dura Extended Duration Substrates and signals were detected on a ChemiDoc MP Imaging System (Bio-Rad). Uncropped blots are shown in Supplementary Fig. 4.

### Co-immunoprecipitation assay

The in vivo co-immunoprecipitation assays were performed as previously described, with minor modifications[60]. In brief, 3 g of material from 12-day-old seedlings grown in long-day conditions was collected at Zeitgeber time (ZT) 7 and cross-linked in 1× phosphate-buffered saline (PBS) with 1 mM disuccinimidyl glutarate with vacuum filtration for 15 min. The tissues were washed and frozen in liquid nitrogen before storing at −80 °C. The tissues were ground to a fine powder in liquid nitrogen, and semi-pure nuclei were extracted in nuclei isolation buffer (10 mM Tris-HCl pH 8.0, 400 mM sucrose, 0.05% Triton X-100, 1 mM PMSF, 5 mM β-mercaptoethanol and 0.25× protease inhibitor cocktail (PIC)). The isolated nuclei were washed 3 times in wash buffer (10 mM Tris-HCl pH 8.0, 250 mM sucrose, 0.5% Triton X-100, 10 mM MgCl$_2$, 1 mM PMSF, 5 mM β-mercaptoethanol and 0.25× PIC). Nuclear proteins were released by brief sonication in the buffer (Tris-HCl pH 7.5, 3 mM EDTA, 0.5% Triton X-100, 150 mM NaCl, 1 mM PMSF, 50 µM MG132, 1 mM DTT, 1× PIC, 1× PIC2 and 1× PIC3). Western blotting was performed with the extracted nuclear proteins using anti-HA (12013819001, Roche) and anti-H3 (Abcam, ab1791) antibodies before immunoprecipitation.

For co-immunoprecipitation, 30 µl anti-HA magnetic beads (Thermo Fisher) was added to the diluted nuclear protein solution (0.5% Triton X-100, 1 mM EDTA, 20 mM Tris-HCl pH 7.5, and 100 mM NaCl and 1× PIC (Sigma-Aldrich)) and rotated for 40 min at 4 °C. The beads were washed five times with immunoprecipitation buffer. Aliquots (3 µl) of beads were boiled with 2× Laemmli buffer (Bio-Rad) for immunoblotting analysis using anti-HA (Roche, 12013819001, 1:1,000-fold dilution), anti-Strep (IBA, 2-1509-001, 1:4,000-fold dilution) and anti-H3 (Abcam, ab1791, 1:4,000 dilution). The remaining beads with the IPed proteins were stored at −80 °C before on-bead digestion for liquid chromatography–mass spectrometry.

### Mass spectrometry and data analysis

The anti-HA magnetic beads with IPed proteins were digested on beads using trypsin or LysC to identify interacting proteins of FD, GRF7 or GRF8, or phosphorylation peptides of FD. In brief, the beads were buffer exchanged and re-dissolved in 25 µl digestion buffer I (50 mM Tris pH 7.5, 2 M urea, 1 mM DTT, 5 ng µl$^{-1}$ trypsin) and incubated for 30 min at 30 °C in a Thermomixer at 400 rpm. Next, beads were pelleted, and the supernatant was transferred to a new tube. Digestion buffer II (50 mM Tris pH 7.5, 2 M urea, 5 mM chloroacetamide) was added to the beads and after mixing the beads were pelleted, the supernatant was collected and combined with the previous one. The combined supernatants were then incubated overnight at 32 °C in a Thermomixer at 400 rpm; samples were protected from light during incubation. The digestion was quenched by adding 1 µl trifluoroacetic acid (TFA) and desalted with C18 Empore disk membranes according to the StageTip protocol[61].

Dried peptides were re-dissolved in 2% acetonitrile (ACN), 0.1% TFA (10 µl) for analysis. Samples were analysed using an EASY-nLC 1200 (Thermo Fisher) coupled to a Q Exactive Plus mass spectrometer (Thermo Fisher). Peptides were separated on 16-cm frit-less silica

emitters (New Objective, 75-µm inner diameter), packed in-house with reversed-phase ReproSil-Pur C18 AQ 1.9 µm resin (Dr. Maisch). Peptides were loaded onto the column and eluted for 115 min using a segmented linear gradient of 5% to 95% solvent B (0 min: 5% B; 0–5 min 5% B; 5–65 min →20% B; 65–90 min →35% B; 90–100 min →55% B; 100–105 min →95% B; 105–115 min 95% B) (solvent A: 0% ACN, 0.1% formic acid; solvent B: 80% ACN, 0.1% formic acid, solvents A and B together constituting 100% of the mobile phase) at a flow rate of 300 nl min$^{-1}$. Mass spectra were acquired in data-dependent acquisition mode with a TOP15 method. MS spectra were acquired in the Orbitrap analyser with a mass range of 300–1,750 $m/z$ at a resolution of 70,000 full width at half maximum (FWHM) and a target value of $3 \times 10^6$ ions. Precursors were selected with an isolation window of 1.3 $m/z$ (Q Exactive Plus). HCD fragmentation was performed at a normalized collision energy of 25. MS/MS spectra were acquired with a target value of $10^5$ ions at a resolution of 17,500 FWHM, a maximum injection time of 55 ms and a fixed first mass of $m/z$ 100. Peptides with a charge of +1, greater than 6, or with unassigned charge state were excluded from fragmentation for MS2, and dynamic exclusion for 30 s prevented repeated selection of precursors.

Alternatively, samples were analysed using an Ultimate 3000 RSLC nano (Thermo Fisher) coupled to an Orbitrap Exploris 480 mass spectrometer equipped with a FAIMS Pro interface for Field asymmetric ion mobility separation (Thermo Fisher). Peptides were pre-concentrated on an Acclaim PepMap 100 pre-column (75 µM × 2 cm, C18, 3 µM or 5 µM, 100 Å, Thermo Fisher) using the loading pump and buffer A (water, 0.1% TFA) with a flow of 7 µl min$^{-1}$ (3 µM), or 15 µl min$^{-1}$ (5 µM) for 5 min. Peptides were separated on 16-cm frit-less silica emitters (New Objective, 75 µm inner diameter), packed in-house with reversed-phase ReproSil-Pur C18 AQ 1.9 µm resin (Dr. Maisch). Peptides were loaded onto the column and eluted for 130 min using a segmented linear gradient of 5% to 95% solvent B (0 min: 5% B; 0–5 min 5% B; 5–65 min →20% B; 65–90 min →35% B; 90–100 min →55% B; 100–105 min →95% B; 105–115 min 95% B; 115–115.1 min →5% B, 115.1–130 min 5% B) at a flow rate of 300 nl min$^{-1}$. Mass spectra were acquired in data-dependent acquisition mode with the TOP_S method using a cycle time of 2 s. For field asymmetric ion mobility separation (FAIMS), two compensation voltages (−45 and −60) were applied and the cycle time was set to 1 s for each experiment. MS spectra were acquired in the Orbitrap analyser with a mass range of 320–1,200 $m/z$ at a resolution of 60,000 FWHM and a normalized AGC target of 300%. Precursors were filtered using the MIPS option (MIPS mode = peptide), the intensity threshold was set to 5,000, Precursors were selected with an isolation window of 1.6 $m/z$. HCD fragmentation was performed at a normalized collision energy of 30%. MS/MS spectra were acquired with a target value of 75% ions at a resolution of 15,000 FWHM, inject time set to auto, and a fixed first mass of $m/z$ 120. Peptides with a charge of +1, greater than 6, or with unassigned charge state were excluded from fragmentation for MS2.

Raw data were processed using MaxQuant software[62] (v.1.6.3.4, http://www.maxquant.org/) with label-free quantification (LFQ) and iBAQ enabled[63]. MS/MS spectra were searched using the Andromeda search engine against a combined database containing the sequences from *A. thaliana* (TAIR10_pep_20101214; ftp://ftp.arabidopsis.org/home/tair/Proteins/TAIR10_protein_lists/) and sequences of 248 common contaminant proteins and decoy sequences. Trypsin or LysC specificity was required and a maximum of two missed cleavages allowed. Minimal peptide length was set to seven amino acids. Carbamidomethylation of cysteine residues was set as fixed, and oxidation of methionine and protein N-terminal acetylation were set as variable modifications. Peptide-spectrum matches and proteins were retained if they were below a false discovery rate (FDR) of 1%.

For interacting protein analyses, statistical analysis of the MaxLFQ values was carried out using Perseus (v.1.5.8.5, http://www.maxquant.org/). Quantified proteins were filtered for reverse hits and hits 'identified by site' and MaxLFQ values were log$_2$-transformed. After grouping samples by condition, only those proteins were retained for the

subsequent analysis that had two valid values in one of the conditions. Two-sample $t$-tests were performed using a permutation-based FDR of 5%. Alternatively, quantified proteins were grouped by condition and only those hits were retained that had three valid values in one of the conditions. Missing values were imputed from a normal distribution (1.8 downshift, separately for each column). Volcano plots were generated in Perseus using an FDR of 5% and an $S_0 = 1$. The Perseus output was exported and further processed using Excel.

To identify phosphorylation of T282, the Phospho (STY)Sites.txt file was manually inspected for the presence and localization of the site. The presence of the site was confirmed by searching individual raw files using ProteomeDiscoverer 2.2 (Thermo Fisher).

### Confocal imaging of SAM cells
SAMs of seedlings grown in long-day conditions were dissected and fixed with 4% (w/v) paraformaldehyde. The fixed samples were washed twice with PBS for 5 min and cleared with ClearSee solution[64] for 2 days in the dark at room temperature. After clearing, samples were washed twice with PBS buffer for 5 min and embedded with 6.5% (w/v) low-melt agarose (Bio-Rad). The embedded samples were sectioned into 70-µm slices using a vibrating blade microtome (Leica VT1000 S) and then stained with dyes. For the FD and GRF co-localization analysis, the cell wall was stained with Renaissance 2200 (0.1% (v/v) in PBS)[65] for 30 min and washed in PBS buffer for 5 min. For the FD-chromatin co-localization analysis, the cell wall was stained with Direct Red 23 (0.5% (w/v) in ClearSee)[66] for 1 h and washed with PBS buffer for 10 min. The nuclear chromatin was then stained with 1 µg ml$^{-1}$ DAPI (Thermo Fisher) for 30 min and washed in PBS buffer for 10 min. The stained samples were mounted onto slides with ProLong Antifade Mountants (Thermo Fisher) for signal preservation. Image collection was performed using a Zeiss LSM 880 confocal microscope. The Renaissance and DAPI signals were detected at 410–503 nm with an excitation wavelength of 405 nm. The mVenus signal was excited with a 514 nm laser and collected at 520–560 nm and mScarlet-I and Direct Red 23 were excited with a 561 nm laser and detected at 566–620 nm. The imaging data were processed using Zen 3.10 (Zeiss) software.

### Single-cell nuclear quantification of FD–FT fluorescence signal
Individual confocal images of *gFT*::FT-mVenus and *gFD*::mScarlet1–FD were processed using Cellpose (2.2.3)[67,68] and Matlab (MathWorks (2022); MATLAB v.9.13.0 (R2022b)). Nuclear segmentation was performed using cyto Cellpose model on the FD-channel.tif file. The cell diameter parameter was automatically calibrated. The output nuclear segmentation.png files were processed using custom-made MATLAB code and adapting the previous method[68] to 2D images. For each confocal image, a curved line was drawn following the parabolic outline of the SAM. A parabolic fit was then performed, accounting for a possible tilt of the SAM. Based on the fitted parabola, a 2D parabolic mask was created. A rectangular mask was also created, extending from the two ends of the parabola up to the inferior edge of the image. These two masks were combined and all intensity values of the pixels outside the new mask were set to 0. Using the previously generated parabolic mask and published WUS/CLV3 data[68], the meristematic tissue was divided into four different regions: central zone (CZ), organizing centre (OC), peripheral zone (PZ) and rib meristem (RM). The first two regions were defined using the height and width of CLV3 and WUS domains, respectively, as proxies. All meristematic tissue below the OC and within the previously generated rectangular mask was considered to be RM. PZ included all nuclei located in the region contained between the OC/CZ and the fitted parabola. Such parametrization of the SAM allows for its compartmentalization it into four different regions and for the assignment of each nucleus to its associated regions if its centroid coordinates are contained that region. Brunner–Munzel test was used to measure statistic significant differences in the median distributions of FD and FT nuclear concentrations between time points. Further details are at https://gitlab.com/GRM_14/gao_ding_et_al_2025/-/tree/011d3d70fc1c0f41670c6f0b860b3c586e8949fd/.

## RNA in situ hybridization

RNA in situ hybridization was performed as described previously[69]. The template for the *FT* probe was transcribed from cDNA using a specific primer pair (Supplementary Table 4) with T3 and T7 polymerase binding sites attached to the forward and reverse primers, respectively.

## RNAscope fluorescent multiplex assays

The RNAscope assay was conducted following the RNAscope Multiplex Fluorescent Assay v.2 protocol provided by ACDBio (materials available at https://acdbio.com). In brief, formalin-fixed, paraffin-embedded tissue samples were used for analyses. Specific probes for *FD*, *FT* and *TFL1* (assigned to channels C1, C2 and C3, respectively) with the following catalogue numbers: 1307011-C1 (*FD*), 1307021-C2 (*FT*), and 1307031-C3 (*TFL1*) were used. To visualize *FD*, *FT* and *AP1* in the same sample, the *AP1* probe was assigned to channel C3 (1569941-C3). The RNAscope 3-plex Negative Control Probe (320871) was used as a negative control. All probes were hybridized overnight at 40 °C.

To visualize targets, TSA Plus fluorophores (diluted with TSA buffer from ACDBio) were applied as follows: TSA Vivid 520 (323271, diluted 1:2,500) for C3, TSA Vivid 570 (323272, diluted 1:1,500) for C1, and TSA Vivid 650 (323273, diluted 1:1,500) for C2. Additionally, Renaissance (0.1% v/v in PBS) was used to stain the cell walls.

Confocal images were captured using a Zeiss LSM 880 confocal microscope. The imaging data were processed and analysed using Zen 3.10 (Zeiss), Fiji (v.2.16.0), Cellpose (v.2.2.3) and Matlab (v.9.13.0 R2022b). The Renaissance signal was detected at 410–503 nm with an excitation wavelength of 405 nm. The filter settings for FITC, Cy3 and Cy5 were used for the TSA Vivid Fluorophore 520, 570 and 650, separately.

## Quantification of RNAscope images

Quantification of RNAscope images at the tissue level was performed using a similar pipeline as the custom-made MATLAB described before, except that first, a sum projection of the images belonging to the same meristem was computed. On this image, the same pipeline as the one previously described was applied to obtain a 2D parabolic mask outlining the SAM. Then, a second 2D parabolic mask with increased curvature was created based on this and all intensity values of the pixels outside the mask were set to 0. This mask was then divided into consecutive 10-μm sections. This allowed fluorescence intensity concentration profiles to be obtained along the SAM longitudinal axis. Concentration was defined as the ratio of total intensity (sum of pixel intensity) to the total area (sum of the pixel area).

## Protein expression and purification for in vitro analysis

Codons of the coding sequences of FD and GRF7 from *A. thaliana* were optimized to *E. coli* and cloned into pMAL-c5X-His (NEB) or a modified pMAL-c5X vector. Transformants carrying the recombinant plasmids were grown in LD medium supplemented with appropriate antibiotic to $OD_{600} = 0.6$ before induction by 0.6 mM IPTG for 16–20 h at 12 °C. The *E. coli* cells were collected by centrifugation, resuspended in wash buffer (25 mM Bis-Tris pH 8.0, 160 mM NaCl and 15 mM imidazole) and sonicated to prepare cell lysates. The proteins were purified using Ni-NTA beads (QIAGEN), the bound proteins were washed 5 times with wash buffer and eluted using elution buffer (25 mM Bis-Tris pH 8.0, 160 mM NaCl and 250 mM imidazole). The eluted proteins were further purified by size-exclusion chromatography (HiLoad 16/600 Superdex 200 pg, GE Healthcare) in buffer containing 25 mM Tris-HCl pH 8.0, 160 mM NaCl and 2% (v/v) glycerol.

## Structural modelling

The structures of GRF7, full-length FD and truncated and mutant FD were predicted using AlphaFold[70] and AlphaFold2[71]. The modelled structure of the FDc–GRF7 and GRF7–FT complex was predicted by ColabFold[39]. The modelled structure of the FDc–DNA complex was based on a bZIP (PAP1)–DNA complex (Protein Data Bank (PDB): 1GD2)[72]. The modelled structure of the FAC–DNA complex was based on the modelled FDc–DNA, FDc–GRF7 and GRF7–FT complexes.

## SEC–MALS

Purified recombinant proteins were quantified by NanoDrop using the protein-specific extinction coefficient and diluted to the desired concentration (as mentioned in the figures). SEC–MALS was performed in buffer (25 mM Tris-HCl pH 8.0, 160 mM NaCl and 2% (v/v) glycerol) on a 10/600 Superdex 200 pg (home packed) column using an AKTÄ pure 25 M chromatography system coupled to a miniDAWN multi-angle light scattering detector (Wyatt Technology) as well as a refractive index detector (Shodex RI-501). Five-hundred microlitres of sample was used per run at a flow rate of 0.5 ml min$^{-1}$. BSA was used as a standard for calibration. Baseline correction, selection of peaks and calculation of molecular masses was performed with the ASTRA 8.2 software package.

## Gel-shift assay (EMSA)

*SEP3* or *LFY* DNA probe (28 bp) covering one G-box binding site was synthesized by annealing single-stranded 5'-Cy5-labeled oligo in annealing buffer (10 mM Tris pH 8.0, 50 mM NaCl, and 1 mM EDTA pH 8.0). Binding reactions with different proteins with different combinations were indicated in the figures and were carried out in buffer containing 10 mM Tris, 50 ng μL Poly (dI-dC), 50 mM KCl, 10 mM KCl, 1 mM DTT, 5% (v/v) glycerol and 0.1% NP-40. Binding reaction tubes were kept on ice for 20 min and were then loaded onto 6% DNA Retardation Gels (Thermo Fisher) and run in 0.5× Tris/Borate/EDTA buffer at room temperature for 60 min at 100 V. Binding signals were visualized using a ChemiDoc MP Imaging System (Bio-Rad). The primers used for DNA probes are listed in Supplementary Table 4.

## Chromatin immunoprecipitation

ChIP methods were described previously with minor modifications[60]. For ChIP–qPCR of 3HA3Flag-FD, 9 g of above-ground tissue from 10-day-old LD-grown seedlings was collected at ZT 7, and cross-linked for 10 min by vacuum filtration in PBS solution containing 1% formaldehyde. For chromatin immunoprecipitation, 50 μl Dynabeads Protein G beads (Thermo Fisher) coated with 20 μl HA antibody (Abcam, ab9110) was incubated for 4 h with 3 ml of the diluted chromatin solution (1% Triton X-100, 1 mM EDTA, 0.08% SDS, 15 mM Tris-HCl, pH 8.0, and 150 mM NaCl). After washing 3 times with wash buffer (1% NP-40, 1 mM EDTA, 0.1% SDS, 0.1% DOC (sodium deoxycholate, Sigma-Aldrich), 20 mM Tris-HCl, pH 8.0, and 150 mM NaCl), the immune complex was eluted from the beads in 400 μl elution buffer (1% SDS and 0.1 M NaHCO$_3$). Next, samples were reverse cross-linked with 5 μl Proteinase K and 20 μl 5 M NaCl at 65 °C overnight and DNA was purified by a MinElute PCR Purification Kit (QIAGEN). Amounts of input and IP DNA were quantified by fluorometry (Promega, Quantus) and the size of the fragments was analysed by ultra-sensitive capillary electrophoresis (Agilent FEMTOpulse), and the resulting DNA was used for ChIP–qPCR. The primers used for ChIP–qPCR are listed in Supplementary Table 4.

## Semi in vivo co-immunoprecipitation and ChIP–qPCR

To perform the semi in vivo co-immunoprecipitation assays for FD and FT, semi-pure nuclei were extracted from 3 g of tissue from 12-day-old *FD::3HA-mCherry-FD*; *fd-3* seedlings grown in long-day conditions as described above. Nuclear proteins were extracted in the sonication buffer (50 mM Tris-HCl (pH 8.0), 2 mM EDTA, 0.5% Triton X-100, 100 mM NaCl, 5 mM MgCl$_2$, 60 mM KCl, 2% (v/v) glycerol, 1 mM PMSF, 50 μM MG132, 1 mM DTT, 1× PIC, 1× PIC2, 1× PIC3). A total of 180 μg *E. coli*-purified wild-type FT–SII (Strep tag II) or mutated FT–SII protein was added to 800 μl FD nuclear protein, and incubated with rotation for interaction at 4 °C for 2 h.

For co-immunoprecipitation, 30 μl anti-Strep agarose beads (Strep-Tactin XT 4Flow resin, IBA Lifesciences, 2-5010) was pre-equilibrated

with the sonication buffer before being added to the protein mixtures. The immunoprecipitation against FT-SII was performed with rotation at 4 °C for 40 min. The beads were washed 4 times in the wash buffer (50 mM Tris-HCl (pH 8.0), 2 mM EDTA, 0.5% Triton X-100, 100 mM NaCl, 5 mM MgCl$_2$, 60 mM KCl, 2% (v/v) glycerol) and then resuspended in 30 µl wash buffer and 10 µl 4× Laemmli loading buffer, followed by boiling for 10 min at 95 °C for immunoblotting analysis using anti-HA–HRP (Roche, 12013819001, 1:1,000 dilution), anti-StrepMAB-Classic–HRP (IBA Lifesciences, 2-1509-001, 1:4,000 dilution) and anti-H3–HRP (Abcam, ab1791, 1:4,000 dilution).

For semi in vivo ChIP–qPCR analysis for purified FT-SII protein, chromatin from 12-day-old *FD::3HA-mCherry-FD*; *fd-3* seedlings grown in long-day conditions was isolated in the above sonication buffer with the addition of 0.24% (w/v) SDS. After interaction of purified FT proteins with the chromatin extractions from in vivo, 0.1 mM disuccinimidyl glutarate followed by 0.1% (v/v) formaldehyde were added to the reaction buffer for 30 min to cross-link FT with interacting proteins and chromatin. ChIP–qPCR was performed as described in 'Chromatin immunoprecipitation'. Primers are listed in Supplementary Table 4.

### Phylogenetic analysis
Nucleotide and protein sequences of bZIP and 14-3-3 gene families were blasted and obtained using 'makeblastdb' module in DIAMOND v.2.16.160[73]. The sequences of both gene families were used for database searching by BLASTP. The initially identified candidate protein sequences in green plants were cut off with an e-value < $10^{-5}$. Then, all candidates of bZIP or 14-3-3 families were verified that contained at least one bZIP domain or that were annotated as 14-3-3 proteins by the Conserved Domains Database[74] (https://www.ncbi. nlm.nih.gov/Structure/cdd/wrpsb.cgi). MAFFT[75] v.7.490 with auto parameters was used for protein sequence alignment. A maximum likelihood algorithm implemented in IQ-TREE v.1.5.5[76] with the Jones–Taylor–Thornton model of evolution under GAMMA rate distribution with bootstrapping criterion (up to a maximum of 1,000 bootstraps) was used for phylogenetic analysis. The obtained trees were visualized using the iTOL[77] (v.6.7.6; http://itol.embl.de/) phylogeny visualization program.

### Ethics and inclusion statement
All data were transparently shared and discussed with all authors during preparation of the manuscript. Authorship criteria were carefully considered, and all contributors were included and their contributions transparently discussed. All materials are available from the authors.

### Materials availability
All mutants, transgenic plants, and all plasmid constructions using *FD*, *14-3-3* and *FT* genes are available from G.C.

### Reporting summary
Further information on research design is available in the Nature Portfolio Reporting Summary linked to this article.

### Data availability
The mass spectrometry proteomics data have been deposited and are accessible at the ProteomeXchange Consortium via the PRIDE[78] partner repository with the dataset identifier PXD067955. The full versions of western blots are available in Supplementary Fig. 4 and the data for graphs are available in Supplementary Table 1. Further information and requests for resources and reagents should be directed to G.C.

### Code availability
Custom code was developed to analyse signal co-localizations at the shoot apex meristem. Source code for analysis signal and figure generation is publicly available at GitHub (https://gitlab.com/GRM_14/gao_ding_et_al_2025/-/tree/011d3d70fc1c0f41670c6f0b860b3c586c8949fd).

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

**Acknowledgements** We thank A. Pajoro and J. Chandler for critical reading of the manuscript; P. Casanova Ferrer for advice on the quantification analyses; B. Li and T. Wang for their contributions to data analysis and the preparation of schematic representations of the SAMs. This work was partly funded by the Deutsche Forschungsgemeinschaft through Cluster of Excellence CEPLAS (EXC 2048/1 project ID: 390686111). N.D. was supported by Chinese Scholarship Council (CSC 202008320398). V.d.S.F. was supported by Alexander von Humboldt-Stiftung (BRA 1210514 HFST-P) and European Union H2020 Marie Skłodowska-Curie Actions (894969). M.C. was supported by Alexander von Humboldt-Stiftung (ITA 1216206 HFST-P). The laboratory of G.C. receives core funding from the Max Planck Society.

**Author contributions** Y.W., D.Y. and S.-Z.Z. contributed equally as co-second authors. Conceptualization: H.G. and G.C. Formal analysis: S.-Z.Z., S.C.S., M.C., V.d.S.F. and P.d.l.R. Fluorescence quantification analyses: G.R.M. In vitro experiments: H.G. Confocal imaging and RNA in situ hybridization: N.D., C.V., H.G., T.T. and U.N. IP–MS: N.D., H.G., S.C.S. and A.H. ChIP and co-immunoprecipitation experiments: N.D. and H.G. Protein structure modelling: H.G., Y.W., D.Y. and E.L. Reconstruction of phylogenetic tree: S.-Z.Z. Funding acquisition: N.D., M.C., V.d.S.F. and G.C. Supervision: H.G., H.N., J.-Y.H., J.C. and G.C. Writing: H.G. and G.C.

**Funding** Open access funding provided by Max Planck Society.

**Competing interests** The authors declare no competing interests.

**Additional information**
**Correspondence and requests for materials** should be addressed to He Gao or George Coupland.

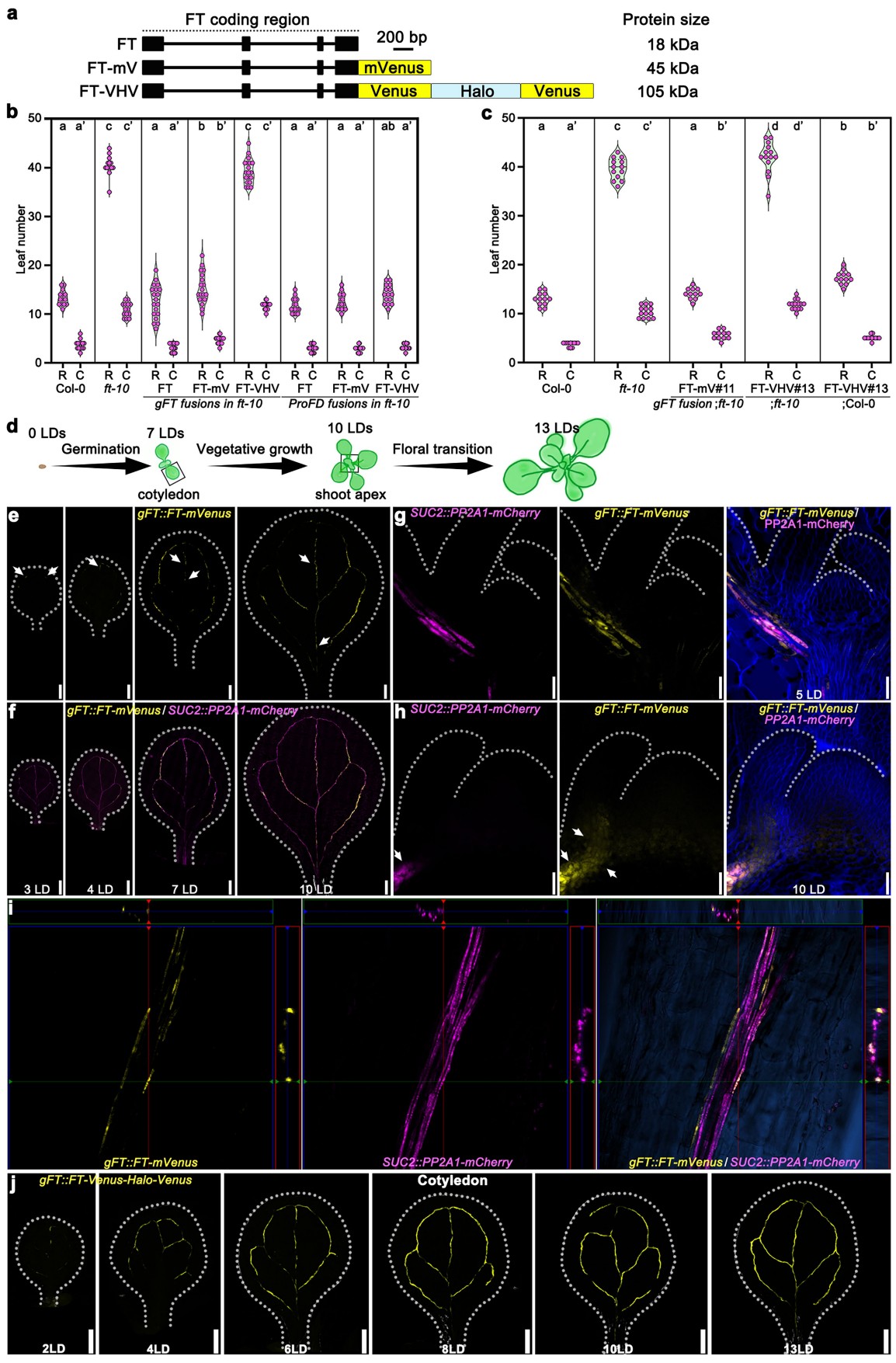

**Extended Data Fig. 1** | See next page for caption.

**Extended Data Fig. 1 | Flowering time of transgenic plants expressing FT and various fluorescent tag fusions driven by *FT* and *FD* promoter and downstream regions. a**, Schematic depicting the constructions of FT fluorescent tag fusions. **b**, Leaf number (R: rosette; C: cauline) of T1 transgenic *ft-10* plants expressing FT, FT-mVenus and FT-Venus-Halo-Venus driven by *FT* and *FD* promoter and downstream genomic regions. Different letters represent significant differences of rosette leaves and cauline leaves among genotypes ($P < 0.05$, using one-way ANOVA followed by Tukey's pairwise multiple comparison, $P = 1.11*10^{-16}$ and $P = 1.11*10^{-16}$, respectively; The sample sizes were as follows: $n = 14$ for Col-0; $n = 14$ for *ft-10*; $n = 19$ for *gFT:FT* in *ft-10*; $n = 19$ for *gFT:FT-mVenus* in *ft-10*; $n = 19$ for *gFT:FT-Venus-Halo-Venus* in *ft-10*; $n = 16$ for *FD:FT* in *ft-10*; $n = 16$ for *FD:FT-mVenus* in *ft-10* and $n = 15$ for *FD:FT-Venus-Halo-Venus* in *ft-10*, respectively, all grown under long-days (LDs). **c**, Leaf number (R: rosette; C: cauline) of T3 homozygous transgenic *ft-10* plants expressing FT-mVenus and FT-Venus-Halo-Venus driven by the *FT* promoter. *gFT::FT-Venus-Halo-Venus;ft-10* #13 was backcrossed to Col-0 to synchronize developmental stages as for *gFT::FT-mVenus; ft-10*. Different letters represent significant differences of rosette leaves and cauline leaves among genotypes ($P < 0.05$, using one-way ANOVA followed by Tukey's pairwise multiple comparison, $P = 1.11*10^{-16}$ and $P = 1.11*10^{-16}$, respectively; The sample sizes were as follows: $n = 12$ for Col-0; $n = 13$ for *ft-10*; $n = 11$ for *gFT:FT-mVenus* in *ft-10*; $n = 14$ for *gFT:FT-Venus-Halo-Venus* in *ft-10* and $n = 12$ *gFT:FT-Venus-Halo-Venus* in Col-0, respectively, all grown under long-days (LDs). **d**, Schematic depicting the timing of floral transition in Arabidopsis. **e,f**, Confocal images of cotyledon cells co-expressing *gFT::FT-mVenus* and *SUC2::PP2A-mCherry* at 3, 4, 7 and 10 LD. **g,h**, Confocal images of cells at the base of cotyledons co-expressing *gFT::FT-mVenus* and *SUC2::PP2A-mCherry* at 5 and 10 LD. **i**, Confocal images of cotyledon cells in the major vein co-expressing *gFT::FT-mVenus* and *pSUC2::PP2A-mCherry*. **j**, Confocal images of FT-Venus-Halo-Venus signal in cotyledons of *gFT::FT-Venus-Halo-Venus*; Col-0 #13 plants under long days. Scale bars = 400 μm. The confocal images in **e**–**j** are representative of three independent cotyledons or meristems.

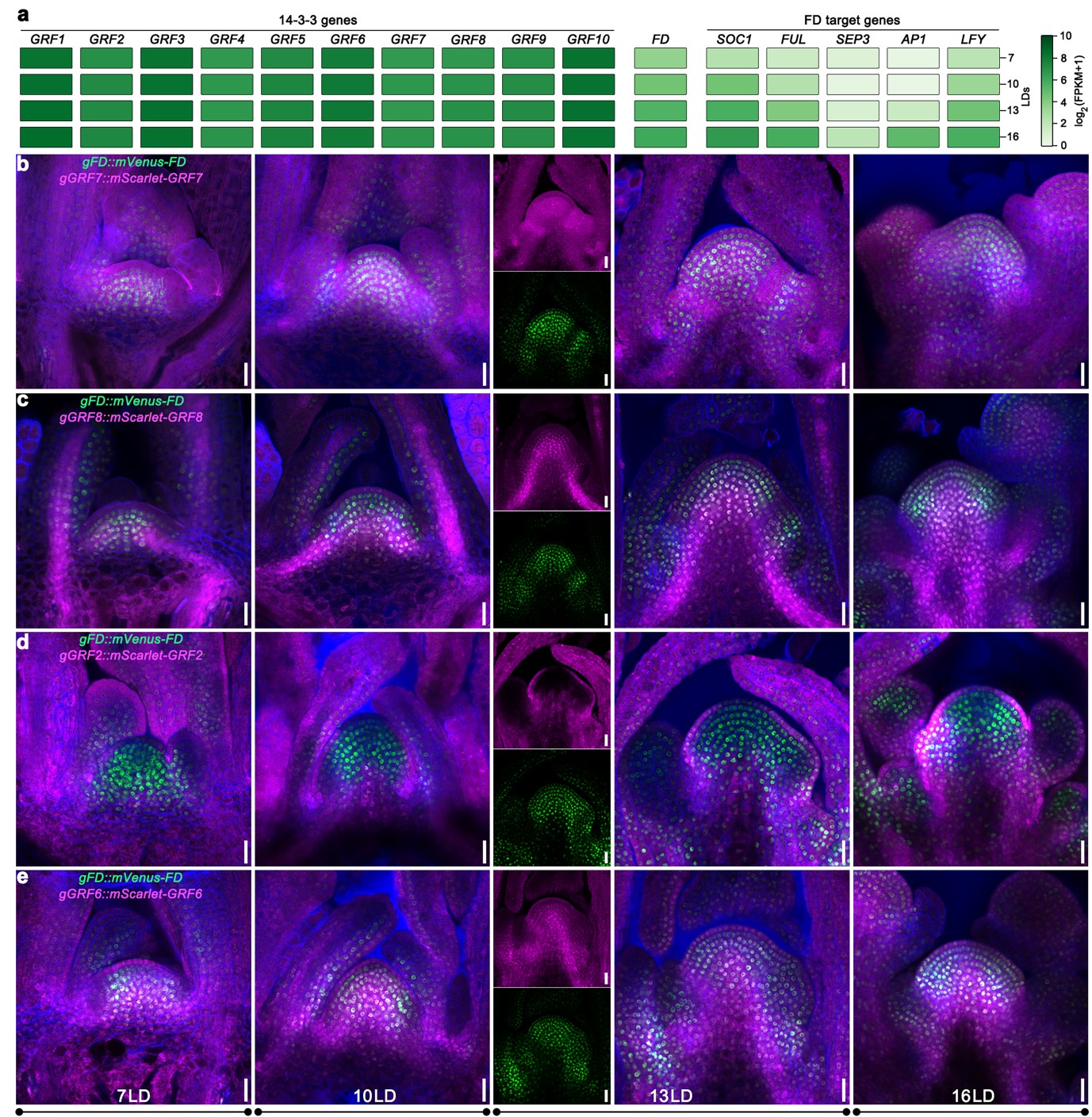

**Extended Data Fig. 2 | Spatiotemporal accumulation of FD and 14-3-3 proteins at the shoot apical meristem of Arabidopsis. a**, Heatmap showing the Z-score normalized expression values of *GRF1* to *GRF10*, *FD* and FD target genes in shoot apical meristems dissected from Col-0 plants grown for 7, 10, 13 and 16 long days (LDs). Data are means of three independent experiments for RNA sequencing[20]. **b,c,d,e**, Confocal images of shoot apical meristem cells co-expressing *gFD::mVenus-FD* and either *gGRF7::mScarlet-I*-GRF7, *gGRF8::mScarlet-I-GRF8*, *gGRF2::mScarlet-I-GRF2* or *gGRF6::mScarlet-I-GRF6* at 7, 10, 13 and 16 LDs. The central panels show the expression pattern of the individual chromophores at 13 LDs. The cell walls (blue) were stained with Renaissance 2200. Scale bars = 20 μm. The confocal images in **b**–**e** are representative of three independent transgenic lines.

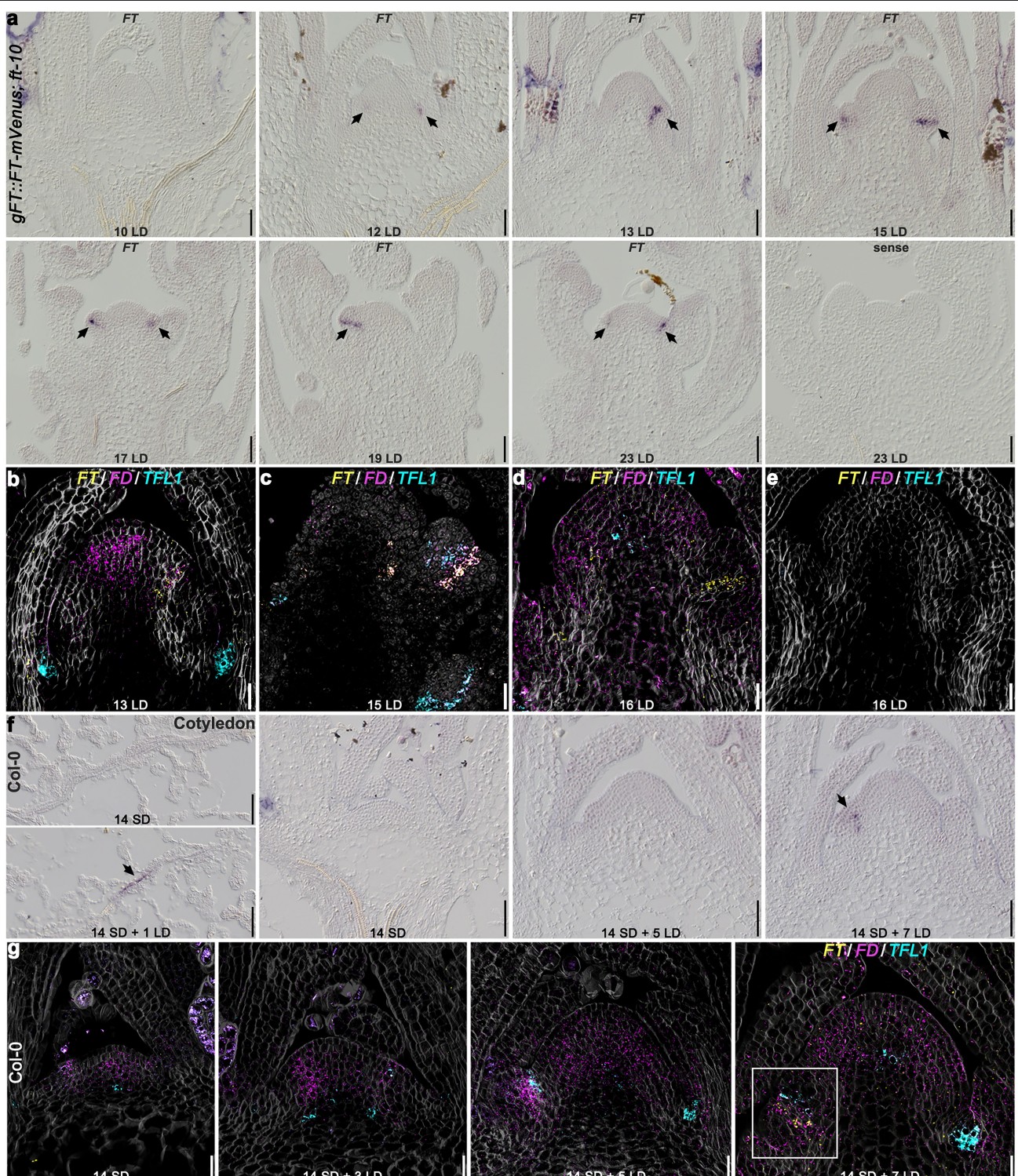

**Extended Data Fig. 3 | Spatiotemporal expression of *FT*, *FD* and *TFL1* mRNAs.**
**a**, RNA *in situ* hybridization images of shoot apical meristem cells expressing *gFT::FT-mVenus* in *ft-10* mutant at 10 to 23 LD. Arrows indicate *FT* signal. Scale bars = 50 μm. **b**–**d**, Confocal images of shoot apical meristem cells co-expressing *gFT::FT-mVenus* transgene (yellow), endogenous *FD* (magenta) and *TFL1* (turquoise) captured using multiplex fluorescent RNA *in situ* hybridization (RNAscope) at 13, 15 and 16 LD. Scale bars = 20 μm. **b**,**d**, Additional confocal sections at 13 and 16 LD, respectively, see also Fig. 2; **c**, a complete multiplex fluorescent RNA *in situ* hybridization section at 15 LD (close up image is shown in Fig. 2i). Probe for *FT* mRNA, *FD* mRNA and *TFL1* mRNA were used. **e**, Control

probe was used as a negative control (Method). Scale bars = 20 μm. **f**, RNA *in situ* hybridization images of cotyledon and shoot apical meristem cells expressing endogenous *FT* in Col-0 plants shifted to long days (LDs) after growth for 14 short days (SDs). Arrows indicate *FT* signal in cotyledons and shoot apices. Scale bars = 50 μm. **g**, Confocal images of Col-0 shoot apical meristem cells co-expressing endogenous *FT* (yellow), *FD* (magenta) and *TFL1* (turquoise) captured using multiplex fluorescent RNA *in situ* hybridization (RNAscope). Cells in the floral primordium that co-express endogenous *FT*, *FD*, and *TFL1* are highlighted with a rectangle. The cell walls (light gray) were stained with Renaissance 2200. Scale bars = 20 μm.

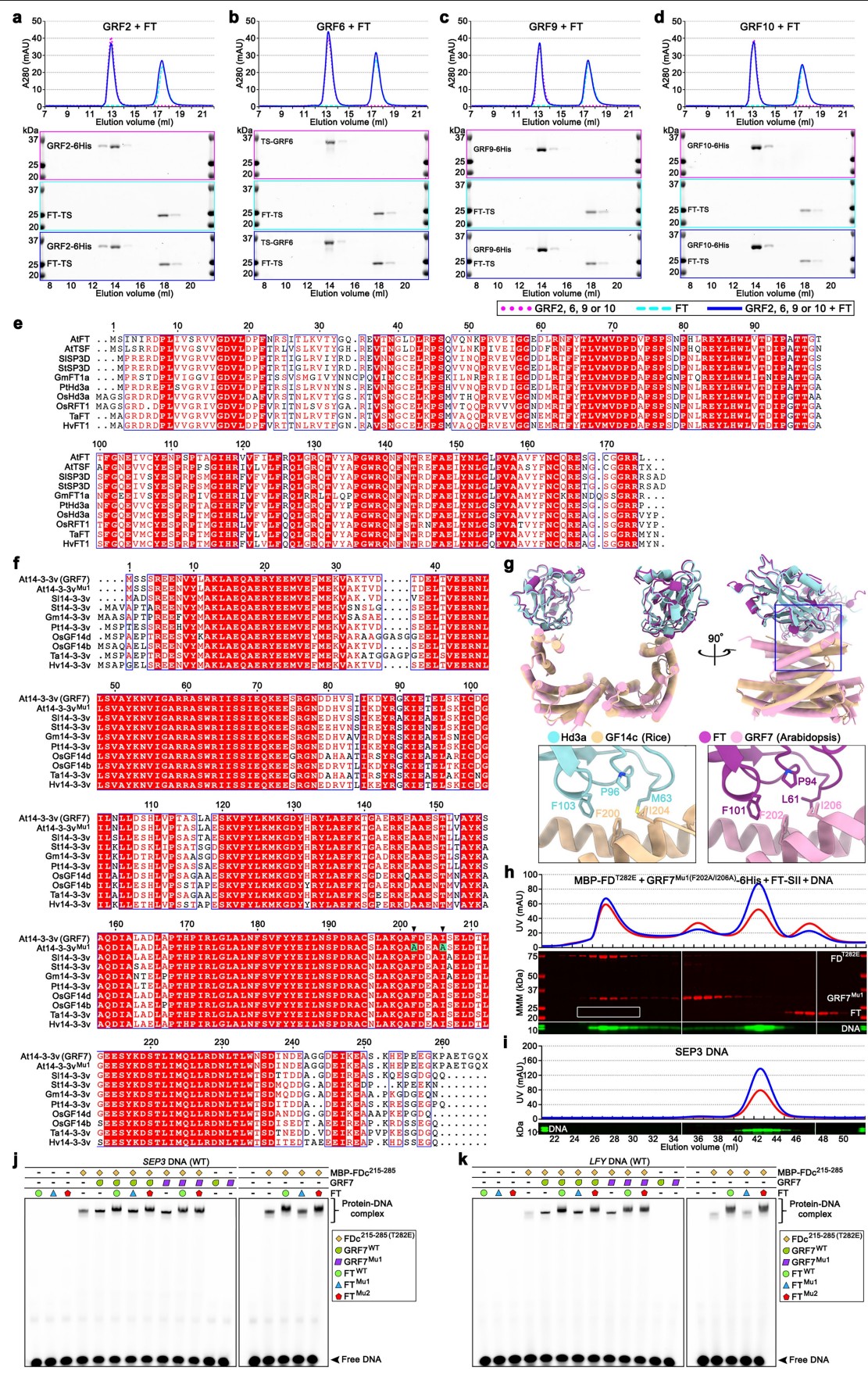

**Extended Data Fig. 4** | See next page for caption.

**Extended Data Fig. 4 | Gel-filtration analysis of interaction between FT-TwinStrep (FT-TS) and 14-3-3 proteins and the interaction interface between 14-3-3 and FT is highly conserved in different species. a–d**, Size-exclusion chromatography and gel analysis of direct interaction between GRF2 (14-3-3ω), GRF6 (14-3-3λ), GRF9 (14-3-3μ), GRF10 (14-3-3ε) and FT proteins. **e** and **f**, Alignment of FT and 14-3-3 proteins in Arabidopsis and some other species, respectively. At: *Arabidopsis thaliana*, Sl: *Solanum lycopersicum*, St: *Solanum tuberosum*, Gm: *Glycine max*, Pt: *Populus trichocarpa*, Os: *Oryza sativa*, Ta: *Triticum aestivum*, Hv: *Hordeum vulgare*. Sequences can be found in Supplementary Table 1. **g**, Structural alignment of FT − 14-3-3 complexes from rice (PDB: 3AXY) and Arabidopsis (predicted by Alphafold2). **h**, Size-exclusion chromatography and gel analysis of MBP-FD$^{T282E}$, FT, DNA and GRF7 mutant proteins. **i**, Size-exclusion chromatography and gel analysis of *SEP3* DNA (24 bp). **j,k**, Gel-shift analysis of the interactions between single MBP-FDc$^{T282E}$ WT or combination of GRF7 and FT WT or mutant proteins with *SEP3* WT (**j**) and *LFY* WT (**k**) DNA probes, respectively.

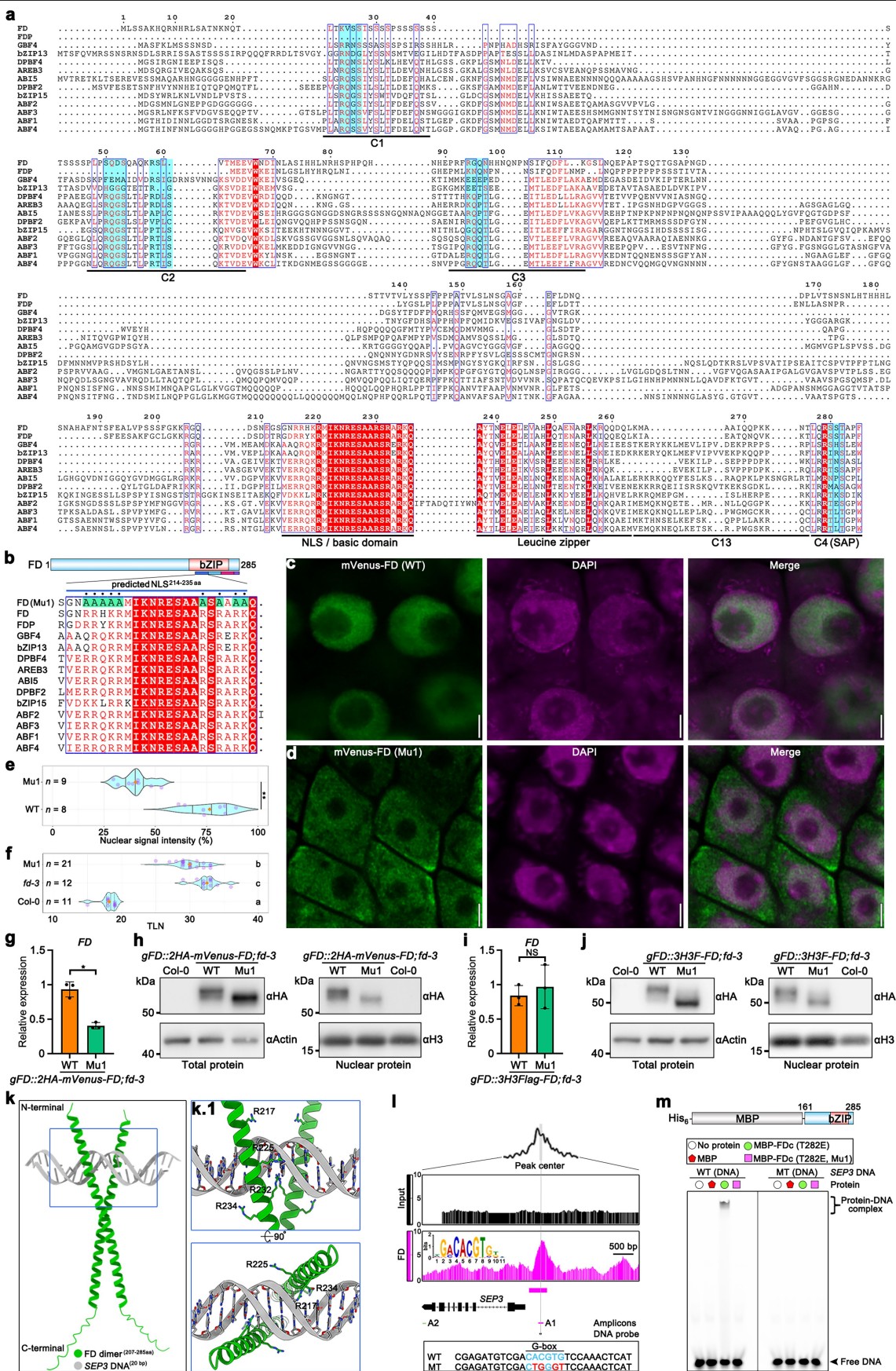

**Extended Data Fig. 5** | See next page for caption.

**Extended Data Fig. 5 | Dual functions of the basic domain of FD. a**, Multiple sequence alignment of group A bZIP proteins in Arabidopsis. Amino-acid residues conserved among all proteins are highlighted in red. Sequences can be found in Supplementary Table 1. **b**, Domain architecture of FD and alignment of the basic domain among group A bZIP proteins of *Arabidopsis*. **c**,**d**, Confocal images of shoot apical meristem cells of *fd-3* plants expressing *gFD::mVenus-FD* wild-type (WT) or Mu1. The nuclei (magenta) were stained with DAPI. Scale bars = 20 μm. **e**, Quantification of nuclear signal intensity of mVenus-FD WT and mVenus-FD Mu1 proteins in shoot apical meristem cells from (**c**) and (**d**). Mann-Whitney U test was used to test whether the median of signal intensity of WT and Mu1 are different. *Alpha* value, WT *vs.* Mu1 = 0.00016. **f**, Flowering time (total leaf number, TLN) of transgenic *fd-3* plants carrying *gFD::3HA3Flag-FD* Mu1. Different letters represent significant differences among genotypes (*P* < 0.05, using one-way ANOVA followed by Tukey's pairwise multiple comparison, $P = 1.11 * 10^{-16}$); *n* = 21 for *gFD::3HA3Flag-FD* Mu1, n = 12 for *fd-3*, and n = 11 for Col-0, respectively, all grown under long-days (LDs). **g**, RT-qPCR analysis of *2HA-mVenus-FD* mRNA levels in shoot apices of transgenic *fd-3* mutant plants carrying *gFD::2HA-mVenus-FD^WT* and *gFD::2HA-mVenus-FD^Mu1*. All values are normalized to *ACTIN2* levels. Data are presented as mean values ± SEM

of three biological replicates. Statistical significance was determined by pairwise one-sided *t*-test [*P* (FD^WT *vs.* FD^Mu1) = 0.001527. Shoot apices were harvested from seedlings grown for 7 LDs. **h**, Western blotting analysis of the abundance of 2HA-mVenus-FD proteins in total (left panel) and nuclear (right panel) extractions in 7 LD-grown seedlings. **i**, RT-qPCR analysis of *3HA-3Falg-FD* mRNA levels in shoot apices of transgenic *fd-3* mutant plants carrying *gFD::3HA-3Flag-FD^WT* and *gFD::3HA-3Flag-FD^Mu1*. All values are normalized to *ACTIN2* levels. Data are presented as mean values ± SEM of three biological replicates. Statistical significance was determined by pairwise one-sided *t*-test [*P* (FD^WT *vs.* FD^Mu1) = 0.563. Shoot apices were harvested as described in (g). **j**, Western blotting analysis of the abundance of 3HA-3Falg-FD proteins in total (left panel) and nuclear (right panel) extractions in 7 LD-grown seedlings. **k**, Modeled structure of the FD^bZIP–DNA complex. **k.1**, Close-up of the predicted FD residues that interact with *SEP3* DNA are colored by heteroatom. **l**, Windows for ChIP-seq analysis of the FD-binding profile to the *SEP3* gene. The *SEP3* DNA probes (28 bp) for EMSA experiments in (**m**) and Extended Data Fig. 4j are shown. **m**, Interactions between FD^bZIP WT and Mu1 proteins and DNA probes analyzed by EMSA (gel shift). Western blots in **h**,**j** are are representative of two independent experiments.

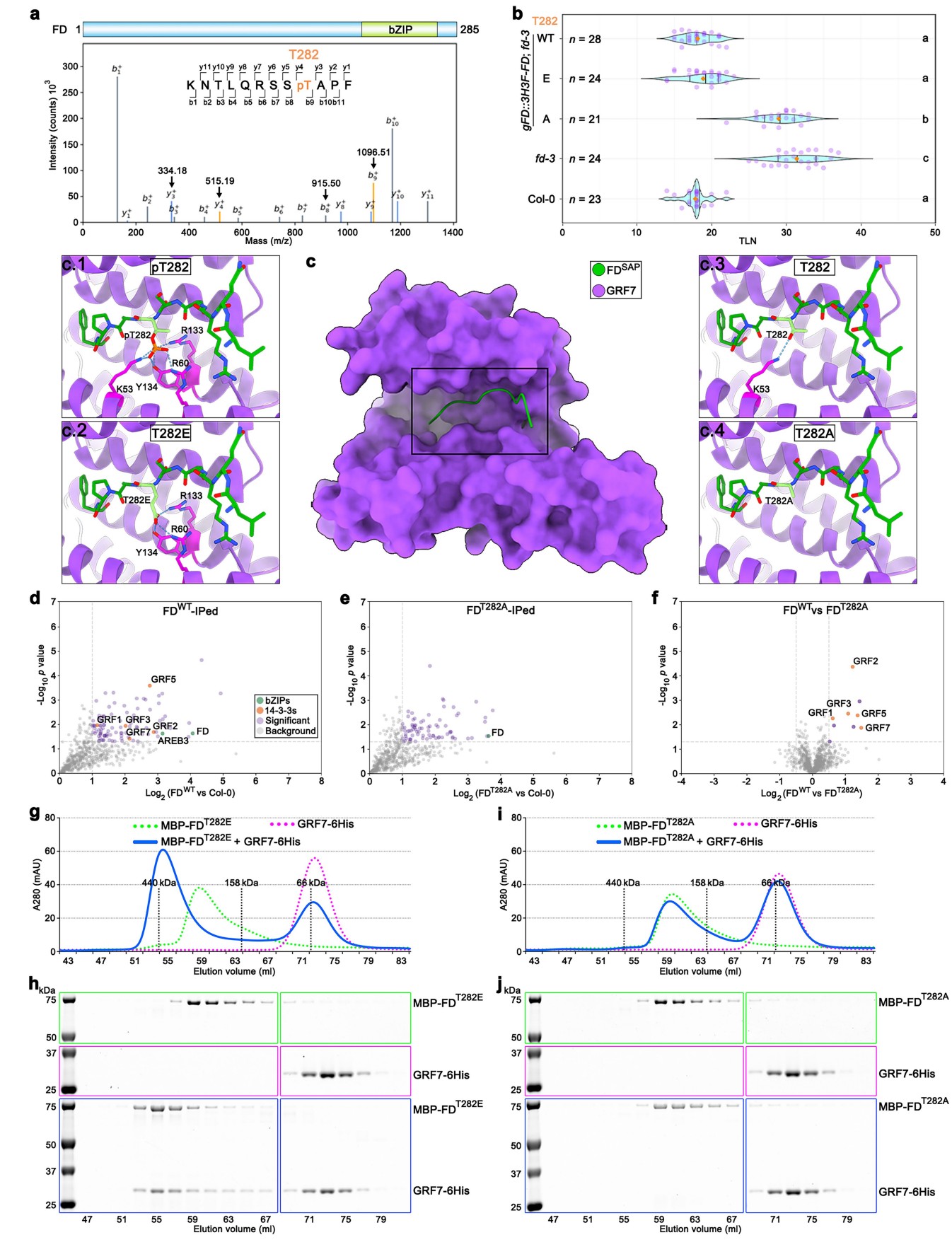

**Extended Data Fig. 6 |** See next page for caption.

**Extended Data Fig. 6 | Phosphorylation at T282 of FD is essential for its interaction with 14-3-3 proteins *in vivo* and *in vitro*. a**, The phosphopeptide of FD was analyzed by LC-MS/MS. The phosphorylation of T282 is revealed in MS2 spectra. The observed fragment ions are labeled on the spectrum and peptide sequence (phosphorylated T282 is marked in orange). **b**, Flowering time (total leaf number, TLN) of transgenic *fd-3* mutant plants carrying *gFD::3HA-3Flag-FD^WT^*, *gFD::3HA-3Flag-FD^T282E^* or *gFD::3HA-3Flag-FD^T282A^* in LDs. Different letters represent significant differences among genotypes ($P < 0.05$), using one-way ANOVA followed by Tukey's pairwise multiple comparison, $P = 1.11 * 10^{-16}$. **c**, Modeled structure of the FD^SAP^–GRF7 complex with phosphorylation, wild type (WT) or mutations on residue T282 (**c.1**–**c.4**). **d**–**f**, Volcano plots showing differential protein abundance based on label-free quantification. The $\log_2$-transformed fold change between genotypes FD^WT^ to Col-0, FD^T282A^ to Col-0 or FD^WT^ to FD^T282A^ is plotted against the $-\log_{10}$ of the adjusted $p$-value. Statistical significance was assessed using a two-sided $t$-test with permutation-based false discovery rate (FDR) correction (AdaPT method). Proteins with $\log_2$ (fold change) $> 1$ and FDR-adjusted $p < 0.05$ are highlighted in distinct colors as significant outliers. **g**–**j**, Size-exclusion chromatography and gel analysis of GRF7 with MBP-FD^2–285(T282E)^ (**g**, **h**) or MBP-FD^2–285(T282A)^ (**i**, **j**). Data are representative of two independent experiments.

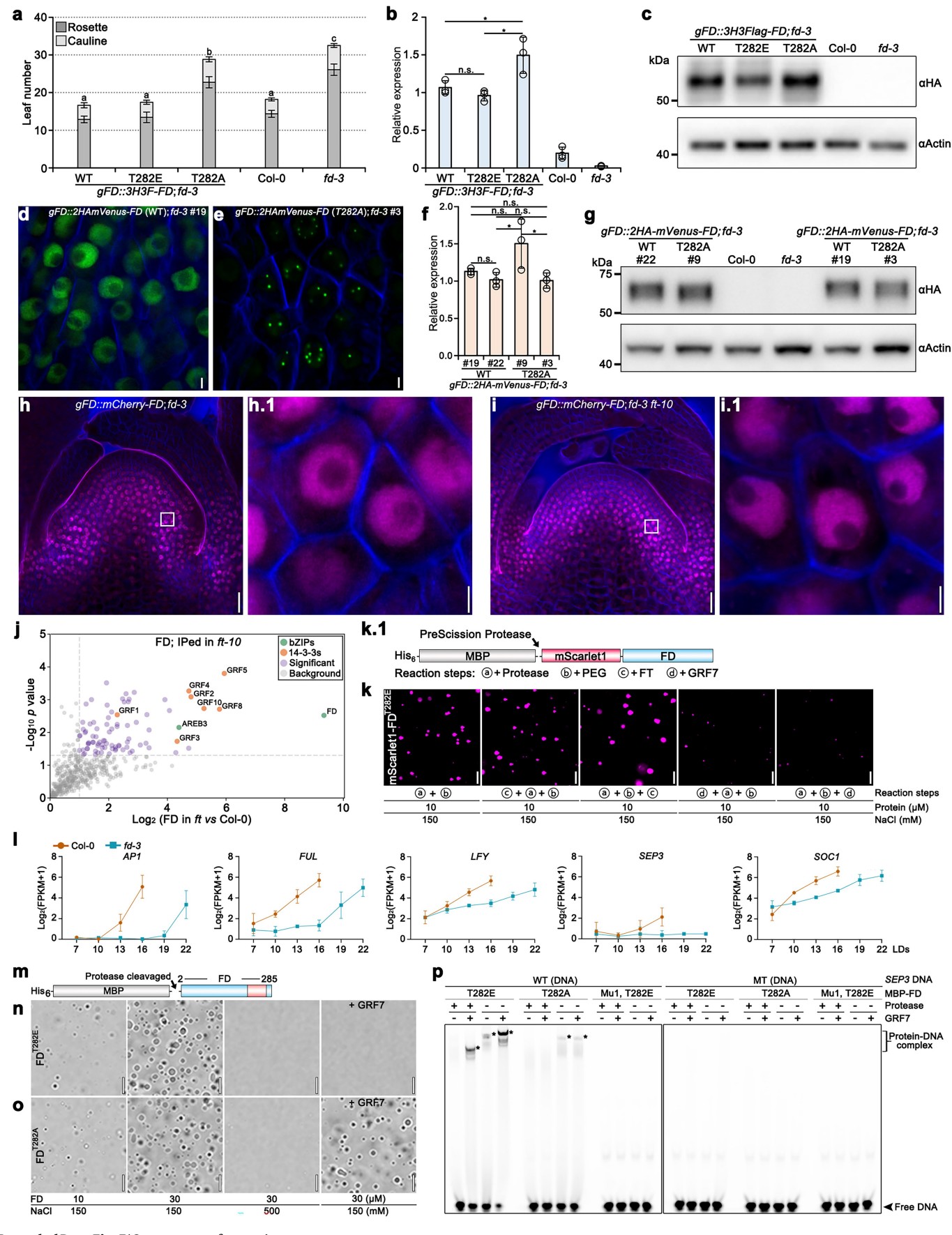

**Extended Data Fig. 7** | See next page for caption.

**Extended Data Fig. 7 | The phosphorylation status at T282 does not affect the *in vivo* stability of FD protein, and 14-3-3 proteins, rather than FT protein, repress FD condensation *in vivo* and *in vitro*. a**, Flowering time [rosette leaf number (R) and cauline leaf number (C)] of T3 homozygous populations of transgenic *fd-3* mutant plants carrying *gFD::3HA-3Flag-FD^{WT}*, *gFD::3HA-3Flag-FD^{T282E}* or *gFD::3HA-3Flag-FD^{T282A}*. Different letters represent significant differences among genotypes ($P < 0.05$, using one-way ANOVA followed by Tukey's pairwise multiple comparison, $P = 1.11 * 10^{-16}$). The sample sizes were as follows: $n = 16$ for *gFD::3HA-3Flag-FD^{WT}*, $n = 14$ for *gFD::3HA-3Flag-FD^{T282E}*, $n = 12$ for *gFD::3HA-3Flag-FD^{T282A}*, and $n = 11$ and 12 for Col-0 and *fd-3* seedlings, respectively, all grown under long-days (LDs). **b**, RT-qPCR analysis of *FD* mRNA levels in shoot apices of transgenic *fd-3* mutant plants carrying *gFD::3HA-3Flag-FD^{WT}*, *gFD::3HA-3Flag-FD^{T282E}*, *gFD::3HA-3Flag-FD^{T282A}*, as well as Col-0 and *fd-3* plants. All values are normalized to *ACTIN2* levels. Data are the mean ± SEM of three biological replicates ($n = 3$). Statistical significance was determined by pairwise two-sided *t*-test [$P$(FD^{WT} *vs.* FD^{T282E}) = 0.1914; $P$(FD^{WT} *vs.* FD^{T282A}) = 0.0444; $P$(FD^{T282E} *vs.* FD^{T282A}) = 0.0210]. Asterisks indicate significant differences (*$P < 0.05$); n.s., not significant ($P > 0.05$). Shoot apices were harvested from seedlings grown for 7 LDs. **c**, Western blotting analysis of the abundance of 3HA-3Flag-FD proteins in 7 LD-grown seedlings of the genotypes described in (**a**). **d**,**e**, Confocal images of shoot apical meristem cells of *fd-3* plants expressing *gFD::mVenus-FD* (green) *WT* #19 or *T282A* #3. Cell walls (blue) were stained with Renaissance 2200. Scale bars = 2 μm. **f**, RT-qPCR analysis of *FD* mRNA levels in shoot apices of transgenic *fd-3* mutant plants carrying *gFD::2HA-mVenus-FD^{WT}* and *gFD::2HA-mVenus-FD^{T282A}*. All values are normalized to *ACTIN2* levels. Data are the mean ± SEM of three biological replicates ($n = 3$). Statistical significance was determined by pairwise two-sided *t*-test [$P$(FD^{WT #19} *vs.* FD^{WT #22}) = 0.8709; $P$(FD^{WT #19} *vs.* FD^{T282A #9}) = 0.1293; $P$(FD^{WT #19} *vs.* FD^{T282A #3}) = 0.8384; $P$(FD^{WT #22} *vs.* FD^{T282A #9}) = 0.0445; $P$(FD^{WT #22} *vs.* FD^{T282A #3}) = 0.9998; $P$(FD^{T282A #9}

*vs.* FD^{T282A #3}) = 0.0401]. Asterisks indicate significant differences (*$P < 0.05$); n.s., not significant ($P > 0.05$). Shoot apices were harvested from seedlings grown for 7 LDs. **g**, Western blotting analysis of the abundance of 2HA-mVenus-FD proteins in 7 LD-grown seedlings of the genotypes described in (**d**) and (**e**). Western blots represent one of two independent biological replicates in (**c**) and (**g**). ACTIN was used as the loading control. **h**, **i**, Confocal images of shoot apical meristem cells of *fd-3* or *fd-3 ft-10* plants expressing *gFD::mCherry-FD* wild type (WT). Scale bars = 20 μm. **h.1**,**i.1**, Close-up images of cells from (N) and (O), respectively. Scale bars = 2 μm. **j**, Volcano plots showing differential protein abundance for FD in *ft-10* mutants based on label-free quantification. The $\log_2$-transformed fold change between genotypes is plotted against the $-\log_{10}$ of the adjusted *p*-value. Statistical significance was assessed using a two-sided *t*-test with permutation-based false discovery rate (FDR) correction (AdaPT method). Proteins with $\log_2$ (fold change) > 1 and FDR-adjusted $p < 0.05$ are highlighted in distinct colors as significant outliers. **k.1**, Schematic of protein fusion used for in vitro phase separation assay and phase diagram of mScarlet1-FD2-285^{T282E} droplets. **k**, In vitro phase separation of mScarlet1-FD2-285^{T282E} droplets in the present or absent of GRF7 or FT. Scale bars = 10 μm. **l**, RNA-seq data for FD target genes in shoot apices of Col-0 wild type and *fd-3* mutant across various developmental stages[20]. Data are the mean ± SEM of three biological replicates ($n = 3$). **m**, Schematic of FD protein fusion used for in vitro phase separation assay. **n**, **o**, Phase diagram of FD^{T282E} and FD^{T282A} droplets, respectively. Formation of FD^{T282E} protein droplets was captured by optical microscopy. Scale bars = 10 μm. **p**, Gel-shift analysis of the interactions between single MBP-FD or combination of GRF7 or protease to remove the MBP tag from MBP-FD fusion proteins with wild-type (WT) or mutant *SEP3* DNA probes. The confocal images in **d**,**e**,**h**,**i** are representative of three independent meristems. The results in **k**,**n**,**o** are representative of three independent experiments.

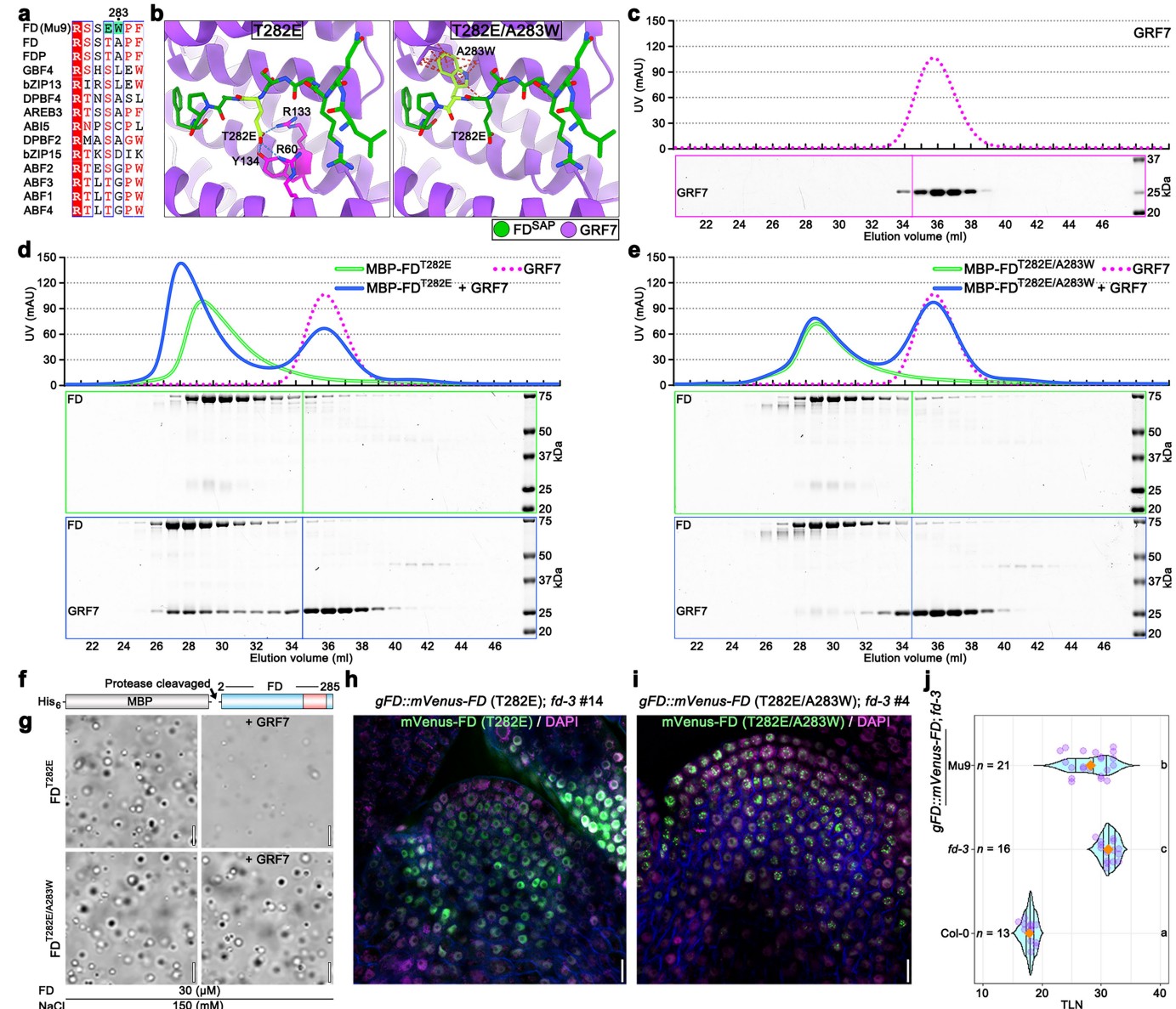

**Extended Data Fig. 8 | The A283W substitution mutation disrupts the interaction between the phosphomimic T282E FD and 14-3-3 proteins, resulting in FD condensation both *in vivo* and *in vitro*. a**, Multiple sequence alignment of the C4 (SAP) domain segment in group A bZIP proteins of Arabidopsis. Mutations in Mu9 are also shown. Amino-acid residues conserved among all proteins are highlighted in red. **b**, Modeled structure of the FD$^{SAP}$–GRF7 complex with mutations on residue T282E (left) and T282E/A283W (right). **c**, Size-exclusion chromatography and gel analysis of GRF7 protein. **d**, Size-exclusion chromatography and gel analysis of MBP-FD$^{2-285(T282E)}$ and GRF7 proteins. **e**, Size-exclusion chromatography and gel analysis of MBP-FD$^{2-285(T282E/A283W)}$ and GRF7 proteins. Data in **d**,**e** are representative of two independent experiments with similar results. **f**, Schematic of FD protein fusion used for in vitro phase separation assay. **g**, Phase diagram of FD$^{T282E}$ (top) and FD$^{T282E/A283W}$ (bottom)

droplets with or without GRF7 protein. Formation of FD$^{T282E}$ and FD$^{T282E/A283W}$ protein droplets was captured by optical microscopy. Scale bars = 10 µm. Data are representative of three independent experiments. **h**,**i**, Confocal images of shoot apical meristem cells of *fd-3* plants expressing *gFD::mVenus-FD$^{T282E}$* or *gFD::mVenus-FD$^{T282E/A283W}$*, respectively. Cell walls (blue) and nuclei (magenta) were stained with Direct Red 23 and DAPI, respectively. Scale bars = 20 µm. The confocal images are representative of three independent meristems. **j**, Flowering time (total leaf number, TLN) of transgenic *fd-3* mutant plants carrying *gFD::mVenus-FD$^{T282E/A283W}$*. Different letters represent significant differences among genotypes ($P < 0.05$, using one-way ANOVA followed by Tukey's pairwise multiple comparison, $P = 1.11*10^{-16}$); $n = 21$ for *gFD::mVenus-FD$^{T282E/A283W}$*, n = 16 for *fd-3*, and n = 13 for Col-0 seedlings respectively, all grown under long-days (LDs).

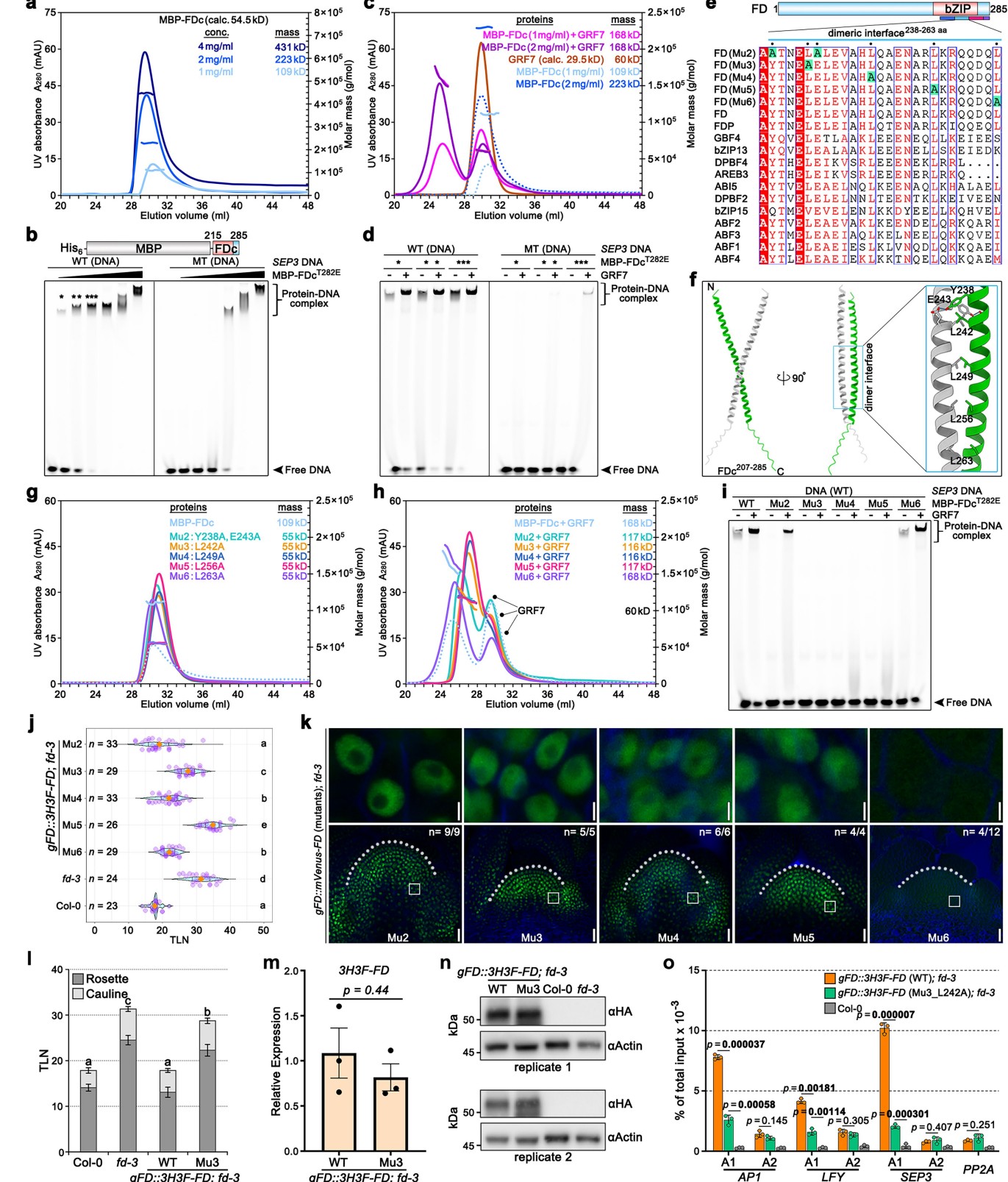

**Extended Data Fig. 9** | See next page for caption.

**Extended Data Fig. 9 | 14-3-3 proteins enhance FD DNA binding by reducing oligomerization, and increasing dimerization, and altering its C-terminal conformation. a,c**, SEC-MALS analysis of purified MBP-FDc$^{T282E}$ protein alone (**a**) or with GRF7 (**c**) at increasing protein concentrations as shown in the graph. **b,d**, Gel-shift analysis of interactions between single MBP-FDc$^{T282E}$ (**b**), or combination of MBP-FDc$^{T282E}$ and GRF7 (**d**) with WT and mutant DNA probes at increasing FD protein concentrations. Similar concentration ranges of MBP-FDc$^{T282E}$ in (**b**) and (**d**) are marked by an asterisk. **e**, Domain architecture of FD and alignment of the dimeric region of group A bZIP proteins of Arabidopsis. Mutations (Mu2 to Mu6) are also shown. **f**, Modeled structure of FDc homodimer and a close-up of the predicted residues that contribute to dimer formation. **g,h**, SEC-MALS analysis of purified MBP-FDc$^{T282E}$ mutant proteins alone (**g**) or with GRF7 (**h**). **i**, Gel-shift analysis of the interactions between single MBP-FDc$^{T282E}$ WT or mutant proteins in the presence/absence of GRF7 and with the WT DNA probe. Curves, elution profiles on column, line traces and molar masses, are as derived from MALS in (**a**), (**c**), (**g**) and (**h**). **j**, Flowering time (total leaf number, TLN) of transgenic *fd-3* plants carrying *gFD::3HA-3Flag-FD* Mu2–Mu6. Different letters represent significant differences among genotypes (*P* < 0.05, using one-way ANOVA followed by Tukey's pairwise multiple comparison, *P* = 1.11*10$^{-16}$); *n* = 33, 29, 33, 26, and 29 for *gFD::3HA-3Flag-FD* Mu2, Mu3, Mu4, Mu 5, and Mu6, respectively; and n = 24 and 23 for *fd-3* and Col-0 seedlings respectively, all grown under long-days (LDs). **k**, Confocal images of shoot apical meristem cells of *fd-3* plants expressing *gFD::mVenus-FD Mu2–Mu6*. Bar = 20 μm (bottom) and 2 μm (top). *n* represents the number of T1 plants for which signal could be detected. **l**, Flowering time [total leaf number, TLN, composed of rosette leaf number (RLN) and cauline leaf number (CLN)] of T3 homozygous populations of transgenic *fd-3* mutant plants carrying *gFD::3HA-3Flag-FD$^{WT}$* and *gFD::3HA-3Flag-FD$^{Mu3,L242A}$*. Different letters represent significant differences among genotypes (*P* < 0.05, using one-way ANOVA followed by Tukey's pairwise multiple comparison, *P* = 1.11*10$^{-16}$); The sample sizes were as follows: *n* = 13 for *gFD::3HA-3Flag-FD$^{WT}$*, *n* = 22 for *gFD::3HA-3Flag-FD$^{Mu3}$*, and *n* = 13 and 16 for Col-0 and *fd-3* seedlings, respectively, all grown under long-days (LDs). See also Supplementary Table 1. **m**, RT-qPCR analysis of *3HA-3Flag-FD* mRNA levels in shoot apices of transgenic *fd-3* mutant plants carrying *gFD::3HA-3Flag-FD$^{WT}$* and *gFD::3HA-3Flag-FD$^{Mu3,L242A}$*. All values are normalized to *ACTIN2* levels. Data are presented as mean values ± SEM of three biological replicates. Statistical significance was determined by pairwise one-sided *t*-test [*P* (FD$^{WT}$ *vs.* FD$^{Mu3}$) = 0.44. **n**, Western blotting analysis of the abundance of 3HA-3Flag-FD proteins in 7 LD-grown seedlings of the genotypes described in **l**. Western blots are representative of two independent experiments. **o**, ChIP–qPCR analysis of FD enrichment in target gene regulatory regions in transgenic *fd-3* plants expressing *gFD::3HA3Flag-FD* and *gFD::3HA3Flag-FD$^{Mu3}$*. Data are represented as means ± SEM of three independent experiments. Statistical significance was determined by pairwise one-sided *t*-test. The significant differences (*P* < 0.05) are highlighted in bold.

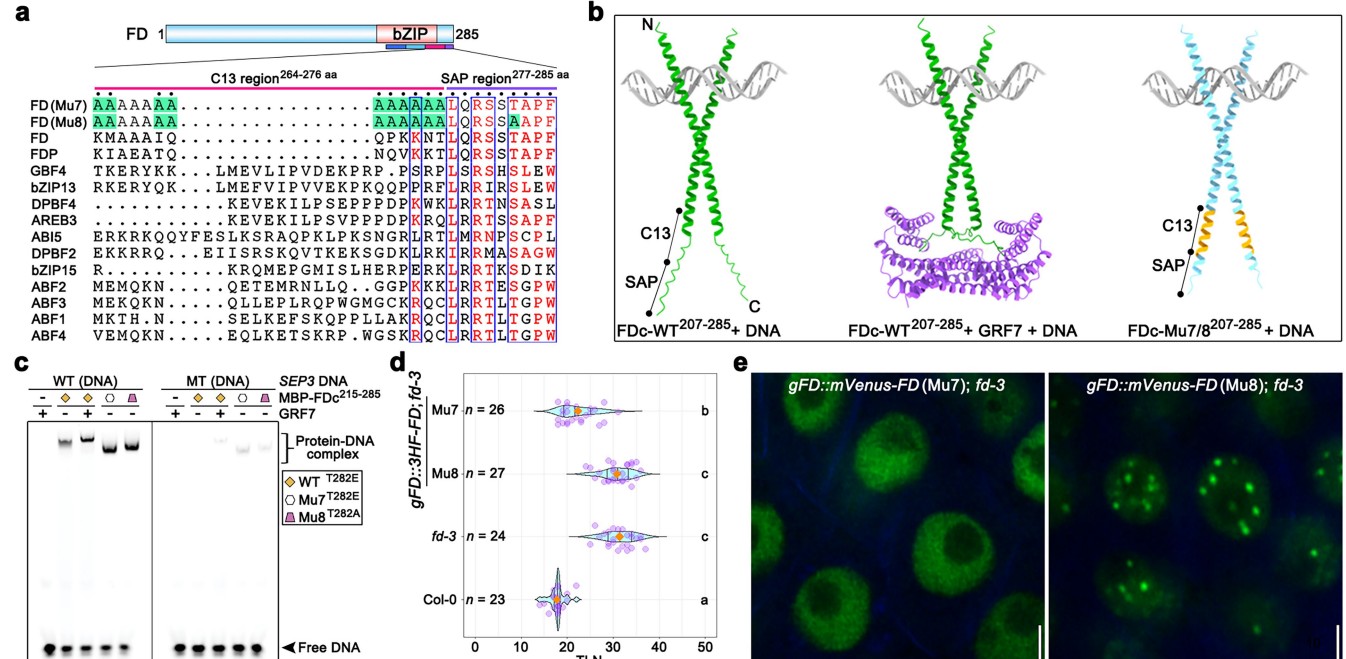

**Extended Data Fig. 10 | 14-3-3 proteins enhance FD DNA binding by altering its C-terminal conformation. a**, Domain architecture of FD and alignment of C13 and SAP regions from group A bZIP proteins of Arabidopsis. Mutations (Mu7 and Mu8) of FD are also shown. **b**, Modeled and predicted structure of the DNA-bound FDc homodimer, DNA-bound FDc and GRF7 heterotetramer, and DNA-bound FDc Mu7 and Mu8. **c**, Gel-shift analysis of the interactions between single MBP-FDc^T282E wild–type (WT) or mutant proteins or combination of GRF7 and MBP-FDc proteins with WT and MT DNA probes. **d**, Flowering time (total leaf number, TLN) of transgenic *fd-3* plants carrying *gFD::3HA3Flag-FD* Mu7 and Mu8.

Different letters represent significant differences among genotypes (*P* < 0.05, using one-way ANOVA followed by Tukey's pairwise multiple comparison, *P* = 1.11 * 10^−16; *n* = 26 and 27 for *gFD::3HA3Flag-FD* Mu7 and Mu8, respectively; and n = 24 and 23 for *fd-3* and Col-0 seedlings respectively, all grown under long-days (LDs). The same data for Col-0 and *fd-3* are also shown in **j. e**, Confocal images of shoot apical meristem cells of *fd-3* plants expressing *gFD::mVenus-FD* Mu7 and Mu8. Scale bars = 2 μm. The confocal images are representative of three independent meristems. Gel-shift assays in **c** were performed twice with similar results.

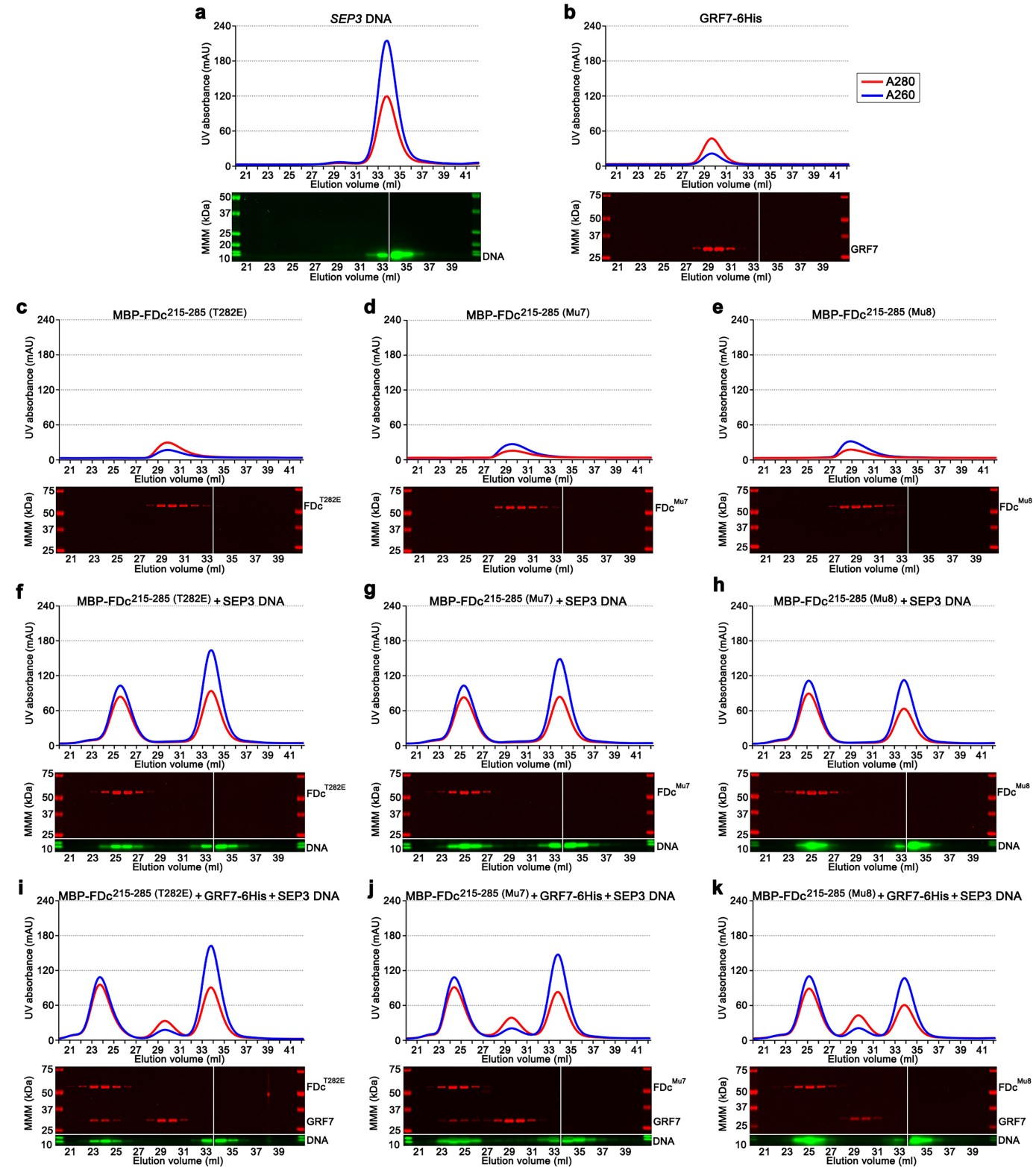

**Extended Data Fig. 11 | Gel-filtration analysis of MBP-FDc$^{215-285}$ WT and Mu7 and Mu8 mutant proteins with or without *SEP3* DNA and in combination with the presence or absence of GRF7-6His protein.** Each panel shows size-exclusion chromatography and gel analysis: **a**, *SEP3* DNA (24 bp). **b**, GRF7-6His. **c–e**, MBP-FDc$^{215-285(T282E)}$, MBP-FDc$^{215-285(Mu7,T282E)}$ and MBP-FDc$^{215-285(Mu8,T282A)}$, respectively. **f–h**, MBP-FDc$^{215-285(T282E)}$, MBP-FDc$^{215-285(Mu7,T282E)}$ and MBP-FDc$^{215-285(Mu8,T282A)}$, respectively, with *SEP3* DNA. **i–k**, MBP-FDc$^{215-285(T282E)}$, MBP-FDc$^{215-285(Mu7,T282E)}$ and MBP-FDc$^{215-285(Mu8,T282A)}$, respectively, with *SEP3* DNA and GRF7-6His. The x- and y-axes indicate the elution volume and the protein absorption at 260 nm and 280 nm, respectively.

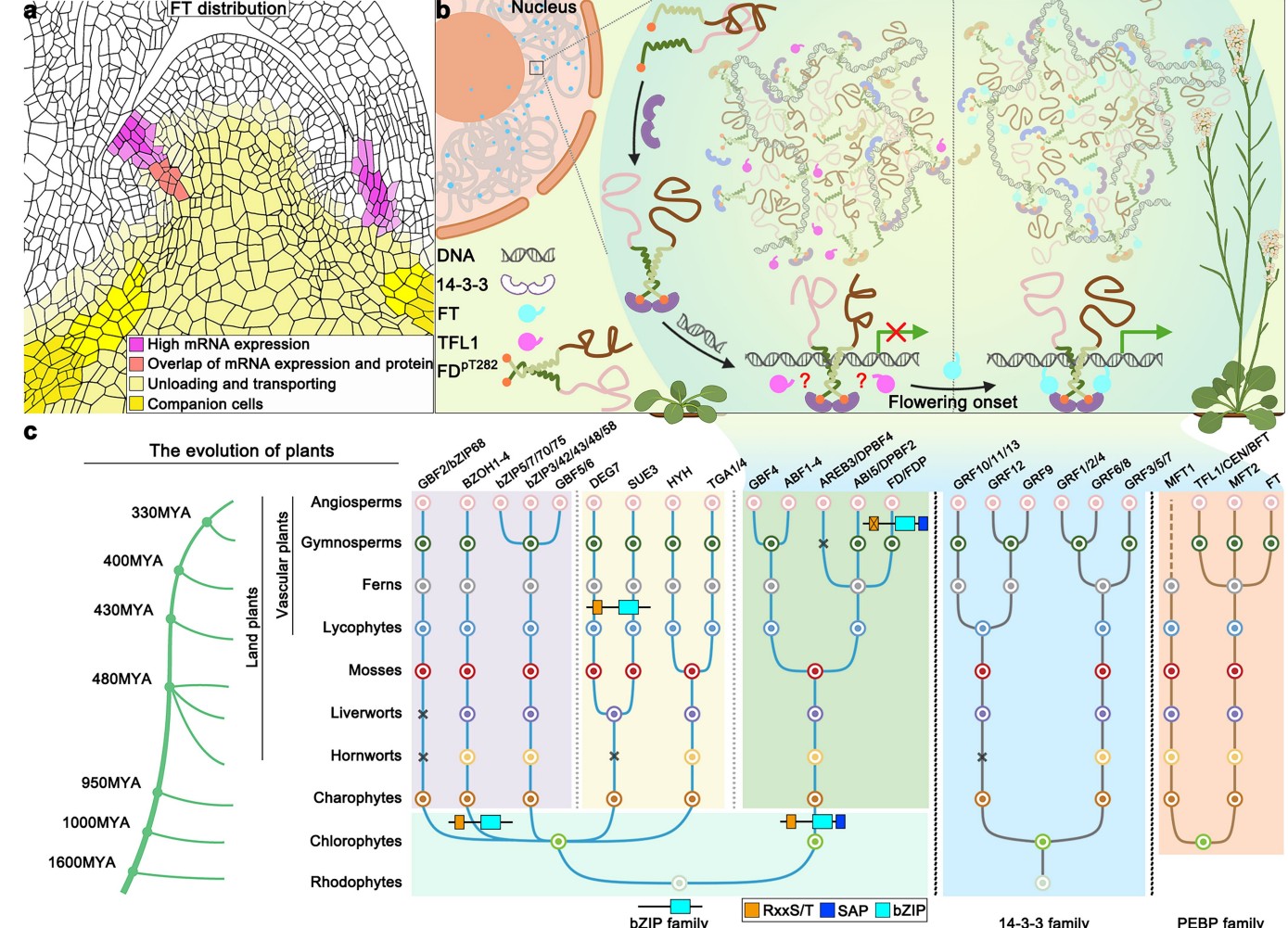

**Extended Data Fig. 12 | Model of Arabidopsis FAC formation and inferred scenario of bZIP, 14-3-3 and PEBP gene family evolution within the plant kingdom based on extended phylogeny. a**, SAM regions where FT is distributed and potential regions where transcribed mRNAs and transported proteins overlap. **b**, Model showing phosphorylated FD, 14-3-3 and FT proteins forming active small phase-separated condensates at specific DNA-binding sites of target genes during flowering. TFL1 is proposed to repress activity of FD in the vegetative stage, but how it is recruited into the FD − 14-3-3 complex is largely unknown. The mechanism in this study probably applies to a large clade of related plant bZIPs, 14-3-3 s and PEBPs, and is therefore highlighted and connected to these families in (**c**). Created in BioRender. Gao, H. (2025) https://BioRender. com/dkwe9w4. **c**, Circles with dots represent the inferred ancestral gene existence at each point of divergence of major plant groups; numbers on the left are estimated time of divergence expressed as millions of years ago (MYA).

# Reporting Summary

## Statistics

For all statistical analyses, confirm that the following items are present in the figure legend, table legend, main text, or Methods section.

| n/a | Confirmed | |
|---|---|---|
| ☐ | ☒ | The exact sample size (*n*) for each experimental group/condition, given as a discrete number and unit of measurement |
| ☐ | ☒ | A statement on whether measurements were taken from distinct samples or whether the same sample was measured repeatedly |
| ☐ | ☒ | The statistical test(s) used AND whether they are one- or two-sided *Only common tests should be described solely by name; describe more complex techniques in the Methods section.* |
| ☐ | ☒ | A description of all covariates tested |
| ☐ | ☒ | A description of any assumptions or corrections, such as tests of normality and adjustment for multiple comparisons |
| ☐ | ☒ | A full description of the statistical parameters including central tendency (e.g. means) or other basic estimates (e.g. regression coefficient) AND variation (e.g. standard deviation) or associated estimates of uncertainty (e.g. confidence intervals) |
| ☐ | ☒ | For null hypothesis testing, the test statistic (e.g. *F*, *t*, *r*) with confidence intervals, effect sizes, degrees of freedom and *P* value noted *Give P values as exact values whenever suitable.* |
| ☒ | ☐ | For Bayesian analysis, information on the choice of priors and Markov chain Monte Carlo settings |
| ☐ | ☒ | For hierarchical and complex designs, identification of the appropriate level for tests and full reporting of outcomes |
| ☒ | ☐ | Estimates of effect sizes (e.g. Cohen's *d*, Pearson's *r*), indicating how they were calculated |

*Our web collection on statistics for biologists contains articles on many of the points above.*

## Software and code

Policy information about availability of computer code

| | |
|---|---|
| Data collection | For confocal microscopes, we used Zeiss LSM980 and LSM880. For size exclusion chromatography and multi-angle light scattering (SEC-MALS), we used AKTÄ pure 25M (AKTÄ) coupled to a miniDAWN multi-angle light scattering detector (Wyatt Technology) as well as a refractive index detector (Shodex RI-501). |
| Data analysis | For confocal microscopes, we used Zen 3.10 software (Zeiss). For imaging data processing and analysis, we used Fiji (2.16.0), Cellpose (2.2.3) and Matlab (9.13.0 2022b). For SEC-MALS, we used ASTRA (8.2) software. For analysis of masspectrometry proteomics data we used Perseus version 1.5.8.5. Custom code is developed to analyze signal co-localizations at the shoot apex meristem. Source code for analysis signal and figure generation is publicly available at GitHub: https://gitlab.com/GRM_14/gao_ding_et_al_2025/-/tree/011d3d70fc1c0f41670c6f0b860b3c586c8949fd |

For manuscripts utilizing custom algorithms or software that are central to the research but not yet described in published literature, software must be made available to editors and reviewers. We strongly encourage code deposition in a community repository (e.g. GitHub). See the Nature Portfolio guidelines for submitting code & software for further information.

## Data

Policy information about availability of data

All manuscripts must include a data availability statement. This statement should provide the following information, where applicable:

- Accession codes, unique identifiers, or web links for publicly available datasets
- A description of any restrictions on data availability
- For clinical datasets or third party data, please ensure that the statement adheres to our policy

> The mass spectrometry proteomics data have been deposited and are accessible at the ProteomeXchange Consortium via the PRIDE78 partner repository with the dataset identifier PXD067955. The full versions of Western blots are shown in Supplementary Fig. 4. The data behind graphs are provided in Supplementary Table 1.

## Research involving human participants, their data, or biological material

Policy information about studies with human participants or human data. See also policy information about sex, gender (identity/presentation), and sexual orientation and race, ethnicity and racism.

| Reporting on sex and gender | n/a |
|---|---|
| Reporting on race, ethnicity, or other socially relevant groupings | n/a |
| Population characteristics | n/a |
| Recruitment | n/a |
| Ethics oversight | n/a |

Note that full information on the approval of the study protocol must also be provided in the manuscript.

# Field-specific reporting

Please select the one below that is the best fit for your research. If you are not sure, read the appropriate sections before making your selection.

☒ Life sciences    ☐ Behavioural & social sciences    ☐ Ecological, evolutionary & environmental sciences

For a reference copy of the document with all sections, see nature.com/documents/nr-reporting-summary-flat.pdf

# Life sciences study design

All studies must disclose on these points even when the disclosure is negative.

| Sample size | Leaf numbers were used to estimate flowering time of Arabidopsis plants (see in figure legends). Sample sizes (n ≥ 10 per group) were chosen based on established norms in flowering time researches to ensure robust detection of the large effect sizes typical in our studies. Statistical analysis was performed via one-way ANOVA followed by Tukey's pairwise multiple comparison. |
|---|---|
| Data exclusions | No data were excluded. |
| Replication | Flowering time for wild type, mutant and T1 transgenic plants were measured one time for each experiment, and flowering time of crucial transgenic plants were further confirmed one time in T3 homozygous plants with similar results. |
| Randomization | Seedlings were grown randomly in conditional growth chambers and data were collected randomly. |
| Blinding | The investigators were blinded for data collection and/or analysis. |

# Reporting for specific materials, systems and methods

We require information from authors about some types of materials, experimental systems and methods used in many studies. Here, indicate whether each material, system or method listed is relevant to your study. If you are not sure if a list item applies to your research, read the appropriate section before selecting a response.

## Materials & experimental systems

| n/a | Involved in the study |
|---|---|
| ☐ | ☒ Antibodies |
| ☒ | ☐ Eukaryotic cell lines |
| ☒ | ☐ Palaeontology and archaeology |
| ☒ | ☐ Animals and other organisms |
| ☒ | ☐ Clinical data |
| ☒ | ☐ Dual use research of concern |
| ☐ | ☒ Plants |

## Methods

| n/a | Involved in the study |
|---|---|
| ☒ | ☐ ChIP-seq |
| ☒ | ☐ Flow cytometry |
| ☒ | ☐ MRI-based neuroimaging |

# Antibodies

| | |
|---|---|
| Antibodies used | The following commercial antibodies were used in this study:<br>anti-HA (Abcam, ab9110); anti-HA-HRP (Roche, 12013819001, 3F10); anti-ALFA-HRP (NanoTag, N1505, 1G5); anti-Actin (Santa Cruz, sc-47778 HRP, C4); anti-StrepMAB-Classic-HRP (IBA, 2-1509-001); anti-H3-HRP (Abcam, ab1791).<br><br>anti-HA (Roche, 12013819001) were used at 1:1000-fold dilution for western blots;<br>anti-Actin (Santa Cruz, sc-47778) were used at 1:5000-fold dilution for western blots;<br>anti-Strep (IBA, 2-1509-001) were used at 1:4000-fold dilution for western blots;<br>anti-ALFA (NanoTag, N1505) were used at 1:2000-fold dilution for western blots;<br>anti-H3 (Abcam, ab1791) were used at 1:4000-fold dilution for western blots. |
| Validation | See Methods, all antibodies used in this study are commercially available and have been validated by the manufacturer. Antibody dilutions were used as described above and were optimized with several experiments before the experiment that is reported in this study. Appropriate positive and negative control samples were used in each experiment to ensure antibody specificity. |

# Plants

| | |
|---|---|
| Seed stocks | The Arabidopsis thaliana Columbia (Col-0) ecotype was used as the main experimental organism and were validated from NASC and GABIKat collection. Col-0 (N70000), fd-3 (SALK_054421), fd-10 (GK_290E08). |
| Novel plant genotypes | Corresponding Col-0, fd-3 or ft-10 mutant plants were grown in the greenhouse under LD conditions and were transformed by the floral dip method using Agrobacterium tumefaciens strain GV3101. The resulting transgenic T1 seeds were screened on half-strength MS medium with supplemented hygromycin for 7 LDs and then they were transferred to soil for the measurement of flowering time. |
| Authentication | Primers used for plasmid construction for transgenic lines are listed in Supplementary information Table 4. |

