## [Peer Review File · Nature]

Florigen activation complex forms via multifaceted assembly in Arabidopsis

Corresponding Author: Dr George Coupland

Version 0:

Reviewer comments:

Referee #1

(Remarks to the Author)

The paper describes a structural analysis of florigen activation complex (FAC), a concept published many years ago and important for the transition to flowering, and suggests a different mechanism of FAC assembly. It is well written, but very detailed and there are aspects that the general audience of Nature will find particularly inaccessible. For example Fig. 1 and 2 are likely to be understood only by biologists with familiarity of shoot apex architecture. The co-localization of FT and FD is most convincing in axillary meristems, but the relevance of these spatiotemporal patterns of the FAC complex remains mostly speculative (Discussion).

Lack of detection of either FT-FD or FT-chromatin interaction in vivo, even after crosslinking, meant a reliance on in vitro and nuclear proteome immunoprecipitation to study FAC. It is still not clear, given the model, why the in vivo interaction is so hard to detect. Reliance on in vitro analysis means some of the findings may not be relevant in vivo, eg. three positively charged arginine residues (R166, R173 and R174) in the FT tail will always bind to negatively charged DNA in vitro. Fig. 3k – why do the FT-strep proteins differing in point mutations run at different sizes?

FD-Mu1 is used to show DNA-dependency of the FT interaction. This is based on FD-Mu1 showing impaired DNA binding, but it also has strongly attenuated nuclear localization in vivo (supplementary text). Since nuclear protein preps of FD WT/Mut were used could lower FD contribute to changes interpreted as the DNA-dependency of the FT interaction? Functionally important in vivo phosphorylation of FD at threonine (T) 282 in 3HA-mCherry-FD proteins supports interaction with 14-3-3 proteins. Mutation of T282 to A resulted in accumulation of large puncta and these were independent of FT. The model whereby 14-3-3 interaction prevents FD oligomerization is compelling, but could anything else be affected by the T282A mutation? This is crucial to know to be able to conclude that 14-3-3 binding is responsible for the degree of enrichment in the ChIP experiments. What is the in vivo evidence that 14-3-3's bind FD to ensure only dimer formation? In summary, this is a multi-faceted study continuing the important story of how FT protein induces flowering. Transmissible florigen and how it activates flowering is of general interest, however, this paper describes experiments that form multiple smaller stories that advance previous understanding rather than combine into one general message.

Referee #2

(Remarks to the Author)

The manuscript by Gao and colleagues present are large set of biochemical and in vivo data shedding new light on the Florigen Activation Complex, the essential regulatory switch for floral transition in diverse plants. The first two figures describe mRNA and protein localization studies of key regulators, including FT, FD and TFL1 with high resolution. Given the wealth of information already available on this aspect, this part is not particularly novel.

Figure 3 presents the core message of the manuscript, namely that the interaction of the florigen FT with the executive transcription factor FD is mediated by 14-3-3 proteins and DNA. Despite prior work in rice describing the role of 14-3-3 proteins in the FT-FT interaction, this part adds important insights by demonstrating that the unstructured c-terminal tail of FT likely is required for stabilizing DNA contacts.

Figure 4 and 5 round off the manuscript focusing on the phase separation properties of FD and how 14-3-3 proteins suppress this behavior to enhance their biological activity.

Overall, this is impressive work and provides key insights into the biochemistry of the most important developmental

transition in plants. Before publication, I would like to see the following issues to be addressed:

1. Figure labelling is inconsistent and confusing at places– some individual panels have labels, others presumably share them their neighbors
2. The expression patterns described in Figures 1 and 2 require quantification. Especially FT expression in axillary meristems at LD13 is currently not convincing, since there is scattered background throughout the section (Fig 2 panels e; e1) and only a single section is shown.
3. The main novelty of the entire first part lies in the identification of highly specific expression domains of FT, FD and TFL at the apex, however there are no attempts to test their role in patterning the tissue. To elucidate the relevance of FT,FD, TFL expression, inducible misexpression experiments using for example the CUC, CLV3, WUS promoters with Lhg4-GR system would be required. This would allow the authors to analyze if the spatiotemporal expression patterns described actually matter. The authors could consider to remove this part and rather develop it into an independent manuscript with functional follow up.
4. I am confused about the use of the MBP-FD(T282E) variant for all gel filtration experiments – this is not explained in the text and the likely reason only becomes clear much later in the manuscript.
5. I find the differential IP of FD and GRF7 with FT Mu1 very surprising, given that in this setting the DNA interaction removed by Mu1 should not play a role. Are there any alternative explanations – please discuss.
6. I could not quite follow the choice of references here and there – e.g. ref. 29, please check.
7. I can't see the reason for calling protein extracts “proteome extracts”.

Referee #3

(Remarks to the Author)

Gao et al. thoroughly demonstrated how the florigen activation complex (FAC) formed at the shoot apical meristem (SAM) during the floral transition in Arabidopsis. The authors utilized biochemistry, imaging, and molecular genetics to nicely elucidate the function of 14-3-3 within the FAC. We know that florigen moves to the SAM and forms the FAC with FD and 14-3-3 to induce floral transition. Although there was a proposed model for the formation of the FAC at the SAM, we still had limited knowledge of how the FAC is formed during the floral induction. This work provides detailed molecular insights into the formation of the FAC at the SAM during the developmental phase transitions, especially revealing the multiple roles of 14-3-3 in this complex. The flow is logical, the experiments were properly finished, and they provided extensive evidence on this matter. Their finding is of great interest to audiences in general plant developmental biology. Below are specific comments I would like them to address.

Specific comments:

- 1) Their previous paper (Corbesier et al. 2007) indicated that fusing GFP to FT (FT-GFP) hindered the FT movement at the SAM. I have a comment about their description of FT-mVenus localization patterns (lines 108 and after). Although FT-mVenus complemented ft mutant flowering phenotypes, it does not prove that FT-mVenus can behave like endogenous FT. Can the authors discuss whether changing from GFP (eGFP) to mVenus (which is another variant of GFP – does monomeric form help?) mitigates the possible issue of hindering FT movement? Do the authors have any evidence of the improvement of FT movement by using mVenus? Abe et al. Development 2019 showed that FT binds to FD around the organizing center of the SAM but was relatively weak around the rib meristem area, where the authors showed the overlap of FD and FT in this manuscript. In Fig. 1, FT-mVenus does not seem visible in the area where FD is highly expressed and also bound to FT in Abe et al. (around the organizing center and L2-L3 layers) in 12- to 13-day-old LD-grown plants. I wonder FT-mVenus may still be less mobile. The authors should cite Abe et al. Development 2019 (<https://doi.org/10.1242/dev.171504>) and discuss their results with this paper's results and their limitations.
- 2) In supplementary methods, the information on plasmid construction is very short. Although the authors provided the sequence information of primers used for generating the constructs, it would be more helpful if they explained how they make these constructs in more detail. For example, they should mention the sizes of promoters they used for the construction of these plasmids. Also, the references or the sources of fluorescence/epitope tags should be provided.
- 3) Some of the flowering time experiments have been performed using T1 generation plants in this manuscript. Because they need to screen the transformants first on the plates to study the flowering time of T1 plants, the authors need to describe how exactly they performed the experiments.

Version 1:

Reviewer comments:

Referee #1

(Remarks to the Author)

The revised version addresses many of the issues raised by the reviewers. For example, the schematic in Fig. 1 helps interpretation of the microscopy images considerably. However, the central conclusion of the paper – multi-faceted assembly of florigen activation complex- is supported in vivo only by Fig.1j, k: but does this distance-to-shoot apex quantification provide sufficient resolution to conclude co-localization? Based on the images in Fig. 1, it looks like most of FD/FT is present in distinct cells rather than co-localizing with each other.

The gel filtration in Fig. 3e shows faint bands indicating FT interaction with FD/GRF7/DNA but most of FT elutes without binding – so the interaction is low affinity even in vitro. Thus, the main advance in this paper is less about assembly of florigen activation complex, and more on the role of 14-3-3s in orderly assembly of FD for DNA binding.

Fig. 4o. Chip-PCR y-axis - % of total input x 10⁻³, these enrichment values are so low they cannot be used to conclude any enrichment of FD on these targets.

Effect of T282A mutation- Data on new mutation were added (A283) but the main text is lacking information about the residue (eg. localization in SAP motif).

Referee #2

(Remarks to the Author)

The revised version of the manuscript is a major step forward and much more convincing and I support publication after minor revisions.

The two parts, namely the characterization of FT and FD expression and the mechanistic elucidation of the FD-GRF-FT-DNA complex, are still fairly disconnected, but are both of high quality. Questions to the relevance of FT RNA accumulation in the apex remain. Similarly, the marginal overlap of FT protein with expression domains of important target genes such as AP1 and LEAFY should be more clearly discussed. Looking at the data, it appears that FT stabilizes the interaction of the FD/GRF complex with chromatin. Maybe the authors could also discuss this angle.

Referee #3

(Remarks to the Author)

In this revised manuscript, the authors successfully responded to this reviewer's previous comments, and I don't have any further comments. The authors have provided extensive, multifaceted evidence on how the FT mechanistically initiates the floral transition at the shoot apical meristem.

Version 2:

Reviewer comments:

Referee #1

(Remarks to the Author)

The authors have made a considerable effort to address outstanding concerns. However, the novelty of this paper is a refinement of the original FAC discovery, so the in vivo interactions are paramount. The concept that FAC forms is central to the conclusions- eg.Line 150 Collectively, our protein and mRNA analyses demonstrate that FT is present in the same cells as FD and 14-3-3 proteins at different stages of the flowering process to enable formation of the FAC. Yet the in vivo co-localization of FT and FD in FAC remains to be convincingly demonstrated. At least this should be made very clear to the reader as they read the title and abstract. This limitation comes together with the relatively small overlap observed between FT and the expression domains of the proposed target genes (mentioned by reviewer 2).

(Remarks on code availability)

Referee #2

(Remarks to the Author)

The authors have addressed all my remaining concerns.

(Remarks on code availability)

-

Version 3:

Reviewer comments:

Referee #1

(Remarks to the Author)

Please thank the authors for the changes. They address my concerns and make the paper much stronger.

(Remarks on code availability)

Referee #1:

The paper describes a structural analysis of florigen activation complex (FAC), a concept published many years ago and important for the transition to flowering, and suggests a different mechanism of FAC assembly. It is well written, but very detailed and there are aspects that the general audience of Nature will find particularly inaccessible.

For example Fig. 1 and 2 are likely to be understood only by biologists with familiarity of shoot apex architecture. The co-localization of FT and FD is most convincing in axillary meristems, but the relevance of these spatiotemporal patterns of the FAC complex remains mostly speculative (Discussion).

We appreciate the referee's suggestions on the text and Figures 1 and 2. To enhance understanding for a broader audience, we have included as a first panel (Figure 1a) a schematic illustrating the timing of developmental transitions in Arabidopsis. We have also edited the manuscript to reduce detail and emphasize the main points. We have moved some of the detail to the Supplementary Information. We have reorganized the RNAscope data in Figure 2 to emphasize the overlap between *FD* and *FT* mRNAs in young floral primordia. In particular, we now show the patterns of *FD* and *FT* mRNAs relative to the

mRNA of the floral meristem identity gene *AP1*, demonstrating their expression patterns in early floral primordia. We have moved some of the original RNAscope images to the Supplementary Information to reduce the detail in the figure. To emphasize the co-localization of the FT and FD proteins detected by confocal microscopy, we have also shown higher magnification images in Figure 1 highlighting the co-localization of FT and FD proteins in the shoot meristem at 12LD and 13LD. We have included more discussion of the relevance of the transcription of *FT* in floral primordia, which has not previously been described, and discuss on p.12 this in the context of the effects of *ft* mutations on inflorescence branch number and floral organ development, and the transcription of *FT-LIKE* genes in the apex of rice plants.

Lack of detection of either FT-FD or FT-chromatin interaction in vivo, even after crosslinking, meant a reliance on in vitro and nuclear proteome immunoprecipitation to study FAC. It is still not clear, given the model, why the in vivo interaction is so hard to detect. Reliance on in vitro analysis means some of the findings may not be relevant in vivo, eg. three positively charged arginine residues (R166, R173 and R174) in the FT tail will always bind to negatively charged DNA in vitro.

The reviewer is right that in the original manuscript we used semi-*in vivo* data to test the FT-chromatin or FT-FD interaction. We have repeatedly tried to directly measure FT binding to DNA or to FD entirely *in vivo* but have not been able to detect this. We believe that this difficulty is due to the very low abundance of the protein in apices after movement from leaves or after transcription in precise spatial domains at specific times. Moreover, the interactions between FT and the FD-14-3-3-DNA complex might be weak and transient. Nevertheless, in rice meristems, where more material can be collected at the appropriate time, FT association with DNA was detected by ChIP-PCR *in vivo*, supporting the idea that our semi-*in vivo* mechanistic data are relevant *in vivo*. A further complication is that we observed a gradual increase in *FD* transcript levels during the floral transition (Extended data Fig. 2). These changes may affect the accuracy of detecting the recruitment strength of FT WT and mutant proteins, as FT^{mu1} flowered much later than the FT^{WT} and FT^{mu2} transgenes thereby causing them to maximally co-accumulate with FD at different times after germination. The semi-*in vivo* experiments addressed these issues by using quantified amounts of *in vitro* purified FT proteins and chromatin or nuclear proteins extracted from plant apices to ensure higher and identical inputs of FT protein for interaction with FD, 14-3-3 proteins or chromatin. This approach is similar to the well-accepted DAP-seq method developed by Joe Ecker's group. We plan to persist with the *in vivo* approach in the future, for example by using heterologous promoters to drive FT and FD expression, but considering the technical difficulty of these experiments and the need to generate appropriate transgenic plants, the experiments would take longer than the proposed revision timeline allows.

We agree that it is possible that in some cases, *in vitro* analysis might not fully represent *in vivo* conditions. However, our *in vitro* data showed that the WT FT protein and therefore the positively charged tail does not always bind DNA, particularly when the protein is present alone (Extended data Fig. 4j,k). Instead, its recruitment to the FAC always requires the pre-assembly of the FD-14-3-3-DNA complex. These *in vitro* observations are highly consistent with our semi-*in vivo* co-immunoprecipitation experiments, which involved co-immunoprecipitation of FD or the non-binding FD mutant (mu1), GRF7 and FT WT or mutant proteins, as well as with our semi-*in vivo* ChIP PCRs conducted using FT WT and mutant proteins. The *in vivo* situation might be more complex, but we believe that these semi-*in vivo* experiments provide an important basis for explaining the mutant phenotypes that we observe *in vivo*.

Fig. 3k – why do the FT-strep proteins differing in point mutations run at different sizes?

The reviewer is right that in Figures 3k and 3i the FT protein and the two mutant proteins migrate at different positions after electrophoresis. Numerous factors can affect protein migration in SDS-PAGE gels. In our case, we assume that SDS binds to the positively charged arginines in the tail of the FT WT protein, potentially slowing its migration from the negative to the positive electrode. However, mutating those arginines to neutral alanines significantly reduces the positive charge in the FT mu1 protein, thereby accelerating its migration. On the other hand, SDS binds effectively to hydrophobic regions of proteins via interactions between its hydrophobic tail and hydrophobic side chains. We assume that mutating phenylalanine (F101) to alanine reduces hydrophobicity in the FT mu2 protein, thereby slowing its migration from the negative to positive electrode. The effect is entirely reproducible as shown in the input and IP panels of both panels for the co-IPs (Figure 3), and consistently detected *in vivo* (Supplementary Table 1_sheet-3). Also, there is no apparent molecular size change when running these proteins by gel-filtration in natural conditions (compare Figures 3e,h and i).

FD-Mu1 is used to show DNA-dependency of the FT interaction. This is based on FD-Mu1 showing impaired DNA binding, but it also has strongly attenuated nuclear localization in vivo (supplementary text). Since nuclear protein preps of FD WT/Mut were used could lower FD contribute to changes interpreted as the DNA-dependency of the FT interaction?

The reviewer is right that Mu1 does influence DNA binding and nuclear localization. In the analysis of nuclear localization by confocal microscopy (Extended Fig. 5c-f), we used T1 lines expressing WT 2HA-mVenus-FD and -Mu1. We noticed that T3 homozygous FD Mu1 accumulated to a lower level in nuclei than FD WT, even when total Mu1 protein levels were higher (Extended data Fig. 5g,h (newly added panels)), supporting the confocal image in Extended Figure 5d showing that Mu1 impaired nuclear localization. Therefore, for the nuclear co-IPs shown in Figure 3M, we used another tagged line (*gFD::3HA3Flag-FD*) that expressed the Mu1 protein at a much higher overall level so that the amount of input protein in the nucleus was very similar to that of WT FD (Extended data Fig. 5i,j (newly added panels)).

Functionally important in vivo phosphorylation of FD at threonine (T) 282 in 3HA-mCherry-FD proteins supports interaction with 14-3-3 proteins. Mutation of T282 to A resulted in accumulation of large puncta and these were independent of FT. The model whereby 14-3-3 interaction prevents FD oligomerization is compelling, but could anything else be affected by the T282A mutation? This is crucial to know to be able to conclude that 14-3-3 binding is responsible for the degree of enrichment in the ChIP experiments. What is the in vivo evidence that 14-3-3's bind FD to ensure only dimer formation?

The reviewer is right that we used T282A to prevent phosphorylation and interaction with 14-3-3 and interpreted the resulting oligomerization as a consequence of the mutant protein not interacting with the 14-3-3. Although it is conceivable that the mutation might have an additional effect on the protein *in vivo*, we did find that *in vitro* the phosphomimic T282E mutant also oligomerized in the absence of the 14-3-3 and was soluble in its presence.

Nevertheless, to address the possibility that T282A has additional effects *in vivo*, we have now made another mutation (A283W) to antagonize the interaction with 14-3-3 at the recognition site (SAP motif). This mutation left the T282 residue intact and it was combined with the T282 phosphomimetic mutation (Mu9^{T282E/A283W}) to ensure that if kinase recognition was disrupted it would still behave as if phosphorylated. *In vitro* the mutant protein did not interact with 14-3-3 protein GRF7 (Extended data Fig. 8d,e). Furthermore, this mutant protein also formed large droplets even when GRF7 was present (Extended data Fig. 8f,g), as we found for FD^{T282A} mutant protein. We then expressed the Mu9^{T282E/A283W} FD protein in *fd-3* plants. Consistently, the FD mutant transgene did not rescue the late-flowering phenotype of *fd-3* mutants, and again, large nuclear speckles were observed at the shoot apical meristem of these FD mutants (Extended data Fig. 8h-j). These new data further confirmed the idea that 14-3-3 interaction prevents oligomerization of FD, and this is not due to an additional effect on the protein *in vivo* of mutating the T282 phosphorylation site.

Concerning the *in vivo* evidence that 14-3-3's bind FD to ensure only dimer formation. Directly measuring FD dimerization with 14-3-3 proteins *in vivo* is challenging, but there is indirect evidence to support it. Firstly, 14-3-3 proteins are dimeric, and therefore the 14-3-3 dimer has two binding sites that can potentially accommodate two phosphorylated FD substrates in a dimer. Secondly, ten 14-3-3 proteins were repeatedly identified *in vivo* by our IP-MS analysis of FD interactors (Supplementary Table 2). Together with our precise SEC-MALS analysis *in vitro*, all of these findings strongly suggest that FD and 14-3-3 proteins form stable tetramers (composed of two 14-3-3 and two FD proteins) *in vivo*. Nevertheless, since dimerization of the leucine zipper of bZIP transcription factors is essential for their DNA binding, it can be measured indirectly by assessing DNA binding *in vivo*. Therefore, we did ChIP-qPCR *in vivo* using one of the monomer FD mutants, Mu3^{L242A}. We did find that *in vivo*, Mu3 had much lower DNA binding activity compared with FD WT protein, although they had comparable mRNA and protein levels (Extended data Fig. 9f-i (newly added data)). These new data suggest that at least the FD C-terminus forms dimers *in vivo* which requires the formation of the leucine zipper, and that binding of dimeric 14-3-3 proteins strongly stabilizes the dimerization of FD C-terminal by preventing its oligomerization, both *in vivo* and *in vitro*, thus ensuring its DNA binding activity.

In summary, this is a multi-faceted study continuing the important story of how FT protein induces flowering. Transmissible florigen and how it activates flowering is of general interest, however, this paper describes experiments that form multiple smaller stories that advance previous understanding rather than combine into one general message.

We are pleased that the reviewer recognizes the importance of studying the role of FT protein in flowering. We have tried to address the issue of integrating the 3 major advances of the paper (two phases of expression in the apex based on movement and specific transcriptional patterns that allow spatio-temporal overlap of FT, FD and 14-3-3 expression; identification of a new interface on the FT protein that interacts with DNA; multiple roles for the 14-3-3 protein in the complex independent of simply recruiting FT) into one major advance in the Summary and Discussion section. We believe that this paper will be influential by advancing understanding of FT regulation and its biochemical function in the complex context of the FAC.

Referee #2 (Remarks to the Author):

The manuscript by Gao and colleagues present are large set of biochemical and in

vivo data shedding new light on the Florigen Activation Complex, the essential regulatory switch for floral transition in diverse plants. The first two figures describe mRNA and protein localization studies of key regulators, including FT, FD and TFL1 with high resolution. Given the wealth of information already available on this aspect, this part is not particularly novel.

We were pleased that the reviewer recognizes the amount of biochemical data we have provided on the Florigen Activation Complex and its importance. However, we believe that the data shown in Figures 1 and 2 are also novel and important. The data in Figure 2 show that *FT* has a second phase of transcription and that occurs adjacent to axillary meristems and in floral primordia. This apical pattern of transcription had not previously been detected, and had been excluded in earlier models of FT action because *FT* transcription was assumed to be limited to the vasculature of the leaves and stem. Using a recently developed multiplex RNA *in situ* hybridization technology we also position *FT* mRNA in the primordium relative to *FD*, *TFL1* and *AP1* mRNAs. We believe that the boundary pattern of *FT* transcription we describe will be important in future models of early floral development, and in integrating models of floral transition in different species, and we have included these points in the Discussion along with a comparison with the *FT-LIKE* genes of rice. In our opinion, Figure 1 is also important in demonstrating that FT protein moves into the meristem when the protein is expressed from its endogenous regulatory sequences and overlaps convincingly with cells expressing FD, because most previous analyses relied on overexpression from heterologous promoters. Moreover, Figure 1 shows that large proteins that cannot move from the vasculature are nevertheless detected in the axillary meristem and primordium, distinguishing movement from the second round of transcription. Therefore, we believe that Figure 1 and 2 are tightly connected in the logical flow of the paper, and link to the subsequent biochemical analysis of FT-FD interaction in the FAC.

Figure 3 presents the core message of the manuscript, namely that the interaction of the florigen FT with the executive transcription factor FD is mediated by 14-3-3 proteins and DNA. Despite prior work in rice describing the role of 14-3-3 proteins in the FT-FD interaction, this part adds important insights by demonstrating that the unstructured c-terminal tail of FT likely is required for stabilizing DNA contacts. Figure 4 and 5 round off the manuscript focusing on the phase separation properties of FD and how 14-3-3 proteins suppress this behavior to enhance their biological activity.

Overall, this is impressive work and provides key insights into the biochemistry of the most important developmental transition in plants. Before publication, I would like to see the following issues to be addressed:

We appreciate the reviewer's comments on the importance and quality of our biochemical work, and support of publication of the manuscript after suitable revision.

1. Figure labelling is inconsistent and confusing at places– some individual panels have labels, others presumably share them their neighbors

Thanks, we have now numbered all of the panels in Figure 1.

2. The expression patterns described in Figures 1 and 2 require quantification. Especially FT expression in axillary meristems at LD13 is currently not convincing, since there is scattered background throughout the section (Fig 2 panels e; e1) and only a single section is shown.

We have entirely reorganized Figure 2 providing new RNAscope images for floral primordia and the shoot apex. Considering the comments of the reviewer, we removed Figure 2 panels e;e1 as part of this process. We have also provided more than one section in the Extended Fig. 3. In addition, we quantified the data in Figure 1 and Figure 2 as requested by the reviewer, and show the quantification data in each figure.

3. The main novelty of the entire first part lies in the identification of highly specific expression domains of FT, FD and TFL at the apex, however there are no attempts to test their role in patterning the tissue. To elucidate the relevance of FT, FD, TFL expression, inducible misexpression experiments using for example the CUC, CLV3, WUS promoters with Lhg4-GR system would be required. This would allow the authors to analyze if the spatiotemporal expression patterns described actually matter. The authors could consider to remove this part and rather develop it into an independent manuscript with functional follow up.

We thank the reviewer for these comments on the floral expression pattern. We would prefer to keep the floral expression patterns because of their novelty and the overlap in FT and FD transcriptional patterns supports the concept that the FAC is formed in primordia as well as in the SAM region. However, the inducible misexpression experiment proposed would take longer than the proposed revision, because we have no materials yet available. We cannot therefore provide these data, but in support of the significance of these patterns, we now cite previous work describing branching and floral defects in *ft* mutants, and previous links to the *BOP* genes, which are also involved in boundary formation in flowers.

4. I am confused about the use of the MBP-FD(T282E) variant for all gel filtration experiments – this is not explained in the text and the likely reason only becomes clear much later in the manuscript.

Thanks. We have now made this clearer earlier in the text and in the supplementary information. The protein is purified from *E. coli* where it is not phosphorylated on T282, so the T282E variant is used as a phosphomimetic variant. The MBP fusion helps solubilize the protein in the absence of the 14-3-3 and therefore enable its purification.

5. I find the differential IP of FD and GRF7 with FT Mu1 very surprising, given that in this setting the DNA interaction removed by Mu1 should not play a role. Are there any alternative explanations – please discuss.

We considered that the reviewer refers here to Figure 3, and the size exclusion chromatography experiments shown in Panels e, h and i, which are related to the IPs shown in panels k,l and m, although the IPs do not include data on the 14-3-3s. The reviewer is right that in our model the FT Mu1 impairs interaction with DNA, and this also prevents the interaction with FD and GRF7. We interpret this to mean that FT assembles into the complex with FD and GRF7 on DNA, and that without the interaction with DNA, the binding of FT to the FD-GRF7 complex is too weak to detect. For example, in size exclusion chromatography, we do not detect FT interacting with FD-GRF7 unless we add DNA, Fig3d vs 3e. We have tried to make this clearer on page 7.

6. I could not quite follow the choice of references here and there – e.g. ref. 29, please check.

We have checked the references. Reference 29 does include a deletion of the SAP domain with which 14-3-3 interacts but in AREB3, not FD. We have made this clearer by adding “and related Group A bZIPs”. We have also included the reference of Park et al there who showed something similar for the FD orthologue in tomato. We deleted the reference of Romera-Branchat et from here (no. 16) which we agree did not seem relevant. We have also checked the references throughout, and included the reference mentioned by reviewer 3.

7. I can't see the reason for calling protein extracts “proteome extracts”.

Thanks, we changed this to “protein extracts” in the text.

Referee #3 (Remarks to the Author):

Gao et al. thoroughly demonstrated how the florigen activation complex (FAC) formed at the shoot apical meristem (SAM) during the floral transition in Arabidopsis. The authors utilized biochemistry, imaging, and molecular genetics to nicely elucidate the function of 14-3-3 within the FAC. We know that florigen moves to the SAM and forms the FAC with FD and 14-3-3 to induce floral transition. Although there was a proposed model for the formation of the FAC at the SAM, we still had limited knowledge of how the FAC is formed during the floral induction. This work provides detailed molecular insights into the formation of the FAC at the SAM during the developmental phase transitions, especially revealing the multiple roles of 14-3-3 in this complex. The flow is logical, the experiments were properly finished, and they provided extensive evidence on this matter. Their finding is of great interest to audiences in general plant developmental biology. Below are specific comments I would like them to address.

We appreciate the reviewer's interest in our work and the supportive comments.

Specific comments:

1) Their previous paper (Corbesier et al. 2007) indicated that fusing GFP to FT (FT-GFP) hindered the FT movement at the SAM. I have a comment about their description of FT-mVenus localization patterns (lines 108 and after). Although FT-mVenus complemented *ft* mutant flowering phenotypes, it does not prove that FT-mVenus can behave like endogenous FT. Can the authors discuss whether changing from GFP (eGFP) to mVenus (which is another variant of GFP – does monomeric form help?) mitigates the possible issue of hindering FT movement? Do the authors have any evidence of the improvement of FT movement by using mVenus?

The reviewer is right that the fusion of FT to mVenus appears to complement the *ft* mutation better than the fusion to eGFP. We think this is probably due to the monomeric status of mVenus compared to the dimeric form of eGFP, which have been well established, and would cause a substantial reduction in the molecular size of the fusion protein *in vivo*. We have not thoroughly compared the effect of fusions to these two forms *in planta*, preferring to focus our attention here on the most modern and sensitive fluorescent proteins. Direct comparison with the literature is difficult because many of the previous experiments were

done using heterologous promoters, whereas here we focused entirely on using the endogenous transcriptional signals of *FT*, which are now much better understood than in 2007 when the first paper was published.

Abe et al. Development 2019 showed that FT binds to FD around the organizing center of the SAM but was relatively weak around the rib meristem area, where the authors showed the overlap of FD and FT in this manuscript. In Fig. 1, FT-mVenus does not seem visible in the area where FD is highly expressed and also bound to FT in Abe et al. (around the organizing center and L2-L3 layers) in 12- to 13-day-old LD-grown plants. I wonder FT-mVenus may still be less mobile. The authors should cite Abe et al. Development 2019 (<https://doi.org/10.1242/dev.171504>) and discuss their results with this paper's results and their limitations.

Thanks for this. The reviewer is right that we should have cited the Abe paper, and we apologize for that oversight. We have now included this paper (Ref 12). The reviewer is also right to point out that we detect the overlap between FT and FD in the lower meristematic region. We have emphasized this by including a panel in revised Figure 1 at higher magnification. This region of overlap might correspond to the lower organizing centre or as the reviewer mentions the rib region. The main issue for us is that the images establish that there is an overlap between the movement of FT when expressed from its own regulatory sequences and the expression of FD at the apex, and therefore supports the significance of the FAC. It is possible that FT- mVenus moves further in the meristem but is below the detection of the confocal microscopy methods we used. As the reviewer mentions, Abe et al detected FT in the organizing centre higher in the meristem than we did, and used a smaller translational fusion to FT, but they also overexpressed the gene from the inducible HS promoter. So, the wider detection pattern might be due to higher levels of the protein as well. We have more thoroughly considered these issues in the Discussion on p.11-12, and have included a higher magnification image in the revised Figure 1 to show the overlap between the FT-mVenus and mCherry-FD proteins in the SAM.

2) In supplementary methods, the information on plasmid construction is very short. Although the authors provided the sequence information of primers used for generating the constructs, it would be more helpful if they explained how they make these constructs in more detail. For example, they should mention the sizes of promoters they used for the construction of these plasmids. Also, the references or the sources of fluorescence/epitope tags should be provided.

We have extended the information on the construction of plasmids, including the length of promoters used, and the length of the 3' tail. We have also provided the epitope tag information. We also cite the references for fluorescence/epitope tags, such as mVenus, mScarlet1 and ALFA tags.

3) Some of the flowering time experiments have been performed using T1 generation plants in this manuscript. Because they need to screen the transformants first on the plates to study the flowering time of T1 plants, the authors need to describe how exactly they performed the experiments.

We have included in the Methods section a description of how the flowering times of T1 plants were determined. We have also repeated several of the flowering-time experiments with established homozygous T3 lines. For example, in Figs 3o and 5j. The flowering-time data for these homozygous T3 plants were also deposited in the sheets of Supplementary Table 1.

Referee #1:

The revised version addresses many of the issues raised by the reviewers. For example, the schematic in Fig. 1 helps interpretation of the microscopy images considerably. However, the central conclusion of the paper – multi-faceted assembly of florigen activation complex- is supported *in vivo* only by Fig.1j, k: but does this distance-to-shoot apex quantification provide sufficient resolution to conclude co-localization? Based on the images in Fig. 1, it looks like most of FD/FT is present in distinct cells rather than co-localizing with each other.

We appreciate the referee's comments on our revised version, and are pleased that the reviewer found that the revisions addressed many of the points raised by the reviewers on the original version. The reviewer questions the evidence for multi-faceted assembly of the florigen complex *in vivo*, and mentions specifically the evidence that FT and FD are co-located. We provided evidence in Figure 1 of the original and revised versions that the FT protein that moves to the shoot meristem overlaps in its distribution with the pattern of expression of FD. The quantification of the images performed in the revised version and mentioned by the reviewer strengthened the argument that was previously based on the visual inspection of the images in, for example, Figure 1m. The resolution provided in the quantification is the distance in μm from the tip of the meristem, and clearly shows an overlap at 13 LDs in the FT and FD protein at a depth of 50 – 100 μm . To enhance comparison of the quantification in Fig. 1j,k and the visual inspection of Fig. 1m, we have added a scale bar in μm in Fig.1m. Moreover, at the subcellular level FD is a nuclear transcription factor, and FT was also previously shown to be nuclear and cytoplasmic, so the two proteins are expected to be present in the same cellular compartment within cells. The co-localization of these proteins in nuclei can also be seen visually in Figure 1m. Our demonstration for the first time in Figure 2 that FD and FT are co-transcribed in regions of the primordium, also supports the argument that in a second developmental context these proteins are present in the same cells. The reviewer is right to point out that FD is more broadly expressed at the shoot apex than FT, and therefore they only co-localize in a sub-set of cells. However, FD is known to have additional functions, such as interacting with the FT-related protein TFL1, so we did not expect co-localization in all cells in which they are expressed. The main point is to demonstrate that there are indeed cells in the shoot apical meristem and the primordium where both proteins are present and the florigen activation complex could form. The conclusion on the multi-faceted assembly of the complex does not only relate to FD and FT, but also to the 14-3-3 proteins, and we provide *in vivo* evidence supporting the model for their involvement. This includes co-immunoprecipitation of FD and 14-3-3 proteins and the demonstration that the mutant forms of FD that do not interact with the 14-3-3 form nuclear condensates *in vivo*.

The gel filtration in Fig. 3e shows faint bands indicating FT interaction with FD/GRF7/DNA but most of FT elutes without binding – so the interaction is low affinity even *in vitro*. Thus, the main advance in this paper is less about assembly of florigen activation complex, and more on the role of 14-3-3s in orderly assembly of FD for DNA binding.

The reviewer is right that the FT protein band in the FT/FD/GRF7/DNA complex looks weaker than the FT protein that does not bind. However, all of the FD protein present in the reaction is present in the FT/FD/GRF7/DNA complex, therefore we conclude that no more FT could incorporate into the complex because FD is limiting. We used the same amount of FD in Fig. 3e as in the other panels to standardize these experiments, and adding excess FT ensures that the maximum amount of complex is formed. Therefore, we do not think that the

presence of FT without binding means that the interaction is low affinity, but rather that it reflects the amount of FD and FT protein added to the reaction. In the gel, the FT protein looks weaker than FD because the protein is of much smaller molecular weight and so is not as heavily stained with Coomassie Blue for detection. Indeed, generally gel filtration is considered a relatively demanding assay for detecting biomolecular interactions, and FT protein recruitment was strongly detected in this method, suggesting a high affinity for the FD-14-3-3-DNA complex. This conclusion is also supported *in vitro* by our EMSA (gel shift) experiments, which provide evidence under relatively mild conditions (Extended Data Fig. 4 j,k).

Fig. 4o. Chip-PCR y-axis - % of total input x 10⁻³, these enrichment values are so low they cannot be used to conclude any enrichment of FD on these targets.

We appreciate the referee's recognition of the signal enrichment for our ChIP experiments in Fig. 4. Many transcription factors exhibit low enrichment at their binding sites due to their expression being limited to specific cells. FD is mainly expressed in shoot apex cells, but we used whole seedlings for our ChIP experiments to obtain sufficient starting material. We think that this might be one of the reasons for its relatively low enrichment ratio compared to the total input. However, our ChIP experiments were carefully performed in triplicate and the differences between WT FD and mutant proteins were significant specifically in target genes and not in negative control regions.

Effect of T282A mutation- Data on new mutation were added (A283) but the main text is lacking information about the residue (eg. localization in SAP motif).

We thank the referee for pointing out that the information on the new mutation was insufficient. In the previous revised version, we specified the mutated residue in the main text, and we illustrated its location in the SAP motif in Extended Data Fig. 8a. We have now made this clearer on lines 269-271 in the current revision:

“This conclusion was supported by generation of a second mutation in the SAP motif in which the alanine at position 283 (Extended Fig. 8a) was converted to tryptophan (A283W).”

Referee #2 (Remarks to the Author):

The revised version of the manuscript is a major step forward and much more convincing and I support publication after minor revisions.

We were pleased that the reviewer appreciated the improvements in the revised version and supports publication of the manuscript.

The two parts, namely the characterization of FT and FD expression and the mechanistic elucidation of the FD-GRF-FT-DNA complex, are still fairly disconnected, but are both of high quality.

We appreciated that the reviewer recognized the high quality of the parts of the manuscript describing FT and FD expression and the elucidation of the FD-GRF-FT-DNA complex. We have tried to minimize the apparent disconnection of these two parts by stressing that the

expression data demonstrate that all of the components of the complex are co-expressed at the shoot apex, and therefore the complex could form *in vivo*. For example, on lines 150-152; “Collectively, our protein and mRNA analyses demonstrate that FT is present in the same cells as FD and 14-3-3 proteins at different stages of the flowering process to enable formation of the FAC.”

Questions to the relevance of FT RNA accumulation in the apex remain. Similarly, the marginal overlap of FT protein with expression domains of important target genes such as AP1 and LEAFY should be more clearly discussed.

We thank the reviewer for raising these points. We mentioned in our response to the previous reviews that we could not directly address the relevance of *FT* transcription in the primordium with new experiments, but included more Discussion of these issues for example in the similarity with the *BOP* genes and inserted a sentence in the Discussion (line 391-393) to emphasize that more analysis will be required; “The precise relationship between these components in the boundary within the floral primordium will require future detailed genetic studies.” The reviewer raises an interesting point with the lack of overlap between FT protein and the expression of important target genes. We think that the most likely explanation in the primordium is that FT is required to initially activate *AP1* transcription in a strip of cells in the primordium, and that as these cells divide the expression domain of *AP1* widens but FT is not involved in maintaining its expression through this process, and we have added a sentence to this effect (line 385-388); “Moreover, although FT was proposed to interact with FD to activate *AP1* transcription⁴⁸, our data suggest that *FT* is only transcribed at the boundary of the *AP1* expression domain, and it may therefore be involved in initiating *AP1* transcription in the primordium rather than maintaining it.” Concerning FT in the lower part of the meristem, we have already inserted a sentence to mention that FT might be more widely distributed than we can detect in the confocal microscope and that the pattern we describe represents the minimum distribution of the protein (lines 376-379), so, considering that we are limited for space, we did not discuss in more detail the overlap between FT and target gene expression patterns at the SAM.

Looking at the data, it appears that FT stabilizes the interaction of the FD/GRF complex with chromatin. Maybe the authors could also discuss this angle.

The reviewer raises an interesting possibility, and we have inserted a sentence in the Discussion to raise this possibility. Line 423-424 “Our results also suggest that the recruitment of FT could stabilize the interaction of the FD–14-3-3 complex with chromatin.”

Referee #3 (Remarks to the Author):

In this revised manuscript, the authors successfully responded to this reviewer’s previous comments, and I don’t have any further comments. The authors have provided extensive, multifaceted evidence on how the FT mechanistically initiates the floral transition at the shoot apical meristem.

We appreciate the reviewer’s interest in our work and comments on the strength of the manuscript.

Reviewer 1

The authors have made a considerable effort to address outstanding concerns. However, the novelty of this paper is a refinement of the original FAC discovery, so the *in vivo* interactions are paramount. The concept that FAC forms is central to the conclusions- eg. Line 150 Collectively, our protein and mRNA analyses demonstrate that FT is present in the same cells as FD and 14-3-3 proteins at different stages of the flowering process to enable formation of the FAC. Yet the *in vivo* co-localization of FT and FD in FAC remains to be convincingly demonstrated. At least this should be made very clear to the reader as they read the title and abstract. This limitation comes together with the relatively small overlap observed between FT and the expression domains of the proposed target genes (mentioned by reviewer 2).

We thank the reviewer for recognizing the effort that we made in addressing the concerns expressed on the previous version. We also appreciate the advice of the reviewer on the need to increase the resolution of the determination of co-localization of FT and FD *in vivo*. As described above in detail, to answer this point, we devised an approach similar to that suggested to the editor by reviewer 2. Specifically, we segmented all nuclei expressing mScarlet1-FD in at least three meristems at each time point. The segmentation of nuclei

containing mScarlet1-FD is more reliable than trying to segment cells in these images, and because the FAC forms in nuclei these provide higher resolution data than segmenting whole cells. We then assessed the level of fluorescence of FT-mVenus in every nucleus expressing mScarlet1-FD. Finally, we placed each nucleus in a meristematic region – the central zone, organizing centre, peripheral zone or rib zone – based on the dimensions of *CLV3* and *WUS* expression in meristems at different stages of floral transition that we published previously (Bertran Garcia de Olalla et al 2024). We present these data in Figure 1J,K as a plot of mScarlet1-FD and FT-mVenus intensity in each segmented nucleus, and show that the median value for FT-mVenus fluorescence is higher at 11, 12 and 13 days than at 10 days in each meristematic region. Two further meristems for each time point are shown in Supplementary Figure 1a-p. Supplementary Figure 1 q,r plots the fluorescence of mScarlet1-FD relative to FT-mVenus in each nucleus analyzed at each time point in the meristems shown in Figure 1 (Supplementary Figure 1q) or in all meristems (Supplementary Figure 1r).

Also, as mentioned above we have improved the representation of the images in Figure 1b-i so that the fluorescent signals are more clearly visible. In particular, we removed the cell wall channel to more clearly visualize fluorescence of mScarlet1-FD and FT-mVenus. In addition, we optimized the look-up table dynamical range for the two proteins to more clearly show differences in fluorophore strength through the transition. These changes were made while keeping the parameters used to capture the images constant for all of them.

Concerning the expression domains, the number of FD-expressing nuclei containing FT is highest in the rib region, as expected from visual inspection of the images. However, quantification of the fluorescence shows that there are also nuclei containing FT-mVenus in nuclei expressing mScarlet1-FD in the organizing centre and even to a lower extent in the central zone. Therefore, there is substantial overlap between FT and FD in regions where target genes are expressed.

We consider these data provide the higher resolution analysis of the overlap between FD and FT requested by the reviewer by showing they are present broadly in the meristem in the same nuclei.

Referee #2 (Remarks to the Author):

The authors have addressed all my remaining concerns.

We appreciate the reviewer's support for the revisions that we made to the previous version. We also found the suggestion of the reviewer on how to increase the resolution of the analysis of co-localization of FT and FD in the confocal images very helpful, and have followed this suggestion in Figure 1j,k and Supplementary Figure 1a-p with the modification of focusing on segmented nuclei, as described in detail above. Moreover, we believe that the improved representation of the images in Figure 1f-i helps address the comment the reviewer makes about needing to look closely at the images to see the overlap.